# Amortized Control of Continuous State Space Feynman-Kac Model for Irregular Time Series

**Byoungwoo Park, Hyungi Lee, Juho Lee**
KAIST
{bw.park, lhk2708, juholee}@kaist.ac.kr

## Abstract

Many real-world datasets, such as healthcare, climate, and economics, are often collected as irregular time series, which poses challenges for accurate modeling. In this paper, we propose the Amortized Control of continuous State Space Model (ACSSM) for continuous dynamical modeling of time series for irregular and discrete observations. We first present a multi-marginal Doob's $h$-transform to construct a continuous dynamical system conditioned on these irregular observations. Following this, we introduce a variational inference algorithm with a tight evidence lower bound (ELBO), leveraging stochastic optimal control (SOC) theory to approximate the intractable Doob's $h$-transform and simulate the conditioned dynamics. To improve efficiency and scalability during both training and inference, ACSSM leverages auxiliary variable to flexibly parameterize the latent dynamics and amortized control. Additionally, it incorporates a simulation-free latent dynamics framework and a transformer-based data assimilation scheme, facilitating parallel inference of the latent states and ELBO computation. Through empirical evaluations across a variety of real-world datasets, ACSSM demonstrates superior performance in tasks such as classification, regression, interpolation, and extrapolation, while maintaining computational efficiency. Code is available at https://github.com/bw-park/ACSSM.

## 1 Introduction

State space models (SSMs) are widely used for modeling time series data (Fraccaro et al., 2017; Rangapuram et al., 2018; de Bézenac et al., 2020). However, SSMs typically assume evenly spaced observations, whereas many real-world datasets, such as healthcare (Silva et al., 2012), climate (Menne et al., 2015), are often collected as irregular time series. This poses challenges for accurate modeling. It provokes the neural differential equations (Chen et al., 2018; Rubanova et al., 2019; Li et al., 2020; Kidger et al., 2020; Zeng et al., 2023) for a continuous-time dynamical modeling of a time-series. Moreover, by incorporating stochastic dynamical models with SSM models, the application of continuous-dynamical state space models (CD-SSM) (Jazwinski, 2007) with neural networks have emerged which allows for more flexible modeling (Schirmer et al., 2022; Ansari et al., 2023). However, they typically rely on the Bayesian recursion to approximate the desired latent posterior distribution (*filtering/smoothing* distribution) involves sequential prediction and updating, resulting in computational costs that scale with the length of observations (Särkkä & García-Fernández, 2020).

Instead of relying on Bayesian recursion, we propose an Amortized Control of continuous State Space Model (ACSSM), a variational inference (VI) algorithm that directly approximates the posterior distribution using a measure-theoretic formulation, enable a simulation of continuous trajectories from such distributions. To achieve this, we introduce a *Feynman-Kac* model (Del Moral, 2011; Chopin et al., 2020), which facilitates a thorough sequential analysis. Within this formulation, we propose a multi-marginal Doob's $h$-transform to provide conditional latent dynamics described by stochastic differential equations (SDEs) which conditioned on a irregularly sampled, set of discrete observations. It extends the previous Doob's $h$-transform (Chopin et al., 2023) by incorporating multi-marginal constraints instead of a single terminal constraint.

Considering that the estimation of the Doob's $h$-transform is infeasible in general, we utilize the theory of stochastic optimal control (SOC) (Fleming & Soner, 2006; Carmona, 2016) to simulate the

conditioned SDEs by approximating the Doob's $h$-transform. It allow us to propose a tight evidence lower bound (ELBO) for the aforementioned VI algorithm by establishing a fundamental connection between the partial differential equations (PDEs) associated with Doob's $h$-transform and SOC. The Doob's $h$-transform often referred to as the *twist*-function in Sequential Monte Carlo (SMC) literature (Guarniero et al., 2017) to approximate the smoothing distributions. Building on this, (Heng et al., 2020) introduced an algorithm to approximate the twisted transition kernel directly, while a recent concurrent study (Lu & Wang, 2024) extended this approach to continuous-time settings. However, both studies primarily emphasize approximation methods rather than practical applications.

In practical situations, the computation of ELBO for a VI algorithm might impractical due to the instability and high memory demands associated with gradient computation of the approximated stochastic dynamics over the entire sequence interval (Liu et al., 2024; Park et al., 2024). To address this issue, we propose two efficient modeling approaches: 1) We establish amortized inference by introducing an auxiliary variable to the latent space, generated by a neural network encoder-decoder. It maps the high-dimensional time-series into a suitable low-dimensional space, allowing more flexible parameterization of the latent dynamics. Moreover, amortization allows the inference of the posterior distribution for a novel time-series sequence without relying on Bayesian recursion by incorporating the learned control function. 2) We leverage the simulation-free property, which enables closed-form sampling from intermediate latent marginal distributions that can be computed in a temporally parallel way. Additionally, we explore a more flexible linear approximation of the drift function in controlled SDEs to enhance the efficiency of the proposed controlled dynamics.

We evaluated ACSSM on several time-series tasks across various real-world datasets. Our experiments show that ACSSM consistently outperforms existing baseline models in each tasks, demonstrating its effectiveness in capturing the underlying dynamics of irregular time-series. Additionally, ACSSM achieves significant computational efficiency, enabling faster training times compared to dynamics-based models that rely on numerical simulations. A summary of the key concepts of ACSSM, along with related works, is provided in Appendix A. We summarize our contributions as follows:

- We extend the theory of Doob's $h$-transform to a multi-marginal cases. This indicates the existence of a class of conditioned SDEs that depend on future observations, where the solutions of these SDEs lead to the true posterior path measure within the framework of CD-SSM.

- We reformulate the simulation of conditioned SDEs as a SOC problem to approximating an impractical Doob's $h$-transform. By leveraging the connection between SOC theory and Doob's $h$-transform, we propose a variational inference algorithm with a tight ELBO.

- For practical real-world applications, we introduce an efficient and scalable modeling approach that enables parallelization of latent dynamic simulation and ELBO computation.

- We demonstrate its superior performance across various real-world irregularly sampled time-series tasks, including per-time point classification, regression, and sequence interpolation and extrapolation, all with computational efficiency.

**Notation**    Throughout this paper, we denote path measure by $\mathbb{P}^{(\cdot)}$, defined on the space of continuous functions $\Omega = C([0, T], \mathbb{R}^d)$. We sometimes denote with $\mathbb{P}$ the expectation as $\mathbb{E}_{\mathbb{P}}^{t, \mathbf{x}}[\cdot] = \mathbb{E}_{\mathbb{P}}[\cdot | \mathbf{X}_t = \mathbf{x}]$, where the stochastic processes corresponding to $\mathbb{P}^{(\cdot)}$ are represented as $\mathbf{X}^{(\cdot)}$ and their time-marginal distribution at time $t \in [0, T]$ is given by the push-forward measure $\mu_t^{(\cdot)} := (\mathbf{X}_t^{(\cdot)})_{\#}\mathbb{P}^{(\cdot)}$ with marginal density $\mathbf{p}_t^{(\cdot)}$. This marginal density represents the Radon-Nikodym derivative $d\mu_t^{(\cdot)}(\mathbf{x}) = \mathbf{p}_t^{(\cdot)}(\mathbf{x})d\mathcal{L}(\mathbf{x})$, where $\mathcal{L}$ denotes the Lebesgue measure. Additionally, for a function $\mathcal{V} : [0, T] \times \mathbb{R}^d \to \mathbb{R}$, we define the first and second derivatives with respect to $\mathbf{x} \in \mathbb{R}^d$ as $\nabla_{\mathbf{x}}\mathcal{V}$ and $\nabla_{\mathbf{xx}}\mathcal{V}$, respectively, and the derivative with respect to time $t \in [0, T]$ as $\partial_t \mathcal{V}$. For a sequence of functions $\{\mathcal{V}_i\}_{i \in [1:k]}$, we will denote $\mathcal{V}_i(t, \mathbf{x}) := \mathcal{V}_{i,t}$ and $[1 : k] = \{1, \cdots, k\}$. Finally, the Kullback-Leibler (KL) divergence between two probability measures $\mu$ and $\nu$ is defined as $D_{\text{KL}}(\mu|\nu) = \int_{\mathbb{R}^d} \log \frac{d\mu}{d\nu}(\mathbf{x})d\mu(\mathbf{x})$ when $\mu$ is absolutely continuous with respect to $\nu$, and $D_{\text{KL}}(\mu|\nu) = +\infty$ otherwise.

## 2    PRELIMINARIES

**Continuous-Discrete State Space Model**    Consider for a set of time-stamps (regular or irregular) $\{t_i\}_{i=0}^k$ over an interval $\mathcal{T} = [0, T]$, *i.e.*, $0 = t_0 \leq \cdots \leq t_k = T$. The CD-SSM assumes a

continuous-time Markov state trajectory $\mathbf{X}_{0:T}$ in latent space $\mathbb{R}^d$ is given as a solution of the SDE:

$$(\textit{Prior State}) \quad d\mathbf{X}_t = b(t, \mathbf{X}_t)dt + d\mathbf{W}_t, \tag{1}$$

where $\mathbf{X}_0 \sim \mu_0$ and $\{\mathbf{W}_t\}_{t \in [0,T]}$ is a $\mathbb{R}^d$-valued Wiener process that is independent of the $\mu_0$. Since $\mathbf{X}_t$ is Markov process, the time-evolution of marginal distribution $\mu_t$ is governed by a transition density, which is the solution to the Fokker-Planck equation assocaited with $\mathbf{X}_t$. This allows us to define a path measure $\mathbb{P}$ that represent the weak solutions of the SDE in (1) over an interval $[0, T]$[1].

For a measurement model $g_i(\mathbf{y}_{t_i}|\mathbf{X}_{t_i})$, we consider the case that we have only access to the realization of the (latent) observation process at each discrete-time stamps $\{t_i\}_{i \in [1:k]}$, *i.e.*, $\mathbf{y}_{t_i} \sim g_i(\mathbf{y}_{t_i}|\mathbf{X}_{t_i}), \forall i \in [1:k]$. In this paper, our goal is to infer the *classes of SDEs* which inducing the *filtering/smoothing* path measure $\mathbb{P}^\star := \mathbb{P}^\star(\cdot|\mathcal{H}_{t_k})$, the conditional distribution over the interval $[0, T]$ for a given $\mathbb{P}$ and a set of observations up to time $t_k$, $\mathcal{H}_{t_k} = \{\mathbf{y}_{t_i}|i \leq k\}$:

$$(\textit{Posterior Dist.}) \quad d\mathbb{P}^\star(\mathbf{X}_{0:T}|\mathcal{H}_{t_k}) = \frac{1}{\mathbf{Z}(\mathcal{H}_{t_k})} \prod_{i=1}^{K} g_i(\mathbf{y}_{t_i}|\mathbf{X}_{t_i})d\mathbb{P}(\mathbf{X}_{0:T}) \tag{2}$$

where the normalizing constant $\mathbf{Z}(\mathcal{H}_{t_k}) = \mathbb{E}_{\mathbb{P}}\left[\prod_{i=1}^{K} g_i(\mathbf{y}_{t_i}|\mathbf{X}_{t_i})\right]$ serve as a observations likelihood. The path measure formulation of the posterior distribution described in (2) referred to as *Feynman-Kac models*. See (Del Moral, 2011; Chopin et al., 2020) for a more comprehensive understanding.

## 3 CONTROLLED CONTINUOUS-DISCRETE STATE SPACE MODEL

In this section, we introduce our proposed model, ACSSM. First, we present the Multi-marginal Doob's $h$-transform, outlining the continuous dynamics for $\mathbb{P}^\star$ in Section 3.1. Then, in Section 3.2, we frame the VI for approximating $\mathbb{P}^\star$ using SOC. To support scalable real-world applications, we discuss efficient modeling and amortized inference in Section 3.3[2].

### 3.1 MULTI MARGINAL DOOB'S $h$-TRANSFORM

Before applying VI to approximate the posterior distribution $\mathbb{P}^\star$ in SOC, we first show that *a class of SDEs exists* whose solutions induce a path measure equivalent to $\mathbb{P}^\star$ in (2). This formulation provides a valuable insight for defining an appropriate objective function for the SOC problem in the next section. To do so, we first define a sequence of normalized potential functions $\{f_i\}_{i \in [1:k]}$, where each $f_i : \mathbb{R}^d \to \mathbb{R}_+$, for all $i \in [1:k]$,

$$f_i(\mathbf{y}_{t_i}|\mathbf{x}_{t_i}) = \frac{g_i(\mathbf{y}_{t_i}|\mathbf{x}_{t_i})}{\mathbf{L}_i(g_i)}, \tag{3}$$

where $\mathbf{L}_i(g_i) = \int_{\mathbb{R}^d} g_{t_i}(\mathbf{y}_{t_i}|\mathbf{x}_{t_i})d\mathbb{P}(\mathbf{x}_{0:T})$, for all $i \in [1:k]$ is the normalization constant. Then, we can observe that the potential functions $\{f_i\}_{i \in [1:k]}$ defined in (3) satisfying the normalizing property *i.e.*, $\mathbb{E}_{\mathbb{P}}[\prod_{i=1}^{k} f_i(\mathbf{y}_{t_i}|\mathbf{x}_{t_i})] = 1$ and $d\mathbb{P}^\star(\mathbf{x}_{0:T}|\mathcal{H}_{t_k}) = \prod_{i=1}^{k} f_i(\mathbf{y}_{t_i}|\mathbf{x}_{t_i})d\mathbb{P}(\mathbf{x}_{0:T})$ from (2). Now, with the choice of reference measure $\mathbb{P}$ induced by Markov process in (1), we can define the conditional SDEs conditioned on $\mathcal{H}_{t_k}$ which inducing the desired path measure $\mathbb{P}^\star$. Note that this is an extension of the original Doob's $h$-transform (Doob, 1957), incorporating multiple marginal constraints. Below, we summarize the relevant result.

**Theorem 3.1** (Multi-marginal Doob's $h$-transform). *Let us define a sequence of functions $\{h_i\}_{i \in [1:k]}$, where each $h_i : [t_{i-1}, t_i) \times \mathbb{R}^d \to \mathbb{R}_+$, for all $i \in [1:k]$, is a conditional expectation $h_i(t, \mathbf{x}_t) := \mathbb{E}_{\mathbb{P}}\left[\prod_{j \geq i}^{k} f_j(\mathbf{y}_{t_j}|\mathbf{X}_{t_j})|\mathbf{X}_t = \mathbf{x}_t\right]$, where $\{f_i\}_{i \in [1:k]}$ is defined in (3). Now, we define a function $h : [0, T] \times \mathbb{R}^d \to \mathbb{R}_+$ by integrating the functions $\{h_i\}_{i \in [1:k]}$,*

$$h(t, \mathbf{x}) := \sum_{i=1}^{k} h_i(t, \mathbf{x})\mathbf{1}_{[t_{i-1}, t_i)}(t). \tag{4}$$

---

[1]See details on Appendix B.2.
[2]Proofs and detailed derivations are provided in Appendix B.

*Then, with the initial condition $\mu_0^\star(d\mathbf{x}_0) = h_1(t_0, \mathbf{x}_0)\mu_0(d\mathbf{x}_0)$, the solution of the following conditional SDE inducing the posterior path measure $\mathbb{P}^\star$ in (2):*

$$(\text{Conditioned State}) \quad d\mathbf{X}_t^\star = [b(t, \mathbf{X}_t^\star) + \nabla_\mathbf{x} \log h(t, \mathbf{X}_t^\star)]\, dt + d\mathbf{W}_t \tag{5}$$

Theorem 3.1 demonstrates that we can obtain sample trajectories from $\mathbb{P}^\star$ in (2) by simulating the dynamics in (5). However, estimating the functions $\{h_i\}_{i\in[1:k]}$ requires both the estimation of the sequence of normalization constants $\{\mathbf{L}_i\}_{i\in[1:k]}$ and the computation of conditional expectations, which is infeasible in general. For these reasons, we propose a VI algorithm to approximate the functions $\{h_i\}_{i\in[1:k]}$ and derive the variational bound for the VI by exploiting the theory of SOC.

## 3.2 STOCHASTIC OPTIMAL CONTROL

The SOC (Fleming & Soner, 2006; Carmona, 2016) is a mathematical framework that deals with the problem of finding control policies in order to achieve certain object. In this paper, we define following *control-affine* SDE, adjusting the prior dynamics in (1) with a Markov control $\alpha : [0, T] \times \mathbb{R}^d \to \mathbb{R}^d$:

$$(\text{Controlled State}) \quad d\mathbf{X}_t^\alpha = [b(t, \mathbf{X}_t^\alpha) + \alpha(t, \mathbf{X}_t^\alpha)]\, dt + d\tilde{\mathbf{W}}_t, \tag{6}$$

where $\mathbf{X}_0^\alpha \sim \mu_0$. We refer to the SDE in (6) as *controlled* SDE. We can expect that for a well-defined function set $\alpha \in \mathbb{A}$, the class of controlled SDE (6) encompass the SDE in (5). This implies that the desired path measure $\mathbb{P}^\star$ can be achieved through the SOC formulation. In general, the goal of SOC is to find the *optimal control* policy $\alpha^\star$ that minimizes a given arbitrary cost function $\mathcal{J}(t, \mathbf{x}_t, \alpha)$ *i.e.,* $\alpha^\star(t, \mathbf{x}_t) = \arg\min_{\alpha \in \mathbb{A}} \mathcal{J}(t, \mathbf{x}_t, \alpha)$ and determine the *value function* $\mathcal{V}(t, \mathbf{x}_t) = \min_{\alpha \in \mathbb{A}} \mathcal{J}(t, \mathbf{x}_t, \alpha)$, where $\mathcal{V}(t, \mathbf{x}_t) := \mathcal{J}(t, \mathbf{x}_t, \alpha^\star) \leq \mathcal{J}(t, \mathbf{x}_t, \alpha)$ holds for any $\alpha \in \mathbb{A}$. Below, we demonstrate how, with a carefully chosen cost function, the theory of SOC establishes a connection between two classes of SDEs (5) and (6). This connection enables the development of a variational inference algorithm with a *tight* evidence lower bound (ELBO). To this end, we consider the following cost function:

$$\mathcal{J}(t, \mathbf{x}_t, \alpha) = \mathbb{E}_{\mathbb{P}^\alpha}\left[\int_t^T \frac{1}{2}\|\alpha(s, \mathbf{X}_s^\alpha)\|^2\, ds - \sum_{i:\{t \leq t_i\}} \log f_i(\mathbf{y}_{t_i}|\mathbf{X}_{t_i}^\alpha)|\mathbf{X}_t^\alpha = \mathbf{x}_t\right]. \tag{7}$$

Then, the value function $\mathcal{V}$ for (7) satisfies the dynamic programming principle (Carmona, 2016):

**Theorem 3.2** (Dynamic Programming Principle). *Let us consider a sequence of left continuous functions $\{\mathcal{V}_i\}_{i\in[1:k+1]}$, where each $\mathcal{V}_i \in C^{1,2}([t_{i-1}, t_i) \times \mathbb{R}^d)$*

$$\mathcal{V}_i(t, \mathbf{x}_t) := \min_{\alpha \in \mathbb{A}} \mathbb{E}_{\mathbb{P}^\alpha}\left[\int_{t_{i-1}}^{t_i} \frac{1}{2}\|\alpha_s\|^2\, ds - \log f_i(\mathbf{y}_{t_i}|\mathbf{X}_{t_i}^\alpha) + \mathcal{V}_{i+1}(t_i, \mathbf{X}_{t_i}^\alpha)|\mathbf{X}_t = \mathbf{x}_t\right], \tag{8}$$

*for all $i \in [1 : k]$ and $\mathcal{V}_{k+1} = 0$. Then, for any $0 \leq t \leq u \leq T$, the value function $\mathcal{V}$ for the cost function in (7) satisfying the recursion defined as follows:*

$$\mathcal{V}(t, \mathbf{x}_t) = \min_{\alpha \in \mathbb{A}} \mathbb{E}_{\mathbb{P}^\alpha}\left[\int_t^{t_{I(u)}} \frac{1}{2}\|\alpha_s\|^2\, ds - \sum_{i:\{t \leq t_i \leq t_{I(u)}\}} \log f_i + \mathcal{V}_{I(u)+1}(t_{I(u)}, \mathbf{X}_{t_{I(u)}}^\alpha)|\mathbf{X}_t^\alpha = \mathbf{x}_t\right], \tag{9}$$

*with the indexing function $I(u) = \max\{i \in [1 : k]|t_i \leq u\}$ and $f_i = f_i(\mathbf{y}_{t_i}|\mathbf{X}_{t_i}^\alpha)$*

The value functions presented in Theorem 3.2 suggest that the objective of the optimal control policy $\alpha$ for the interval $[t_{i-1}, t_i)$ is not just to minimize the negative log-potential $-\log f_i(\mathbf{y}_{t_i}|\cdot)$ for the immediate observation $\mathbf{y}_{t_i}$. Instead, it also involves considering future costs $\{\mathcal{V}_j\}_{j\in[i+1:k]}$ and the corresponding future observations $\{\mathbf{y}_{t_j}\}_{j\in[i+1:k]}$, since $\mathcal{V}_i$ follows a recursive structure. Since our goal is to approximate $\mathbb{P}^\star$ in (5), it is natural that the optimal control policy $\alpha^\star$ should reflect the future observations $\{\mathbf{y}_{t_j}\}_{j\in[i+1:k]}$, as the $h$-function inherently does. Next, we will derive the explicit form of the optimal control policy.

**Theorem 3.3** (Verification Theorem). *Suppose there exist a sequence of left continuous functions $\mathcal{V}_i(t, \mathbf{x}) \in C^{1,2}([t_{i-1}, t_i), \mathbb{R}^d)$, for all $i \in [1 : k]$, satisfying the following Hamiltonian-Jacobi-Bellman (HJB) equation:*

$$\partial_t \mathcal{V}_{i,t} + \mathcal{A}_t \mathcal{V}_{i,t} + \min_{\alpha \in \mathbb{A}}\left[(\nabla_\mathbf{x} \mathcal{V}_{i,t})^\top \alpha_{i,t} + \frac{1}{2}\|\alpha_{i,t}\|^2\right] = 0, \quad t_{i-1} \leq t < t_i \tag{10}$$

$$\mathcal{V}_i(t_i, \mathbf{x}) = -\log f_i(\mathbf{y}_{t_i}|\mathbf{x}) + \mathcal{V}_{i+1}(t_i, \mathbf{x}), \quad t = t_i, \quad \forall i \in [1 : k], \tag{11}$$

*where a minimum is attained by* $\alpha_i^\star(t, \mathbf{x}) = \nabla_\mathbf{x} \mathcal{V}_i(t, \mathbf{x})$. *Now, define a function* $\alpha : [0, T] \times \mathbb{R}^d \to \mathbb{R}^d$ *by integrating the optimal controls* $\{\alpha_i\}_{i \in [1:k]}$,

$$\alpha^\star(t, \mathbf{x}) := \sum_{i=1}^k \alpha_i^\star(t, \mathbf{x}) \mathbf{1}_{[t_{i-1}, t_i)}(t) \tag{12}$$

*Then,* $\mathcal{V}(t, \mathbf{x}_t) = \mathcal{J}(t, \mathbf{x}_t, \alpha^\star) \leq \mathcal{J}(t, \mathbf{x}_t, \alpha)$ *holds for any* $(t, \mathbf{x}_t) \in [0, T] \times \mathbb{R}^d$ *and* $\alpha \in \mathbb{A}$.

Note that the optimal control (12) for the cost function (7) share a similar structure as in (4). Since the theory of PDEs establishes a fundamental link between various classes of PDEs and SDEs (Richter & Berner, 2022; Berner et al., 2024), it allows us to reveal the inherent connection between (12) and (4), thereby enable us to simulate the conditional SDE (5) in an alternative way.

**Lemma 3.4** (Hopf-Cole Transformation). *The* $h$ *function satisfying the following linear PDE:*

$$\partial_t h_{i,t} + \mathcal{A}_t h_{i,t} = 0, \quad t_{i-1} \leq t < t_i \tag{13}$$

$$h_i(t_i, \mathbf{x}) = f_i(\mathbf{y}_{t_i}|\mathbf{x}) h_{i+1}(t_i, \mathbf{x}), \quad t = t_i, \quad \forall i \in [1:k]. \tag{14}$$

*Moreover, for a logarithm transformation* $\mathcal{V} = -\log h$, $\mathcal{V}$ *satisfying the HJB equation in (10-11).*

According to Lemma 3.4, the solution of linear PDE in (13-14) is negative exponential to the solution of the HJB equation in (10-11). Therefore, it leads to the following corollary:

**Corollary 3.5** (Optimal Control). *For optimal control* $\alpha^\star$ *induced by the cost function (7) with dynamics (6), it satisfies* $\alpha^\star = \nabla_\mathbf{x} \log h$. *In other words, we can simulate the conditional SDEs in (5) by simulating the controlled SDE (6) with optimal control* $\alpha^\star$.

Corollary 3.5 states that the Markov process induced by $\alpha^\star$ and $\nabla_\mathbf{x} \log h$ is equivalent under same initial condition. However, comparing the $\mathbb{P}^\alpha$ induced by the controlled dynamics in (6) with an initial condition $\mu_0$, the conditioned dynamics (5) has an intial condition $\mu_0^\star$ to inducing the desired path measure $\mathbb{P}^\star$. In other words, although we find the optimal control $\alpha^\star$, the constant discrepancy between $\mu_0$ and $\mu_0^\star$ still remain, thereby keeping $\mathbb{P}^\star$ and $\mathbb{P}^{\alpha^\star}$ misaligned. Fortunately, a surrogate cost function can be derived from the variational representation under the KL-divergence, allowing us to find the optimal control while minimizing the discrepancy.

**Theorem 3.6** (Tight Variational Bound). *Let us assume that the path measure* $\mathbb{P}^\alpha$ *induced by (6) for any* $\alpha \in \mathbb{A}$ *satisfies* $D_{KL}(\mathbb{P}^\alpha | \mathbb{P}^\star) < \infty$. *Then, for a cost function* $\mathcal{J}$ *in (7) and* $\mu_0^\star$ *in (5), it holds:*

$$D_{KL}(\mathbb{P}^\alpha | \mathbb{P}^\star) = D_{KL}(\mu_0 | \mu_0^\star) + \mathbb{E}_{\mathbf{x}_0 \sim \mu_0}[\mathcal{J}(0, \mathbf{x}_0, \alpha)] = \mathcal{L}(\alpha) + \log \mathbf{Z}(\mathcal{H}_{t_k}) \geq 0, \tag{15}$$

*where the objective function* $\mathcal{L}(\alpha)$ *(a negative ELBO) is given by:*

$$\mathcal{L}(\alpha) = \mathbb{E}_{\mathbb{P}^\alpha}\left[\int_0^T \frac{1}{2} \|\alpha(s, \mathbf{X}_s^\alpha)\|^2 ds - \sum_i^k \log g_i(\mathbf{y}_{t_i}|\mathbf{X}_{t_i}^\alpha)\right] \geq -\log \mathbf{Z}(\mathcal{H}_{t_k}). \tag{16}$$

*Moreover, assume that* $\mathcal{L}(\alpha)$ *has a global minimum* $\alpha^\star = \arg\min_{\alpha \in \mathbb{A}} \mathcal{L}(\alpha)$. *Then the equality holds in (16) i.e.,* $\mathcal{L}(\alpha^\star) = -\log \mathbf{Z}(\mathcal{H}_{t_k})$ *and* $\mu_0^\star = \mu_0$ *almost everywhere with respect to* $\mu_0$.

Theorem 3.6 suggests that the optimal control $\alpha^\star = \arg\min_\alpha \mathcal{L}(\alpha)$ provides the tight variational bound for the likelihood functions $\{g_i\}_{i \in [1:k]}$ and the prior path measure $\mathbb{P}$ induced by (1). Furthermore, $\alpha^\star$ ensures that $\mu_0^\star = \mu_0$, indicating that simulating the controlled SDE in (6) with $\alpha^\star$ and initial condition $\mu_0$ generates trajectories from the posterior path measure $\mathbb{P}^\star$ in (2). In practice, the optimal control $\alpha^\star$ can be approximated using a highly flexible neural network, which serves as a function approximator (*i.e.*, $\alpha := \alpha(\cdot; \theta)$), optimized through gradient descent-based optimization (Li et al., 2020; Zhang & Chen, 2022; Vargas et al., 2023).

However, applying gradient descent-based optimization necessitates computing gradients through the simulated diffusion process over the interval $[0, T]$ to estimate the objective function (16) for neural network training. This approach can become slow, unstable, and memory-intensive as the time horizon or dimension of latent space increases (Iakovlev et al., 2023; Park et al., 2024). It contrasts with the philosophy of many recent generative models (Ho et al., 2020; Song et al., 2020), which aim to decompose the generative process and solve the sub-problems jointly. Additionally, for inference, it requires numerical simulations such as Euler-Maruyama solvers (Kloeden & Platen, 2013), which can also be time-consuming for a long time series. It motivated us to propose an efficient and scalable modeling approach for real-world applications described in the next section.

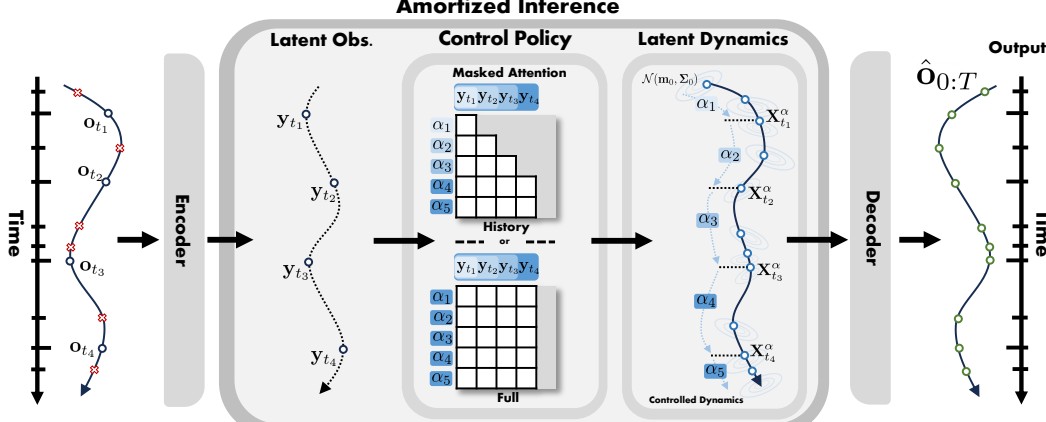

Figure 1: **Conceptual illustration**. Given the observed time stamps $\mathcal{T} = \{t_i\}_{i \in [1:4]}$ and the unseen time stamps $\mathcal{T}_u$ ($\times$ in figure), the encoder maps the input time series $\{\mathbf{o}_t\}_{t \in \mathcal{T}}$ into auxiliary variables $\{\mathbf{y}_t\}_{t \in \mathcal{T}}$. These auxiliary variables are then utilized to compute the control policies $\{\alpha\}_{i \in [1:5]}$ through a masked attention mechanism that relies on two different assimilation schemes. The computed policies $\{\alpha\}_{i \in [1:5]}$ control the prior dynamics $\mathbb{P}$ over the interval $[0, T]$ to approximate the posterior $\mathbb{P}^\star$ in the latent space. Finally, the sample path $\mathbf{X}_{0:T}^\alpha \sim \mathbb{P}^\alpha$ are decoded to generate predictions across the complete time stamps $\mathcal{T}' = \mathcal{T} \cup \mathcal{T}_u$ ($\circ$ in figure), over the entire interval $[0, T]$.

### 3.3 EFFICIENT MODELING OF THE LATENT SYSTEM

The linear approximation of the general drift function provides a *simulation-free* property for the dynamics and significantly enhances scalability while ensuring high-quality performance (Deng et al., 2024). Motivate by this property, we investigate a class of linear SDEs to improve the efficiency of the proposed controlled dynamics, ensuring superior performance compared to other baselines. We introduce the following affine linear SDEs:

$$d\mathbf{X}_t = [-\mathbf{A}\mathbf{X}_t + \alpha]\, dt + d\mathbf{W}_t, \quad \text{where} \quad \mathbf{X}_0 \sim \mathcal{N}(\mathbf{m}_0, \boldsymbol{\Sigma}_0), \tag{17}$$

and a matrix $\mathbf{A} \in \mathbb{R}^{d \times d}$ and a vector $\alpha \in \mathbb{R}^d$. The solutions for $\mathbf{X}_t$ in (17) has a closed-form Gaussian distribution for any $t \in [0, T]$, where the mean $\mathbf{m}_t$ and covariance $\boldsymbol{\Sigma}_t$ can be explicitly computed by solving the ODEs (Särkkä & Solin, 2019):

$$\mathbf{m}_t = e^{-\mathbf{A}t}\mathbf{m}_0 - \mathbf{A}^{-1}(e^{-\mathbf{A}t} - \mathbf{I})\alpha \tag{18}$$

$$\boldsymbol{\Sigma}_t = e^{-\mathbf{A}t}\boldsymbol{\Sigma}_0 e^{-\mathbf{A}^\top t} + \int_0^t e^{-\mathbf{A}(t-s)} e^{-\mathbf{A}^\top (t-s)} ds. \tag{19}$$

However, calculating the moments $\mathbf{m}_t$ and $\boldsymbol{\Sigma}_t$ in (18) involves computing matrix exponentials, inversions, and performing numerical integration. These operations can be computationally intensive, especially for large matrices or when high precision is required. These computations can be simplifies by restricting the matrix $\mathbf{A}$ to be a diagonal or semi-positive definite (SPD).

**Remark 3.7** (Diagonalization). *Since SPD matrix $\mathbf{A}$ admits the eigen-decomposition $\mathbf{A} = \mathbf{E}\mathbf{D}\mathbf{E}^\top$ with $\mathbf{E} \in \mathbb{R}^{d \times d}$ and $\mathbf{D} \in diag(\mathbb{R}^d)$, the process $\mathbf{X}_t$ expressed in a standard basis can be transformed to a $\hat{\mathbf{X}}_t$ which have diagonalized drift function. In the space spanned by the eigen-basis $\mathbf{E}$, the dynamics in (1) can be rewritten into:*

$$d\hat{\mathbf{X}}_t = \left[-\mathbf{D}\hat{\mathbf{X}}_t + \hat{\alpha}\right] dt + d\hat{\mathbf{W}}_t, \quad \text{where} \quad \hat{\mathbf{X}}_0 \sim \mathcal{N}(\hat{\mathbf{m}}_0, \hat{\boldsymbol{\Sigma}}_0), \tag{20}$$

*$\hat{\mathbf{X}}_t = \mathbf{E}^\top \mathbf{X}_t$, $\hat{\alpha} = \mathbf{E}^\top \alpha$, $\hat{\mathbf{W}}_t = \mathbf{E}^\top \mathbf{W}_t$, $\hat{\mathbf{m}}_t = \mathbf{E}^\top \mathbf{m}_t$ and $\hat{\boldsymbol{\Sigma}}_t = \mathbf{E}^\top \boldsymbol{\Sigma}_t \mathbf{E}$. Note that $\hat{\mathbf{W}}_t \stackrel{d}{=} \mathbf{E}^\top \mathbf{W}_t$ for any $t \in [0, T]$ due to the orthonormality of $\mathbf{E}$, so $\hat{\mathbf{W}}_t$ can be regarded as a standard Wiener process. Because of $\mathbf{D} = diag(\boldsymbol{\lambda})$, where $\boldsymbol{\lambda} = \{\lambda_1, \cdots, \lambda_d\}$ and each $\lambda_i \geq 0$ for all $i \in [1:d]$, we can obtain the state distributions of $\mathbf{X}_t$ for any $t \in [0, T]$ by solving ODEs in (18-19) analytically, without the need for numerical computation. The results are then transforming back to the standard basis i.e., $\mathbf{m}_t = \mathbf{E}\hat{\mathbf{m}}_t$, $\boldsymbol{\Sigma}_t = \mathbf{E}\hat{\boldsymbol{\Sigma}}_t \mathbf{E}^\top$.*

**Locally Linear Approximation**   To leverage the advantages of linear SDEs in (17) which offer simulation-free property, we aim to linearize the drift function in (6). However, the naïve formulation described in (17) may limit the expressiveness of the latent dynamics for real-world applications. Hence, we introduce a parameterization strategy inspired by (Becker et al., 2019; Klushyn et al., 2021a), which leverage neural networks to enhance the flexibility by fully incorporating an attentive structure with a given observations $\mathbf{y}_{0:T}$ while maintaining a linear formulation:

$$d\mathbf{X}_t^\alpha = [-\mathbf{A}_t\mathbf{X}_t + \alpha_t]\, dt + d\tilde{\mathbf{W}}_t, \tag{21}$$

where the approximated drift function is constructed as affine formulation with following components:

$$\mathbf{A}_t = \sum_{l=1}^{L} w_\theta^{(l)}(\mathbf{z}_t)\mathbf{A}^{(l)}, \quad \alpha_t = \mathbf{B}_\theta \mathbf{z}_t. \tag{22}$$

The matrix $\mathbf{A}_t$ is given by a convex combination of $L$ trainable base matrices $\{\mathbf{A}^{(l)}\}_{l\in[1:L]}$, where the weights $\mathbf{w}_\theta = \text{softmax}(\mathbf{f}_\theta(\mathbf{z}_t))$ are produced by the neural network $\mathbf{f}_\theta$. Additionally, $\mathbf{B}_\theta \in \mathbb{R}^{d\times d}$ is a trainable matrix. The latent variable $\mathbf{z}_t$ is produced by the transformer $\mathbf{T}_\theta$, which encodes the given observations $\mathbf{y}_{0:T}$, depending on the task-specific information assimilation scheme.

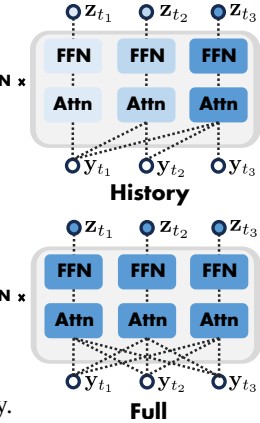

Figure 2: Two types of information assimilation schemes.

Figure 2 illustrates two assimilation schemes using masked attention mechanism[3]: the history assimilation scheme, the transformer $\mathbf{T}_\theta$ encodes information up to the current time $t$ and outputs $\mathbf{z}_t$, i.e., $\mathbf{z}_t = \mathbf{T}_\theta(\mathcal{H}_t)$, and the full assimilation scheme, the transformer $\mathbf{T}_\theta$ encodes information over the entire interval $[0, T]$ and outputs $\mathbf{z}_t$, i.e., $\mathbf{z}_t = \mathbf{T}_\theta(\mathcal{H}_T)$. Note that this general formulation brought from the control formulation enables more flexible use of information encoded from observations. In contrast, previous Kalman-filtering based CD-SSM method (Schirmer et al., 2022) relies on recurrent updates, which limits them to typically using historical information only.

Furthermore, since observations are updated only at discrete time steps $\{t_i\}_{i\in[1:k]}$, the latent variables $\mathbf{z}_t$ remain constant within any interval $t \in [t_{i-1}, t_i)$ for all $i \in [1 : k]$, making $\mathbf{A}_i$ and $\alpha_i$ constant as well. As a result, the dynamics (21) remain linear over local intervals. This structure enables us to derive a closed-form solution for the intermediate latent states.

**Theorem 3.8** (Simulation-free estimation). *Let us consider sequences of SPD matrices $\{\mathbf{A}_i\}_{i\in[1:k]}$ that admit the eigen-decomposition $\mathbf{A}_i = \mathbf{E}\mathbf{D}_i\mathbf{E}^\top$ with $\mathbf{E} \in \mathbb{R}^{d\times d}$ and $\mathbf{D}_i \in diag(\mathbb{R}^d) \succeq 0$ for all $i \in [1 : k]$, control vectors $\{\alpha_i\}_{i\in[1:k]}$ and following control-affine SDEs for all $i \in [1 : k]$:*

$$d\mathbf{X}_t = [-\mathbf{A}_i\mathbf{X}_t + \alpha_i]\, dt + \sigma d\mathbf{W}_t, \quad t \in [t_{i-1}, t_i). \tag{23}$$

*Then, with $\mathbf{X}_0 \sim \mathcal{N}(\mathbf{m}_0, \mathbf{\Sigma}_0)$, the solution of (23) is a Gaussian process $\mathcal{N}(\mathbf{m}_{t_i}, \mathbf{\Sigma}_{t_i})$ with:*

$$\mathbf{m}_{t_i} = \mathbf{E}\left(e^{-\sum_{j=1}^{i}(t_j-t_{j-1})\mathbf{D}_j}\hat{\mathbf{m}}_{t_0} - \sum_{k=1}^{i} e^{-\sum_{j=k}^{i}(t_j-t_{j-1})\mathbf{D}_j}\mathbf{D}_k^{-1}\left(\mathbf{I} - e^{(t_k-t_{k-1})\mathbf{D}_k}\right)\hat{\alpha}_k\right),$$

$$\mathbf{\Sigma}_{t_i} = \mathbf{E}\left(e^{-2\sum_{j=1}^{i}(t_j-t_{j-1})\mathbf{D}_j}\hat{\mathbf{\Sigma}}_{t_0} - \frac{1}{2}\sum_{k=1}^{i} e^{-2\sum_{j=k}^{i}(t_j-t_{j-1})\mathbf{D}_j}\mathbf{D}_k^{-1}\left(\mathbf{I} - e^{2(t_k-t_{k-1})\mathbf{D}_k}\right)\right)\mathbf{E}^\top,$$

*where $\hat{\mathbf{m}}_{t_i} = \mathbf{E}^\top\mathbf{m}_{t_i}, \hat{\mathbf{\Sigma}}_{t_i} = \mathbf{E}^\top\mathbf{\Sigma}_{t_i}\mathbf{E}$ and $\hat{\alpha}_i = \mathbf{E}^\top\alpha_i$ for all $i \in [1 : k]$.*

**Parallel Computation**   Given an associative operator $\otimes$ and a sequence of elements $[s_{t_1}, \cdots s_{t_K}]$, the parallel scan algorithm computes the all-prefix-sum which returns the sequence

$$[s_{t_1}, (s_{t_1} \otimes s_{t_2}), \cdots, (s_{t_1} \otimes s_{t_2} \otimes \cdots \otimes s_{t_K})] \tag{24}$$

in $\mathcal{O}(\log K)$ time. Leveraging the linear formulation described in Theorem 3.8 and the inherent parallel nature of the transformer architecture for sequential structure, our method can be integrated with the parallel scan algorithm (Blelloch, 1990) resulting efficient computation of the marginal Gaussian distributions by computing both moments $\{\mathbf{m}_t\}_{t\in[0,T]}$ and $\{\mathbf{\Sigma}_t\}_{t\in[0,T]}$ in a parallel sense[4].

---

[3]See Appendix D.2 for more details.

[4]See Appendix C for more details.

**Remark 3.9** (Non-Markov Control). *Note that (21-22) involves approximating the Markov control by a non-Markov control $\alpha(\mathcal{H}_t) := \alpha^\theta$, parameterized by neural network $\theta$. However, Theorem 3.3 establishes that the optimal control should be Markov, as it is verified by the HJB equation (Van Handel, 2007). In our case, we expect that with a high-capacity neural network $\theta$, the local minimum $\theta^M$, obtained after $M$ gradient descent steps $\theta^{m+1} = \theta^m - \nabla_\theta \mathcal{L}(\alpha^{\theta^m})$ yields $\mathcal{L}(\alpha^\star) \approx \mathcal{L}(\alpha^{\theta^m \to \theta^M})$.*

**Auxiliary Variable**   Moreover, to enhance flexibility, we treat $\mathbf{y}_{0:T}$ as an auxiliary variable in the latent space which is produced by a neural network encoder $q_\phi$ applied to the given time series data $\mathbf{o}_{0:T}$ *i.e.*, $\mathbf{y}_{0:T} \sim q_\phi(\mathbf{y}_{0:T}|\mathbf{o}_{0:T})$, where it is factorized as

$$q_\phi(\mathbf{y}_{0:T}|\mathbf{o}_{0:T}) = \prod_{i=1}^{k} q_\phi(\mathbf{y}_{t_i}|\mathbf{o}_{t_i}) = \prod_{i=1}^{k} \mathcal{N}(\mathbf{y}_{t_i}|\mathbf{q}_\phi(\mathbf{o}_{t_i}), \mathbf{\Sigma}_q) \tag{25}$$

with a fixed variance $\mathbf{\Sigma}_q$. Additionally, it enables the modeling of nonlinear emission distributions through a neural network decoder $p_\psi(\mathbf{o}_{0:T}|\mathbf{y}_{0:T})$, where it is factorized as

$$p_\psi(\mathbf{o}_{0:T} \mid \mathbf{y}_{0:T}) = \prod_{i=1}^{k} p_\psi(\mathbf{o}_{t_i} \mid \mathbf{y}_{t_i}), \tag{26}$$

with the likelihood function $p_\psi$ depending on the task at hand. This formulation first maps the time series $\mathbf{o}_{0:T}$ into a suitable low-dimensional space $\mathbf{y}_{0:T}$, allowing more efficient modeling of the latent dynamics $\mathbf{X}_{0:T}^\alpha$. The information capturing the underlying physical dynamics resides in a much lower-dimensional space compared to the original sequence (Fraccaro et al., 2017). Therefore, performing generative modeling in this reduced latent space (rather than directly in the high-dimensional domain, e.g., pixel values in an image sequence) offers greater flexibility in parameterization.

## 3.4   TRAINING AND INFERENCE

**Training Objective Function**   We jointly train, in an end-to-end manner, the amortization parameters $\{\phi, \psi\}$ for the encoder-decoder pair, along with the parameters of the latent dynamics $\theta = \{\mathbf{f}_\theta, \mathbf{B}_\theta, \mathbf{T}_\theta, \mathbf{m}_0, \mathbf{\Sigma}_0, \{\mathbf{A}^{(l)}\}_{l \in [1:L]}\}$, which include the parameters required for controlled latent dynamics. The training is achieved by maximizing the evidence lower bound (ELBO) of the observation log-likelihood for a given the time series $\mathbf{o}_{0:T}$:

$$\log p_\psi(\mathbf{o}_{0:T}) \geq \mathbb{E}_{\mathcal{H}_T \sim q_\phi(\mathbf{y}_{0:T}|\mathbf{o}_{0:T})} \left[ \log \frac{\prod_{i=1}^{K} p_\psi(\mathbf{o}_{t_i}|\mathbf{y}_{t_i}) g(\mathbf{y}_{0:T})}{\prod_{i=1}^{K} q_\phi(\mathbf{y}_{t_i}|\mathbf{o}_{t_i})} \right] \tag{27}$$

$$\geq \mathbb{E}_{\mathcal{H}_T \sim q_\phi(\mathbf{y}_{0:T}|\mathbf{o}_{0:T})} \left[ \sum_{i=1}^{K} \log p_\psi(\mathbf{o}_{t_i}|\mathbf{y}_{t_i}) - \mathcal{L}(\theta) \right] = \text{ELBO}(\psi, \phi, \theta) \tag{28}$$

Since $\mathbf{Z}(\mathcal{H}_{t_k}) = g(\mathbf{y}_{0:T})$, the prior over auxiliary variable $g(\mathbf{y}_{0:T})$ can be computed using the ELBO $\mathcal{L}(\theta)$ proposed in (16) as part of our variational inference procedure for the latent posterior $\mathbb{P}^\star$ in proposed Sec 2, including all latent parameters $\theta$ containing the control $\alpha^\theta$. Note that our modeling is computationally favorable for both training and inference, since estimating marginal distributions in a latent space can be parallelized. This allows for the generation of a latent trajectory over the entire interval without the need for numerical simulations. The overall training and inference processes are summarized in the Algorithm 3 and Algorithm 4 in Appendix, respectively.

## 4   EXPERIMENT

In this section, we present empirical results demonstrating the effectiveness of ACSSM in modeling real-world irregular time-series data. The primary objective was to evaluate its capability to capture the underlying dynamics across various datasets. To demonstrate the applicability of the ACSSM, we conducted experiments on four tasks: per-time regression/classification and sequence interpolation/extrapolation, using four datasets: Human Activity, USHCN (Menne et al., 2015), and Physionet (Silva et al., 2012). We compare our approach against various baselines including RNN architecture (RKN-$\Delta_t$ (Becker et al., 2019), GRU-$\Delta_t$ (Chung et al., 2014), GRU-D (Che

et al., 2018)) as well as dynamics-based models (Latent-ODE (Chen et al., 2018; Rubanova et al., 2019), ODE-RNN (Rubanova et al., 2019), GRU-ODE-B (De Brouwer et al., 2019), CRU (Schirmer et al., 2022), Latent-SDE$_H$ (Zeng et al., 2023)) and attention-based models (mTAND (Shukla & Marlin, 2021)), which have been developed for modeling irregular time series data. We reported the averaged results over five runs with different seed. The best results are highlighted in **bold**, while the second-best results are shown in blue. Additional experimental details can be found in Appendix D.

## 4.1 PER TIME POINT CLASSIFICATION & REGRESSION

**Human Activity Classification**  We investigated the classification performance of our proposed model. For this purpose, we trained the model on the Human Activity dataset, which contains time-series data from five individuals performing various activities such as walking, sitting, lying, and standing etc. The dataset includes 12 features in total, representing 3D positions captured by four sensors attached to the belt, chest, and both ankles. Following the pre-processing approach proposed by (Rubanova et al., 2019), the dataset comprises 6,554 sequences, each with 211 time points. The task is to classify each time point into one of seven activities.

Table 1: Test Accuracy (%).

| Model | Acc |
|---|---|
| Latent-ODE[†] | $87.0 \pm 2.8$ |
| Latent-SDE$_H$[‡] | $90.6 \pm 0.4$ |
| mTAND[†] | $91.1 \pm 0.2$ |
| **ACSSM** (Ours) | $\mathbf{91.4 \pm 0.4}$ |

[†] result from (Shukla & Marlin, 2021).
[‡] result from (Zeng et al., 2023).

Table 1 reports the test accuracy, showing that ACSSM outperforms all baseline models. We employed the full assimilation scheme to maintain consistency with the other baselines, which infer the latent state using full observation. It is important to note that the two dynamical models, Latent-ODE and Latent-SDE$_H$, incorporate a parameterized vector field in their latent dynamics, thereby rely on numerical solvers to infer intermediate states. Besides, mTAND is an attention-based method that does not depend on dynamical or state-space models, thus avoiding numerical simulation. We believe the significant performance improvement of our model comes from its integration of an attention mechanism into dynamical models. Simulation-free dynamics avoid numerical simulations while maintaining temporal structure, leading to more stable learning.

**Pendulum Regression**  Next, we explored the problem of sequence regression using pendulum experiment (Becker et al., 2019), where the goal is to infer the sine and cosine of the pendulum angle from irregularly observed, noisy pendulum images (Schirmer et al., 2022). To assess our performance, we compared it with previous dynamics-based models, reporting the regression MSE on a held-out test set as shown in Table 2. We employed the full assimilation scheme. The experimental results demonstrated that our proposed method outperformed existing models, delivering superior performance. These findings highlight that, even when linearizing the drift function, the amortization and proposed neural network based locally linear dynamics in Sec 3.3 preserves the expressivity of our approach, enabling more accurate inference of non-linear systems.

Table 2: Test MSE ($\times 10^{-3}$).

| Model | MSE |
|---|---|
| Latent-ODE[†] | $15.70 \pm 0.29$ |
| CRU[†] | $4.63 \pm 1.07$ |
| Latent-SDE$_H$[‡] | $3.84 \pm 0.35$ |
| S5[⋆] | $3.41 \pm 0.27$ |
| mTAND[‡] | $3.20 \pm 0.60$ |
| **ACSSM** (Ours) | $\mathbf{2.98 \pm 0.30}$ |

[†] result from (Schirmer et al., 2022).
[‡] result from (Zeng et al., 2023).
[⋆] result from (Smith et al., 2023).

Moreover, particularly in comparison to CRU, we believe that the significant performance improvement stems from the fundamental differences in how information is leveraged. To infer intermediate angular values, utilizing not only past information but also future positions of the pendulum can enhance the accuracy of these predictions. In this regard, while CRU relies solely on the past positions of the pendulum for its predictions, our model is capable of utilizing both past and future positions.

## 4.2 SEQUENCE INTERPOLATION & EXTRAPOLATION

**Datasets**  We benchmark the models on two real-world datasets, USHCN and Physionet. The USHCN dataset (Menne et al., 2015) contains 1,218 daily measurements from weather stations across the US with five variables over a four-year period. The Physionet dataset (Silva et al., 2012) contains 8,000 multivariate clinical time-series of 41 features recorded over 48 hours.[5]

**Interpolation**  We begin by evaluating the effectiveness of ACSSM on the interpolation task. Following the approach of Schirmer et al. (2022) and Rubanova et al. (2019), each model is required

---

[5]See Appendix D.1 for details on the datasets.

Table 3: Test MSE ($\times 10^{-2}$) for inter/extra-polation on USHCN and Physionet.

| Model | Interpolation | | Extrapolation | | Runtime (sec./epoch) | |
|---|---|---|---|---|---|---|
| | USHCN | Physionet | USHCN | Physionet | USHCN | Physionet |
| mTAND[†] | $1.766 \pm 0.009$ | $0.208 \pm 0.025$ | $2.360 \pm 0.038$ | $\mathbf{0.340 \pm 0.020}$ | 7 | 10 |
| RKN-$\Delta_t^†$ | $0.009 \pm 0.002$ | $0.186 \pm 0.030$ | $1.491 \pm 0.272$ | $0.703 \pm 0.050$ | 94 | 39 |
| GRU-$\Delta_t^†$ | $0.090 \pm 0.059$ | $0.271 \pm 0.057$ | $2.081 \pm 0.054$ | $0.870 \pm 0.077$ | 3 | 5 |
| GRU-D[†] | $0.944 \pm 0.011$ | $0.338 \pm 0.027$ | $1.718 \pm 0.015$ | $0.873 \pm 0.071$ | 292 | 5736 |
| Latent-ODE[†] | $1.798 \pm 0.009$ | $0.212 \pm 0.027$ | $2.034 \pm 0.005$ | $0.725 \pm 0.072$ | 110 | 791 |
| ODE-RNN[†] | $0.831 \pm 0.008$ | $0.236 \pm 0.009$ | $1.955 \pm 0.466$ | $0.467 \pm 0.006$ | 81 | 299 |
| GRU-ODE-B[†] | $0.841 \pm 0.142$ | $0.521 \pm 0.038$ | $5.437 \pm 1.020$ | $0.798 \pm 0.071$ | 389 | 90 |
| CRU[†] | $0.016 \pm 0.006$ | $0.182 \pm 0.091$ | $1.273 \pm 0.066$ | $0.629 \pm 0.093$ | 122 (57.8)[*] | 114 (63.5)[*] |
| **ACSSM** (Ours) | $\mathbf{0.006 \pm 0.001}$ | $\mathbf{0.116 \pm 0.011}$ | $\mathbf{0.941 \pm 0.014}$ | $0.627 \pm 0.019$ | 2.3 | 3.8 |

† result from (Schirmer et al., 2022).

to infer all time points $t \in \mathcal{T}'$ based on a subset of observations $\mathbf{o}_{t \in \mathcal{T}}$ where $\mathcal{T} \subseteq \mathcal{T}'$. For the interpolation task, the encoded observations $\mathbf{y}_{t \in \mathcal{T}}$ were assimilated by using the full assimilation scheme for the construction of the accurate smoothing distribution. The interpolation results presented in Table 3, where we report the test MSE evaluated on the entire time points $\mathcal{T}'$. For all datasets, ACSSM outperforms other baselines in terms of test MSE. It clearly indicate the expressiveness of the ACSSM, trajectories $\mathbf{X}_{t \in \mathcal{T}'}^\alpha$ sampled from approximated path measure over the entire interval $\mathcal{T}'$ are contain sufficient information for generating accurate predictions.

**Extrapolation** We evaluated ACSSM's performance on the extrapolation task following the experimental setup of Schirmer et al. (2022). Each model infer values for all time stamps $t \in \mathcal{T}'$, where $\mathcal{T}'$ denotes the union of observed time stamps $\mathcal{T} = \{t_i\}_{i \in [1:k]}$ and unseen time stamps $\mathcal{T}_u = \{t_i\}_{i \in [k+1:N]}$, i.e., $\mathcal{T}' = \mathcal{T} \cup \mathcal{T}_u$. For the Physionet dataset, input time stamps $\mathcal{T}$ covered the first 24 hours, while target time stamps $\mathcal{T}'$ spanned the rest hours. In the USHCN dataset, the timeline was split evenly, with $t_k = \frac{N}{2}$. We report the test MSE for unseen time stamps $\mathcal{T}_u = \mathcal{T}' - \mathcal{T}$ based on the observations on time stamps $\mathcal{T}$. For modeling an accurate filtering distribution, we employed the history assimilation scheme for this task. As illustrated in Table 3, ACSSM consistently outperformed all baseline models in terms of MSE on the USHCN dataset, achieving a significant performance gain over the second-best model. For the Physionet dataset, ACSSM exhibited comparable performance.

**Computational Efficiency** To evaluate the training costs in comparison to dynamics-based models that depend on numerical simulations, we re-ran the CRU model on the same hardware used for training our model, indicated by $*$ in Table 3. Specifically, we utilized a single NVIDIA RTX A6000 GPU. As illustrated in Table 3, ACSSM significantly lowers training costs compared to dynamics-based models. Notable, ACSSM demonstrated a runtime that was more than $16-25\times$ faster than CRU, even though both models aim to approximate $\mathbb{P}^\star$ as well. It highlight that the latent modelling approach discussed in Sec 3.3 improves efficiency while achieving an accurate approximation of $\mathbb{P}^\star$.

## 5 Conclusion and limitation

In this work, we proposed the method for modeling time series with irregular and discrete observations, which we called ACSSM. By using a multi-marginal Doob's $h$-transform and a variational inference algorithm by exploiting the theory of SOC, ACSSM efficiently simulates conditioned dynamics. It leverages amortized inference, a simulation-free latent dynamics framework, and a transformer-based data assimilation scheme for scalable and parallel inference. Empirical results show that ACSSM outperforms in various tasks such as classification, regression, and extrapolation while maintaining computational efficiency across real-world datasets.

Although we present the theoretical basis of our method, a thorough analysis in followings remain an open challenge. The variational gap due to the linear approximation may lead to cumulative errors over time, which requires further examination specially for a long time-series such as LLM. Unlike SMC variants (Heng et al., 2020; Lu & Wang, 2024) that use particle-based importance weighting to mitigate approximation errors, ACSSM depends on high-capacity neural networks to accurately approximate the optimal control. Incorporating multiple controls, akin to a multi-agent dynamics approach (Han & Hu, 2020), may alleviate these challenges by enhancing flexibility and robustness.

## Reproducibility Statement

On the theoretical part, all proofs and assumptions are left to Appendix B due the space constraint. The training and inference algorithms are detailed in Algorithm 3 and Algorithm 4, respectively. Additional implementation details, such as data preprocessing are included in Appendix D. We believe these details are sufficient for interested readers to reproduce the results.

## Ethics Statement

In this work, we proposed a method for modeling irregular time series for practical applications, suggesting that ACSSM does not directly influence ethical or societal issues in a positive or negative way. However, because ACSSM can be applied to health-care datasets, we believe it has the potential to benefit society by improving health and well-being of people.

## Acknowledgments

This work was partly supported by Institute of Information & communications Technology Planning & Evaluation(IITP) grant funded by the Korea government(MSIT) (No.RS-2019-II190075, Artificial Intelligence Graduate School Program(KAIST), No.RS-2022-II220713, Meta-learning Applicable to Real-world Problems, No.RS-2024-00509279, Global AI Frontier Lab).

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

## A    BRIEF REVIEWS ON THE KEY CONCEPTS AND RELATED WORKS

In this section, we provide a brief overview of key concepts to help clarify the foundation of our proposed method. Additionally, we review related works to offer a deeper understanding.

**Probabilistic SSMs**    Bayesian filtering and smoothing (Särkkä, 2013) serve as fundamental state estimation techniques for probabilistic SSMs. Formally, SSMs can be defined as follows:

$$\text{(Latent transition) } \mathbf{X}_{t_i} \sim \mathbf{p}_i(\mathbf{x}_{t_{i-1}}, d\mathbf{x}_{t_i}), \mathbf{X}_0 \sim \mathbf{p}_0(\mathbf{X}_0) \quad \text{(Observation) } \mathbf{y}_{t_i} \sim g_{t_i}(\mathbf{y}_{t_i}|\mathbf{X}_{t_i}). \tag{29}$$

The SSMs consist of the $\mathbb{R}^d$ valued latent variable $\{\mathbf{X}_t\}_{t \geq 0}$, which are assumed to follow a time-inhomogeneous Markov process with Markov transition densities $\{\mathbf{p}_i\}_{i \in [1:k]}$, and the observations $\{\mathbf{y}_{t_i}\}_{i \in [1:k]}$ are assumed to be generated from an observation (emission) density $g_i(\cdot|\mathbf{X}_{t_i})$. Then, the goal is to obtain the *filtering/smoothing* distribution, for given observations $\{\mathbf{y}_{t_i}\}_{i \in [1:k]}$, given by:

$$\mathbf{p}(\mathbf{X}_{0:T}|\mathcal{H}_{t_k}) = \frac{1}{\mathbf{Z}(\mathcal{H}_{t_k})}\mathbf{p}(\mathcal{H}_{t_k}|\mathbf{X}_{0:T})\mathbf{p}(\mathbf{X}_{0:T}), \tag{30}$$

where $\mathbf{p}(\mathbf{X}_{0:T})$ is the prior distribution and $\mathbf{Z}(\mathcal{H}_{t_k})$ is a normalization constant, defined as:

$$\mathbf{Z}(\mathcal{H}_{t_k}) = \int \mathbf{p}(\mathcal{H}_{t_k}|\mathbf{X}_{0:T})\mathbf{p}_0(\mathbf{X}_0)\prod_{i=1}^k \mathbf{p}_i(\mathbf{X}_{i-1}, \mathbf{X}_i)d\mathbf{X}_{0:T}. \tag{31}$$

To obtain the *filtering/smoothing* distribution in (30), it typically relies on recursive Bayesian updates, which scale proportionally with the number of observations, resulting in a computational complexity of $\mathcal{O}(k)$ in this context. Previous works (Doerr et al., 2018; Becker et al., 2019; Klushyn et al., 2021b) have proposed RNN-based approximate Bayesian inference methods. However, these models generally assume evenly spaced observations, making it challenging to accurately model irregular time series. On the other hand, deterministic linear SSMs such as S4 (Gu et al., 2021), S5 (Smith et al., 2023), and Mamba (Gu & Dao, 2023) have been introduced, demonstrating improved inference efficiency through parallel computing algorithms.

In contrast, we propose a *probabilistic* SSM that enables efficient and powerful modeling of latent systems. Our approach supports parallel computation, leading to significant gains in both training and inference efficiency. Specifically, we incorporate a parallel scan algorithm (Blelloch, 1990) into probabilistic inference, effectively reducing the computational complexity from $\mathcal{O}(k)$ to $\mathcal{O}(\log k)$.

**Twist function for Conditioned SSMs**    To sequentially sample from the smoothing distribution (30), *i.e.*, $\mathbf{p}(\mathbf{X}_t|\mathcal{H}_{t_k})$ over all $t \in [0, T]$ (referred to here as the *conditioned sampling problem*), it is necessary to compute the marginal distribution $\mathbf{p}(\mathbf{X}_t|\mathcal{H}_{t_k})$. However, this involves an expectation over the marginalized distribution $\mathbf{p}(\mathbf{X}_t|\mathcal{H}_{t_k}) = \int \mathbf{p}(\mathbf{X}_{0:T}|\mathcal{H}_{t_k})d\mathbf{X}_{t:T}$, which is generally intractable. Fortunately, the distribution $\mathbf{p}(\mathbf{X}_t|\mathcal{H}_{t_k})$ can be factorized as (Chopin et al., 2020):

$$\mathbf{p}(\mathbf{X}_t|\mathcal{H}_{t_k}) \propto \mathbf{p}(\mathbf{X}_t|\mathcal{H}_t)\int \mathbf{p}(\mathcal{H}_{t:t_k}|\mathbf{X}_{t:T})d\mathbf{X}_{t:T} \tag{32}$$

$$= \mathbf{p}(\mathcal{H}_t|\mathbf{X}_t)\mathbf{p}(\mathbf{X}_t)\int \mathbf{p}(\mathcal{H}_{t:t_k}|\mathbf{X}_{t:T})d\mathbf{X}_{t:T}. \tag{33}$$

Thus, by approximating the term $\psi(\mathbf{X}_t) := \int \mathbf{p}(\mathcal{H}_{t:t_k}|\mathbf{X}_{t:T})d\mathbf{X}_{t:T}$, the *conditioned sampling problem* can be effectively addressed. Here, the intractable term $\psi$, often referred to as a *twist function*, has been the focus of various approximation algorithms proposed in the Sequential Monte Carlo (SMC) literature (Guarniero et al., 2017; Heng et al., 2020; Lawson et al., 2022; Lu & Wang, 2024). Recently, these methods have also been adapted to large language models (LLMs) (Zhao et al., 2024), further demonstrating their versatility in solving controlled language generation problems.

It is worth noting that our $h$-function defined in (4) serves as an instance of such a twist function within the Feynman-Kac representation. Essentially, it enables us to establish a connection between twisted SSMs and the multi-marginal Doob's $h$-transform.

**Feynman-Kac Models** The Feynman-Kac model provides substantial expressive advantages for analyzing SSMs (Chopin et al., 2020). By offering a flexible measure-theoretic framework, it allows for efficient representation of conditioned SSMs in (30) through the corresponding Feynman-Kac formulae (Del Moral, 2011). Notably, the posterior distribution in (30) can be expressed via a Feynman-Kac model, as described in (2). In the machine learning literature, the Feynman-Kac model has been utilized for abstracting processes such as LLM fine-tuning (Lew et al., 2023) and diffusion based sampler (Phillips et al., 2024). Furthermore, we extend the Feynman-Kac model to continuous settings for time-series modeling. For a more in-depth understanding, refer to (Chopin et al., 2020).

**Continuous Dynamical Models** To accurately model irregular time-series data, neural differential equation families have been proposed as an effective paradigm. This is because the latent, continuous dynamics underlying physical time-series can be well-approximated using data-driven methods with parameterized vector fields. Specifically, Neural ODE (Chen et al., 2018) introduced neural network parameterized vector fields for continuous-time dynamical modeling of time-series data. Building on this, Latent-ODE (Rubanova et al., 2019) proposed latent dynamics by encoding the entire dataset into an initial state, ODE-RNN as a continuous encoder alternative to standard RNNs. GRU-ODE-B (De Brouwer et al., 2019) incorporated Bayesian principle into Neural ODEs to enable online updates for new observations. On the stochastic dynamics side, Latent-SDE (Li et al., 2020) introduced a variational bound for posterior inference in SDEs, while Latent-SDE$_H$ (Zeng et al., 2023) focused on stochastic dynamics evolving within a homogeneous latent space. These neural differential equations generally rely on numerical simulations with dynamic solvers, which can result in significant computation times for both training and inference.

To model continuous time-series using probabilistic SSMs, CD-SSM (Jazwinski, 2007) extends the discrete transitions of latent variables to stochastic transitions governed by SDEs. This approach has inspired neural network-based CD-SSM methods (Schirmer et al., 2022; Ansari et al., 2023) for handling irregularly sampled real-world time-series datasets. Compared to our approach, while Schirmer et al. (2022); Ansari et al. (2023) also leverage locally linear dynamics to improve scalability, they still require numerical approximations to infer Gaussian moments, which limits their ability to fully utilize parallel computation. In contrast, our method successfully leverages parallel computation, as demonstrated by Theorem 3.8, significantly reducing both training and inference costs in conditioned state-space modeling. Moreover, our SOC formulation with approximated control $\alpha$ offers flexibility in sequential modeling, allowing for various information assimilation schemes utilizing powerful neural network architectures, such as transformers.

**Doob's $h$-transform for Conditioned SDEs** In contrast to conditioned SSMs using a *twist function* for discrete transitions, we propose a conditioned SDEs for continuous transitions to model irregular time-series data. This involves extending the traditional Doob's $h$-transform to multi-marginal settings. Specifically, the Doob's $h$-transform (Doob, 1957; Rogers & Williams, 2000; Chetrite & Touchette, 2015) is a technique in probability theory that modifies the behavior of a Markov process, effectively conditioning it to reach a desired state or outcome. It can be understood as reweighting the paths of a stochastic process to make certain events more probable.

In the machine learning context, the Doob's $h$-transform has been applied to tasks such as diffusion models (Ye et al., 2022; Liu et al., 2023; Peluchetti, 2023; Shi et al., 2024; Denker et al., 2024; Park et al., 2024), simulating diffusion bridges (Heng et al., 2021; Baker et al., 2024), posterior approximations (Park et al., 2024), online filtering (Chopin et al., 2023). However, prior works typically focus on single-marginal cases, where for a finite time horizon $[0, T]$, the conditioning is solely on a terminal constraint $\mathbb{P}(\mathbf{X}_T \in A)$ for some set $A \in \mathcal{B}(\mathcal{X})$.

One of our contribution is extends this concept to multi-marginal cases, capturing the continuous dynamics of the posterior distribution conditioned on a collection of observations. Specifically, in Theorem 3.1,we define (multi-marginal) conditional SDEs where the constraint is given by a set of marginals, $\mathbb{P}(\mathbf{X}_{t_1} \in A_1, \dots, \mathbf{X}_{t_k} \in A_k)$ for any $A_i \in \mathcal{B}(\mathbb{R}^d)$ and $i \in [1 : k]$. Since inferring the corresponding $h$-function in these multi-marginal settings is more complex, we reformulate this challenge as a SOC problem. This reformulation also requires extensions of existing theoretical results, such as Theorems 3.2 and 3.3. Additionally, we establish a tight variational bound in Theorem 3.6 demonstrating that our proposed multi-marginal Doob's $h$-transform can be efficiently approximated using the proposed SOC objective.

## B  PROOFS AND DERIVATIONS

In this section, we present the proofs and derivations for all relevant theorems, lemmas, and corollaries We first restate the core concepts of stochastic calculus, which will be used without further explanation.

**Assumptions.** Throughout the paper, we work with a probability space $(\Omega, \mathcal{F}, \{\mathcal{F}_t\}_{t \in [0,T]}, \mathbb{P})$, where the filtration $\{\mathcal{F}_t\}_{t \in [0,T]}$ supports an $\mathbb{R}^d$-valued $\mathcal{F}_t$-adapted Wiener process $\mathbf{W}_t$ for all $t \in [0,T]$. It is important to note that the $\mathbb{P}$-null set is included in $\mathcal{F}_0$, indicating that any event with probability zero at time 0 is measurable in the initial $\sigma$-algebra.

We assume that $b$ and $\alpha$ satisfy following conditions:

- (Lipschitz condition): For any $t \in [0,T]$, $w \in \Omega$, and $\mathbf{x}, \mathbf{x}' \in \mathbb{R}^d$, where $c_0 > 0$ is a Lipschitz constant, the functions satisfy the inequality $|b(t, w, \mathbf{x}) - b(t, w, \mathbf{x}')| \le c_0 |\mathbf{x} - \mathbf{x}'|$.

- (Linear growth condition): For every $\mathbf{x} \in \mathbb{R}^d$, the $\mathcal{F}_t$-progressively measurable processes $b(t, \mathbf{x})_{t \in [0,T]}$ satisfy $\mathbb{E}\left[\int_0^T |b_s|^2 ds\right] < \infty$ and $|b(t, \mathbf{x})| \le c_1(1 + |\mathbf{x}|)$ for $t \in [0,T]$ and $c_1 > 0$.

- (Control function): For any $t \in [0,T]$, $w \in \Omega$, $\mathbf{x} \in \mathbb{R}^d$, and $\theta, \theta' \in \Theta$, the control function $\alpha$ is $L$-Lipschitz function, $|\alpha(t, \mathbf{x}, \theta) - \alpha(t, \mathbf{x}, \theta')| \le L|\theta - \theta'|$. Moreover it satisfy $\mathbb{E}\left[\int_0^T |\alpha_s^2| ds\right] < \infty$.

**Definition B.1** (Infinitesimal Generator). *Let us consider an Itô diffusion process of the form:*

$$d\mathbf{X}_t = b(t, \mathbf{X}_t)dt + \sigma(t)^\top d\mathbf{W}_t, \tag{34}$$

*Then, an infinitesimal generator of the above diffusion process is given by:*

$$\mathcal{A}_t f = \lim_{t \downarrow 0^+} \frac{\mathbb{E}[f(\mathbf{X}_t)] - f(\mathbf{x})}{t} = \nabla_{\mathbf{x}} f^\top b + \frac{1}{2} Trace\left[\sigma\sigma^\top \nabla_{\mathbf{xx}} f\right]. \tag{35}$$

**Definition B.2** (Path Measure). *Let us consider a path-sequence of random variables $\{\mathbf{X}_{t_i}\}_{i \in [1:N]}$ over an interval $0 = t_i \le \cdots \le t_N = T$, where each $\mathbf{X}_{t_i}$ taking values in measurable space $(\mathbb{R}^d, \mathcal{B}(\mathbb{R}^d))$ with a Borel $\sigma$-algebra $\mathcal{B}(\mathbb{R}^d)$. Then, the set of random variables $\{\mathbf{X}_{t_i}\}_{i \in [1:N]}$ is given by the probability measure $\mathbb{P} \in C([0,T], \mathbb{R}^d)$:*

$$\mathbb{P}(\mathbf{X}_{t_0} \in d\mathbf{x}_{t_0}, \cdots, \mathbf{X}_{t_N} \in d\mathbf{x}_{t_N}) = \mathbb{P}(d\mathbf{x}_0) \prod_{i=1}^N \mathbf{P}_i(\mathbf{x}_{t_{i-1}}, d\mathbf{x}_{t_i}), \tag{36}$$

*where $\{\mathbf{P}_i\}_{i=0}^N$ is a sequence of probability kernels from $(\mathbb{R}^d, \mathcal{B}(\mathbb{R}^d))$ to $(\mathbb{R}^d, \mathcal{B}(\mathbb{R}^d))$, for any event $A \in \mathcal{B}(\mathbb{R}^d)$, $\mathbf{P}_i(\mathbf{x}_{t_{i-1}}, A) = \int_A \mathbf{p}_i(\mathbf{x}_{t_{i-1}}, \mathbf{x}_{t_i})d\mathbf{x}_{t_i}$, where $\mathbf{p}_i(\mathbf{x}_{t_{i-1}}, \mathbf{x}_{t_i}) := \mathbf{p}(t_i, \mathbf{x}_{t_i}|t_{i-1}, \mathbf{x}_{t_{i-1}})$ is a transition density obtained by a solution of the Fokker-Placnk equation (Risken & Frank, 2012):*

$$\partial_t \mathbf{p}_t(\mathbf{x}_t) = \mathcal{A}_t^\star \mathbf{p}_t = -\nabla_{\mathbf{x}} \cdot (b\mathbf{p}_t) + \frac{1}{2} Trace\left[\sigma\sigma^\top \nabla_{\mathbf{xx}} \mathbf{p}_t\right], \tag{37}$$

*where $\mathcal{A}^\star$ is an adjoint operator of the generator in (35) and $\mathbf{p}_t$ is the Radon-Nikodym density of $\mu_t$ with respect to the Lebesgue measure. By taking $N \to \infty$, we get the path measure $\mathbb{P}(\mathbf{X}_{0:T} \in d\mathbf{x}_{0:T})$ which describes the weak solutions of the SDE of the form in equation (34).*

**Lemma B.3** (Itô's formula). *Let $v(t, x)$ be $C^1$ in $t$ and $C^2$ in $x$ and let $\mathbf{X}_t$ be the Itô diffusion process of the form in equation (34). Then, the stochastic process $v(t, \mathbf{X}_t)$ is also an Itô diffusion process satisfying:*

$$dv(t, \mathbf{X}_t) = [\partial_t v(t, \mathbf{X}_t) + \mathcal{A}_t v(t, \mathbf{X}_t)] dt + \nabla_{\mathbf{x}} v(t, \mathbf{X}_t)^\top \sigma(t) d\mathbf{W}_t. \tag{38}$$

**Theorem B.4** (Girsanov Theorem). *Consider the two Itô diffusion processes of form*

$$d\mathbf{X}_t = b(t, \mathbf{X}_t)dt + \sigma(t, \mathbf{X}_t)^\top d\mathbf{W}_t, \quad t \in [0,T], \tag{39}$$

$$d\mathbf{Y}_t = \tilde{b}(t, \mathbf{Y}_t)dt + \sigma(t, \mathbf{Y}_t)^\top d\mathbf{W}_t, \quad t \in [0,T], \tag{40}$$

*where both drift functions $b, \tilde{b}$ and the diffusion function $\sigma$ assumed to be invertible are adapted to $\mathcal{F}_t$ and $\mathbf{W}_{[0,T]}$ is $\mathbb{P}$-Wiener process. Moreover, consider $\mathbb{P}$ as the path measures induced by*

*(39). Let us define $H_t := \sigma^{-1}(\tilde{b} - b)$ which is assumed to be satisfying the Novikov's condition (i.e., $\mathbb{E}_{\mathbb{P}}\left[\exp\left(\frac{1}{2}\int_0^T \|H_s\|^2 ds\right)\right] < \infty$), and the $\mathbb{P}$-martingale process*

$$\mathbf{M}_t := \exp\left(\int_0^1 H_s^\top d\mathbf{W}_s - \frac{1}{2}\int_0^t \|H_s\|^2 ds\right) \tag{41}$$

*satisfies $\mathbb{E}_{\mathbb{P}}[\mathbf{M}_T] = 1$. Then for the path measure $\mathbb{Q}$ given as $d\mathbb{Q} = \mathbf{M}_T d\mathbb{P}$, the process $\tilde{\mathbf{W}}_t = \mathbf{W}_t - \int_0^t H_s ds$ is a $\mathbb{Q}$-Wiener process and $\mathbf{Y}_t$ can be represented as*

$$d\mathbf{Y}_t = b(t, \mathbf{Y}_t)dt + \sigma(t, \mathbf{Y}_t)^\top d\tilde{\mathbf{W}}_t, \quad t \in [0, T]. \tag{42}$$

*Therefore $\mathbb{Q}$-law of the process $\mathbf{Y}_t$ is same as $\mathbb{P}$-law of the process $\mathbf{X}_t$.*

### B.1 PROOF OF THEOREM 3.1

We start the section by showing the normalizing property of $\{f_i\}_{i\in[1:k]}$ in (3). By definition, it satisfied that

$$\prod_{i=1}^k \mathbf{L}_i(g_i) = \prod_{i=1}^k \int_{\mathbb{R}^d} g_i(\mathbf{y}_{t_i}|\mathbf{x}_{t_i})d\mathbb{P}(\mathbf{x}_{0:T}) \overset{(i)}{=} \mathbb{E}_{\mathbb{P}}\left[\prod_{i=1}^k g_i(\mathbf{y}_{t_i}|\mathbf{x}_{t_i})\right] = \mathbf{Z}(\mathcal{H}_{t_k}), \tag{43}$$

where $(i)$ follows from the conditional indenpendency of $\mathbf{y}_{t_i}$ given $\mathbf{x}_{t_i}$ for all $i \in [1:k]$. Hence, we get the normalizing property:

$$\mathbb{E}_{\mathbb{P}}\left[\prod_{i=1}^k f_i(\mathbf{x}_{t_i})\right] = \mathbb{E}_{\mathbb{P}}\left[\frac{\prod_{i=1}^k g_i(\mathbf{y}_{t_i}|\mathbf{x}_{t_i})}{\prod_{i=1}^k \mathbf{L}_{t_i}(g_{t_i})}\right] = \frac{1}{\mathbf{Z}(\mathcal{H}_{t_k})}\mathbb{E}_{\mathbb{P}}\left[\prod_{i=1}^k g_i(\mathbf{y}_{t_i}|\mathbf{x}_{t_i})\right] = 1. \tag{44}$$

**Theorem 3.1** (Multi-marginal Doob's $h$-transform). *Let us define a sequence of functions $\{h_i\}_{i\in[1:k]}$, where each $h_i : [t_{i-1}, t_i) \times \mathbb{R}^d \to \mathbb{R}_+$, for all $i \in [1:k]$, is a conditional expectation $h_i(t, \mathbf{x}_t) := \mathbb{E}_{\mathbb{P}}\left[\prod_{j\geq i}^k f_j(\mathbf{y}_{t_j}|\mathbf{X}_{t_j})|\mathbf{X}_t = \mathbf{x}_t\right]$, where $\{f_i\}_{i\in[1:k]}$ is defined in (3). Now, we define a function $h : [0, T] \times \mathbb{R}^d \to \mathbb{R}_+$ by integrating the functions $\{h_i\}_{i\in[1:k]}$,*

$$h(t, \mathbf{x}) := \sum_{i=1}^k h_i(t, \mathbf{x})\mathbf{1}_{[t_{i-1}, t_i)}(t). \tag{45}$$

*Then, with the initial condition $\mu_0^\star(d\mathbf{x}_0) = h_1(t_0, \mathbf{x}_0)\mu_0(d\mathbf{x}_0)$, the solution of the following conditional SDE inducing the posterior path measure $\mathbb{P}^\star$ in (2):*

$$\text{(Conditioned State)} \quad d\mathbf{X}_t^\star = [b(t, \mathbf{X}_t^\star) + \nabla_{\mathbf{x}}\log h(t, \mathbf{X}_t^\star)]\,dt + d\mathbf{W}_t \tag{46}$$

*Proof.* We start with the interval $[t_{i-1}, t_i)$ without loss of generality. For all $t \in [t_{i-1}, t_i)$ and for any $A_{t_i} \subset \mathcal{B}(\mathbb{R}^d)$, the transition kernel of the conditioned process is defined by the transition kernel of the original Markov process $\mathbf{P}_i$ and $h$-function defined in (3.1):

$$\mathbf{P}_i^{h_i}(\mathbf{X}_{t_i} \in A|\mathbf{X}_t = \mathbf{x}) := \mathbf{P}_i^{h_i}(\mathbf{x}_t, A) = \frac{h_i(t_i, \mathbf{X}_{t_i})}{h_i(t, \mathbf{x}_t)}\mathbf{P}_{t_i}(\mathbf{x}_t, d\mathbf{x}_{t_i}). \tag{47}$$

By the definition of $h$, the transition kernel $\mathbf{P}_{t_i}^h(\mathbf{x}_t, A)$ is a probability kernel:

$$\int_{\mathbb{R}^d} \mathbf{P}_i^{h_i}(\mathbf{x}_t, d\mathbf{x}_{t_i}) = \int_{\mathbb{R}^d} \frac{h_i(t_i, \mathbf{X}_{t_i})}{h_i(t, \mathbf{x}_t)}\mathbf{P}_i(\mathbf{x}_t, d\mathbf{x}_{t_i}) \tag{48}$$

$$= \frac{\int_{\mathbb{R}^d} h_i(t_i, \mathbf{X}_{t_i})\mathbf{P}_i(\mathbf{x}_t, d\mathbf{x}_{t_i})}{h_i(t, \mathbf{x}_t)} \tag{49}$$

$$\overset{(i)}{=} \frac{\int_{\mathbb{R}^d} f_i(\mathbf{y}_{t_i}|\mathbf{X}_{t_i})h_{i+1}(t_i, \mathbf{X}_{t_i})\mathbf{P}_i(\mathbf{x}_t, d\mathbf{x}_{t_i})}{\mathbb{E}_{\mathbb{P}}\left[\prod_{j=i}^k f_j(\mathbf{y}_{t_j}|\mathbf{X}_{t_j})|\mathbf{X}_t = \mathbf{x}_t\right]} \tag{50}$$

$$= \frac{\mathbb{E}_{\mathbb{P}}\left[\prod_{j=i}^k f_j(\mathbf{y}_{t_j}|\mathbf{X}_{t_j})|\mathbf{X}_t = \mathbf{x}_t\right]}{\mathbb{E}_{\mathbb{P}}\left[\prod_{j=i}^k f_j(\mathbf{y}_{t_j}|\mathbf{X}_{t_j})|\mathbf{X}_t = \mathbf{x}_t\right]} = 1, \tag{51}$$

where $(i)$ follows from the recursion established by the definition of $h$ in Theorem 3.1:

$$h_i(t_i, \mathbf{x}_{t_i}) = f_i(\mathbf{y}_{t_i}|\mathbf{X}_{t_i})h_{i+1}(t_i, \mathbf{x}_{t_i}), \quad \forall i \in [1:k-1]. \tag{52}$$

Now, the infinitesimal generator for $\mathbf{P}^h$ can be computed for any $\varphi \in C^{1,2}([0,T] \times \mathbb{R}^d)$ and for all $\mathbb{P}$-almost $\mathbf{x} \in \mathbb{R}^d$:

$$\mathcal{A}_t^{h_i}\varphi_t = \lim_{s \downarrow 0} \frac{\mathbb{E}_{\mathbb{P}^h}\left[\varphi(t_s, \mathbf{X}_{t+s})|\mathbf{X}_t = \mathbf{x}\right] - \varphi(t, \mathbf{x})}{s} \tag{53}$$

$$= \lim_{s \downarrow 0} \frac{\mathbb{E}_{\mathbb{P}}\left[[\varphi(t_s, \mathbf{X}_{t+s}) - \varphi(t, \mathbf{x})]\frac{\mathbf{P}_i^{h_i}(\mathbf{x}, d\mathbf{x}_{t+s})}{\mathbf{P}_i(\mathbf{x}, d\mathbf{x}_{t+s})}|\mathbf{X}_t = \mathbf{x}\right]}{s} \tag{54}$$

$$= \lim_{s \downarrow 0} \frac{\mathbb{E}_{\mathbb{P}}\left[[\varphi(t_s, \mathbf{X}_{t+s}) - \varphi(t, \mathbf{x})]\frac{h_i(t+s, \mathbf{X}_{t+s})}{h_i(t, \mathbf{x})}|\mathbf{X}_t = \mathbf{x}\right]}{s} \tag{55}$$

$$= \lim_{s \downarrow 0} \frac{\mathbb{E}_{\mathbb{P}}\left[[\varphi(t_s, \mathbf{X}_{t+s}) - \varphi(t, \mathbf{x})]\left[\frac{h_i(t+s, \mathbf{X}_{t+s}) - h_i(t, \mathbf{x})}{h_i(t, \mathbf{x})} + 1\right]|\mathbf{X}_t = \mathbf{x}\right]}{s} \tag{56}$$

$$\overset{(i)}{=} \mathcal{A}_t\varphi_t + \lim_{s \downarrow 0} \frac{\mathbb{E}_{\mathbb{P}}\left[[\varphi(t_s, \mathbf{X}_{t+s}) - \varphi(t, \mathbf{x})][h_i(t+s, \mathbf{X}_{t+s}) - h_i(t, \mathbf{x})]|\mathbf{X}_t = \mathbf{x}\right]}{sh_i(t, \mathbf{x})}, \tag{57}$$

where $(i)$ follows from the definition of the infinitesimal generator. Now, we shall compute the second term of the RHS of the equation (57) adapted from (Léonard, 2011). By employing basic stochastic calculus, for a stochastic process $\varphi_t = \varphi(t, \mathbf{X}_t)$ and $h_{i,t} = h_i(t, \mathbf{X}_t)$,

$$\varphi_{t+s}h_{i,t+s} = \varphi_0 h_{i,0} + \int_0^{t+s} \varphi_u dh_{i,u} + \int_0^{t+s} h_{i,u} d\varphi_u + [\varphi, h_i]_{t+s} \tag{58}$$

$$\varphi_t h_{i,t} = \varphi_0 h_{i,0} + \int_0^t \varphi_u dh_{i,u} + \int_0^t h_{i,u} d\varphi_u + [\varphi, h_i]_t, \tag{59}$$

where $[\varphi, h_i]_t = \int_0^t d\varphi_t dh_{i,t}$ is quadratic variation of $\varphi$ and $h_i$. By subtracting equation (59) from equation (58),

$$\varphi_{t+s}h_{i,t+s} - \varphi_t h_{i,t} = \int_t^{t+s} \varphi_u dh_{i,u} + \int_t^{t+s} h_{i,u} d\varphi_u + [\varphi, h_i]_{t+s} - [\varphi, h_i]_t \tag{60}$$

Applying integration by parts leads to the following equation

$$(\varphi_{t+s} - \varphi_t)(h_{i,t+s} - h_{i,t}) = \varphi_{t+s}h_{i,t+s} - \varphi_t h_{i,t} - \varphi_t(h_{i,t+s} - h_{i,t}) - h_{i,t}(\varphi_{t+s} - \varphi_t) \tag{61}$$

$$= \int_t^{t+s} (\varphi_u - \varphi_t)dh_{i,u} + \int_t^{t+s} (h_{i,u} - h_{i,t})d\varphi_u + [\varphi, h_i]_{t+s} - [\varphi, h_i]_t,$$

where $\varphi_t(h_{i,t+s} - h_{i,t}) = \int_t^{t+s} \varphi_t dh_{i,u}$. Moreover, by applying Itô's formula, we get

$$d\varphi_t = \mathcal{A}_t\varphi_t dt + (\nabla_{\mathbf{x}}\varphi)^\top d\mathbf{W}_t, \quad dh_{i,t} = \mathcal{A}_t h_{i,t} dt + (\nabla_{\mathbf{x}}h_{i,t})^\top d\mathbf{W}_t. \tag{62}$$

Therefore, since $\mathbf{W}_t$ is $\mathbb{P}$-martingale,

$$\mathbb{E}_{\mathbb{P}}\left[(\varphi_{t+s} - \varphi_t)(h_{i,t+s} - h_{i,t})|\mathbf{X}_t = \mathbf{x}\right] \tag{63}$$

$$= \mathbb{E}_{\mathbb{P}}\left[\int_s^{t+s} (\varphi_u - \varphi_t)\mathcal{A}_u h_{i,u} du + \int_s^{t+s} (h_{i,u} - h_{i,t})\mathcal{A}_u \varphi_u du + [\varphi, h_i]_{t+s} - [\varphi, h_i]_t|\mathbf{X}_t = \mathbf{x}\right] \tag{64}$$

$$= \underbrace{\mathbb{E}_{\mathbb{P}}^{t,\mathbf{x}}\left[\int_t^{t+s} (\varphi_u - \varphi_t)\mathcal{A}_u h_{i,u} du\right]}_{(A)} + \underbrace{\mathbb{E}_{\mathbb{P}}^{t,\mathbf{x}}\left[\int_t^{t+s} (h_{i,u} - h_{i,t})\mathcal{A}_u \varphi_u du\right]}_{(B)} + \underbrace{\mathbb{E}_{\mathbb{P}}^{t,\mathbf{x}}\left[[\varphi, h_i]_{t+s} - [\varphi, h_i]_t\right]}_{(C)}.$$

For a first term **(A)**, the Hölder's inequality with $1/p + 1/q = 1$ and $p, q \geq 1$ yields,

$$\mathbb{E}_{\mathbb{P}}^{t,\mathbf{x}} \left[ \int_t^{t+s} (\varphi_u - \varphi_t) \mathcal{A}_u h_{i,u} du \right] \leq \left( \mathbb{E}_{\mathbb{P}}^{t,\mathbf{x}} \left[ \int_t^{t+s} |\varphi_u - \varphi_t|^q du \right] \right)^{1/q} \left( \mathbb{E}_{\mathbb{P}}^{t,\mathbf{x}} \left[ \int_t^{t+s} |\mathcal{A}_t h_{i,u}|^p du \right] \right)^{1/p} \tag{65}$$

$$= \left( \mathbb{E}_{\mathbb{P}}^{t,\mathbf{x}} \left[ \int_t^{t+s} |\varphi_u - \varphi_t|^q du \right] \right)^{1/q} \left( \int_t^{t+s} \mathbb{E}_{\mathbb{P}}^{t,\mathbf{x}} \left[ |\mathcal{A}_t h_{i,u}|^p \right] du \right)^{1/p} \tag{66}$$

Given the bounded and continuous function $\varphi$, the dominated convergence theorem yields $(\lim_{s \downarrow 0} \mathbb{E}_{\mathbb{P}}^{t,\mathbf{x}} \left[ \int_t^{t+s} |\varphi_u - \varphi_t|^q du \right])^{1/q} = (\mathbb{E}_{\mathbb{P}}^{t,\mathbf{x}} \left[ \lim_{s \downarrow 0} \int_t^{t+s} |\varphi_u - \varphi_t|^q du \right])^{1/q} = 0$ and since $h_i \in C^{1,2}([t_{i-1}, t_i), \mathbb{R}^d)$ and boundedness of $b$ in Assumptions B, following inequality holds

$$|\mathcal{A}_u h_{i,u}|^p \leq |\partial_t h_{i,t}|^p + |(\nabla_{\mathbf{x}} h_{i,t}^T) b|^p + |\frac{1}{2} Trace \left[ \nabla_{\mathbf{xx}} h_{i,t} \right]|^p < \infty, \tag{67}$$

for any $u \in [t_{i-1}, t_i)$ and $\mathbb{P}$ almost surely. In other words, $\sup_{u \in [t, t+s]} \mathbb{E}_{\mathbb{P}}^{t,\mathbf{x}} \left[ |\mathcal{A}_u h_{i,u}|^p \right] < \infty, \forall t \in [0, T], s > 0$, and $p > 1$. Therefore we get $\lim_{s \downarrow 0} \mathbb{E}_{\mathbb{P}}^{t,\mathbf{x}} \left[ \int_t^{t+s} (\varphi_u - \varphi_t) \mathcal{A}_u h_{i,u} du \right] = 0$ and we can get the similar result for the second term **(B)**, *i.e.*, $\lim_{s \downarrow 0} \mathbb{E}_{\mathbb{P}}^{t,\mathbf{x}} \left[ \int_t^{t+s} (h_{i,u} - h_{i,t}) \mathcal{A}_u \varphi_u du \right] = 0$.

For the last term **(C)**, by the definition of the quadratic variation of $\varphi$ and $h_i$ yields:

$$\mathbb{E}_{\mathbb{P}}^{t,\mathbf{x}} \left[ [\varphi, h_i]_{t+s} - [\varphi, h_i]_t \right] = \mathbb{E}_{\mathbb{P}}^{t,\mathbf{x}} \left[ \int_t^{t+s} d\varphi_u dh_{i,u} \right] = \mathbb{E}_{\mathbb{P}}^{t,\mathbf{x}} \left[ \int_t^{t+s} (\nabla_{\mathbf{x}} \varphi_u)^\top \nabla_{\mathbf{x}} h_{i,u} du \right] \tag{68}$$

Subsequently, by taking the limit from (57), we get the following result:

$$\lim_{s \downarrow 0} \frac{\mathbb{E}_{\mathbb{P}} \left[ [\varphi(t_s, \mathbf{X}_{t+s}) - \varphi(t, \mathbf{x})] [h_i(t+s, \mathbf{X}_{t+s}) - h_i(t, \mathbf{x})] |\mathbf{X}_t = \mathbf{x} \right]}{s h_i(t, \mathbf{x})} \tag{69}$$

$$= \lim_{s \downarrow 0} \frac{\mathbb{E}_{\mathbb{P}} \left[ \int_t^{t+s} (\nabla_{\mathbf{x}} \varphi(u, \mathbf{X}_u))^\top \nabla_{\mathbf{x}} h_i(u, \mathbf{X}_u) du |\mathbf{X}_t = \mathbf{x} \right]}{s h_i(t, \mathbf{x})} \tag{70}$$

$$= (\nabla_{\mathbf{x}} \varphi(t, \mathbf{X}_t))^\top \nabla_{\mathbf{x}} \log h_i(t, \mathbf{X}_t), \tag{71}$$

Therefore, the infinitesimal generator for $\mathbf{P}_i^{h_i}$ is defined by:

$$\mathcal{A}_t^{h_i} \varphi_t = \mathcal{A}_t \varphi_t + (\nabla_{\mathbf{x}} \varphi_t)^\top \nabla_{\mathbf{x}} \log h_{i,t} \tag{72}$$

which shows that the conditioned SDE for an interval $[t_{i-1}, t_i)$ is given by

$$d\mathbf{X}_t^h = [b(t, \mathbf{X}_t) + \nabla_{\mathbf{x}} \log h_i(t, \mathbf{X}_t)] dt + d\mathbf{W}_t \tag{73}$$

Hence, integrating the generators over the entire interval yields:

$$\mathcal{A}_t^h \varphi = \mathcal{A}_t \varphi_t + \sum_{i=1}^k \left[ (\nabla_{\mathbf{x}} \varphi_t)^\top \nabla_{\mathbf{x}} \log h_{i,t} \right] \mathbf{1}_{[t_{i-1}, t_i)}(t) \tag{74}$$

$$= \mathcal{A}_t \varphi_t + (\nabla_{\mathbf{x}} \varphi_t)^\top \nabla_{\mathbf{x}} \log h_t. \tag{75}$$

It implies that the conditioned SDE for the entire interval $[0, T]$ is given by:

$$d\mathbf{X}_t^h = [b(t, \mathbf{X}_t) + \nabla_{\mathbf{x}} \log h(t, \mathbf{X}_t)] dt + d\mathbf{W}_t. \tag{76}$$

Now, assume that $\mathbf{X}_t^h \sim \mu_0^\star(\mathbf{x})$ where $\mu_0^\star(\mathbf{x})$ is absolutely continuous with $\mu_0(\mathbf{x})$. Then, the path measure induced by $\mathbf{X}_t^h$ can be computed as:

$$d\mathbb{P}^h(\mathbf{x}_{0:T}) = d\mu_0^\star(\mathbf{x}_0) \prod_{i=1}^{k} \left[ \prod_{j=1}^{N} \mathbf{P}_{i(j)}^{h_i}(\mathbf{x}_{t_{i(j-1)}}, d\mathbf{x}_{t_i(j)}) \right] \tag{77}$$

$$= d\mu_0^\star(\mathbf{x}_0) \prod_{i=1}^{k} \frac{h_i(t_i, \mathbf{x}_{t_i})}{h_i(t_{i-1}, \mathbf{x}_{t_{i-1}})} \left[ \prod_{j=1}^{N} \mathbf{P}_{i(j)}(\mathbf{x}_{t_{i(j-1)}}, d\mathbf{x}_{t_i(j)}) \right] \tag{78}$$

$$= d\mu_0^\star(\mathbf{x}_0) \prod_{i=1}^{k} \frac{h_{i+1}(t_i, \mathbf{x}_{t_i}) f_i(\mathbf{y}_{t_i}|\mathbf{x}_{t_i})}{h_i(t_{i-1}, \mathbf{x}_{t_{i-1}})} \left[ \prod_{j=1}^{N} \mathbf{P}_{i(j)}(\mathbf{x}_{t_{i(j-1)}}, d\mathbf{x}_{t_i(j)}) \right] \tag{79}$$

$$\overset{N\uparrow\infty}{=} \frac{d\mu_0^\star}{d\mu_0}(\mathbf{x}_0) \frac{h_{k+1}(t_k, \mathbf{x}_{t_k})}{h_1(t_0, \mathbf{x}_0)} \prod_{i=1}^{k} f_i(\mathbf{y}_{t_i}|\mathbf{x}_{t_i}) d\mathbb{P}(\mathbf{x}_{0:T}) \tag{80}$$

where, for all $i \in [1 : k]$, we define a increasing sequence $\{i(j)\}_{j\in[0:N]}$ with $i(0) = i - 1$, $i(1) = i - 1 + \frac{1}{N}$ and $i(N) = i$. Hence, for a $d\mu_0^\star(\mathbf{x}_0) = h_1(t_0, \mathbf{x}_0) d\mu_0(\mathbf{x}_0)$ and $h_{k+1} = 1$ yields

$$d\mathbb{P}^h(\mathbf{x}_{0:T}) = \prod_{i=1}^{k} f_i(\mathbf{y}_{t_i}|\mathbf{x}_{t_i}) d\mathbb{P}(\mathbf{x}_{0:T}) \tag{81}$$

$$= \frac{1}{\mathbf{Z}(\mathcal{H}_{t_k})} \prod_{i=1}^{k} g_i(\mathbf{y}_{t_i}|\mathbf{x}_{t_i}) d\mathbb{P}(\mathbf{x}_{0:T}) \tag{82}$$

$$= d\mathbb{P}^\star(\mathbf{x}_{0:T}). \tag{83}$$

It concludes the proof. $\qquad\square$

### B.2 PROOF OF THEOREM 3.2

**Theorem 3.2** (Dynamic Programming Principle). *Let us consider a sequence of left continuous functions* $\{\mathcal{V}_i\}_{i\in[1:k+1]}$, *where each* $\mathcal{V}_i \in C^{1,2}([t_{i-1}, t_i) \times \mathbb{R}^d)$

$$\mathcal{V}_i(t, \mathbf{x}_t) := \min_{\alpha \in \mathbb{A}} \mathbb{E}_{\mathbb{P}^\alpha} \left[ \int_{t_{i-1}}^{t_i} \frac{1}{2} \|\alpha_s\|^2 ds - \log f_i(\mathbf{y}_{t_i}|\mathbf{X}_{t_i}^\alpha) + \mathcal{V}_{i+1}(t_i, \mathbf{X}_{t_i}^\alpha)|\mathbf{X}_t = \mathbf{x}_t \right], \tag{84}$$

*for all* $i \in [1 : k]$ *and* $\mathcal{V}_{k+1} = 0$. *Then, for any* $0 \le t \le u \le T$, *the value function* $\mathcal{V}$ *for the cost function in (7) satisfying the recursion defined as follows:*

$$\mathcal{V}(t, \mathbf{x}_t) = \min_{\alpha \in \mathbb{A}} \mathbb{E}_{\mathbb{P}^\alpha} \left[ \int_{t}^{t_{I(u)}} \frac{1}{2} \|\alpha_s\|^2 ds - \sum_{i:\{t \le t_i \le t_{I(u)}\}} \log f_i(\mathbf{y}_{t_i}|\mathbf{X}_{t_i}^\alpha) + \mathcal{V}_{I(u)+1}(t_{I(u)}, \mathbf{X}_{t_{I(u)}}^\alpha)|\mathbf{X}_t^\alpha = \mathbf{x}_t \right], \tag{85}$$

*with the indexing function* $I(u) = \max\{i \in [1 : k]|t_i \le u\}$.

*Proof.* Following the approach used in the proof of the standard dynamic programming principle with the flow property induced by Markov control (Van Handel, 2007), we can apply similar methods to our cost function. We start the proof by establishing the recursion of $\mathcal{J}$. Let us define the sequence of left continuous cost functions $\{\mathcal{J}_i\}_{i\in[1:k+1]}$:

$$\mathcal{J}_i(t, \mathbf{x}_t, \alpha) := \mathbb{E}_{\mathbb{P}^\alpha}^{t, \mathbf{x}_t} \left[ \int_{t_{i-1}}^{t_i} \frac{1}{2} \|\alpha_s\|^2 ds - \log f_i(\mathbf{y}_{t_i}|\mathbf{X}_{t_i}^\alpha) + \mathcal{J}_{i+1}(t_i, \mathbf{X}_{t_i}^\alpha, \alpha) \right], \tag{86}$$

where we denote $\mathbb{E}_{\mathbb{P}}^{\mathbf{t}, \mathbf{x}}[\cdot] = \mathbb{E}_{\mathbb{P}}[\cdot|\mathbf{X}_t = \mathbf{x}]$, for all $i \in [1 : k]$ and $\mathcal{J}_{k+1} = 0$. Since $\mathbb{P}^\alpha$ is Markov process, it satisfying following recursion, for any $0 \le t \le u \le T$,

$$\mathcal{J}(t, \mathbf{x}_t, \alpha) = \mathbb{E}_{\mathbb{P}^\alpha}^{t, \mathbf{x}_t} \left[ \int_{t}^{t_{I(u)}} \frac{1}{2} \|\alpha_s\|^2 ds - \sum_{i:\{t \le t_i \le t_{I(u)}\}} \log f_i(\mathbf{y}_{t_i}|\mathbf{X}_{t_i}^\alpha) + \mathcal{J}_{I(u)+1}(t_{I(u)}, \mathbf{X}_{t_{I(u)}}^\alpha, \alpha) \right], \tag{87}$$

with the indexing function $I(u) = \max\{i \in [1:k] | t_i \leq u\}$.

For any $\epsilon > 0$, there exists a control $\alpha' \in \mathbb{A}[t, T]$ such that

$$\mathcal{V}(t, \mathbf{x}) + \epsilon \geq \mathcal{J}(t, \mathbf{x}, \alpha') \tag{88}$$

$$= \mathbb{E}^{t,\mathbf{x}_t}_{\mathbb{P}^{\alpha'}} \left[ \int_t^{t_{I(u)}} \frac{1}{2} \|\alpha'_s\|^2 \, ds - \sum_{i:\{t \leq t_i \leq t_{I(u)}\}} \log f_i(\mathbf{y}_{t_i} | \mathbf{X}^{\alpha'}_{t_i}) + \mathcal{J}_{I(u)+1}(t_{I(u)}, \mathbf{X}^{\alpha'}_{t_{I(u)}}, \alpha') \right] \tag{89}$$

$$\geq \mathbb{E}^{t,\mathbf{x}_t}_{\mathbb{P}^{\alpha'}} \left[ \int_t^{t_{I(u)}} \frac{1}{2} \|\alpha'_s\|^2 \, ds - \sum_{i:\{t \leq t_i \leq t_{I(u)}\}} \log f_i(\mathbf{y}_{t_i} | \mathbf{X}^{\alpha'}_{t_i}) + \mathcal{V}_{I(u)+1}(t_{I(u)}, \mathbf{X}^{\alpha'}_{t_{I(u)}}) \right] \tag{90}$$

$$\geq \min_{\alpha' \in \mathbb{A}[t,T]} \mathbb{E}^{t,\mathbf{x}_t}_{\mathbb{P}^{\alpha'}} \left[ \int_t^{t_{I(u)}} \frac{1}{2} \|\alpha'_s\|^2 \, ds - \sum_{i:\{t \leq t_i \leq t_{I(u)}\}} \log f_i(\mathbf{y}_{t_i} | \mathbf{X}^{\alpha'}_{t_i}) + \mathcal{V}_{I(u)+1}(t_{I(u)}, \mathbf{X}^{\alpha'}_{t_{I(u)}}) \right] \tag{91}$$

Since $\epsilon$ was arbitrary, limiting $\epsilon \to 0$ we get:

$$\mathcal{V}(t, \mathbf{x}) \geq \min_{\alpha' \in \mathbb{A}[t,T]} \mathbb{E}^{t,\mathbf{x}_t}_{\mathbb{P}^{\alpha'}} \left[ \int_t^{t_{I(u)}} \frac{1}{2} \|\alpha'_s\|^2 \, ds - \sum_{i:\{t \leq t_i \leq t_{I(u)}\}} \log f_i(\mathbf{y}_{t_i} | \mathbf{X}^{\alpha'}_{t_i}) + \mathcal{V}_{I(u)+1}(t_{I(u)}, \mathbf{X}^{\alpha'}_{t_{I(u)}}) \right]. \tag{92}$$

For the reverse direction, consider the control $\tilde{\alpha} \in \mathbb{A}[t, T]$ obtained from integrating:

$$\tilde{\alpha}_s := \begin{cases} \alpha^1_s, & s \in [t, t_{I(u)}) \\ \alpha^2_s & s \in [t_{I(u)}, T]. \end{cases} \tag{93}$$

Then, by following the definition of the value function

$$\mathcal{J}(t, \mathbf{x}, \tilde{\alpha}) \geq \min_{\alpha^2 \in \mathbb{A}[t_{I(u)}, T]} \mathcal{J}(t, \mathbf{x}, \tilde{\alpha}) \tag{94}$$

$$= \mathbb{E}^{t,\mathbf{x}_t}_{\mathbb{P}^{\alpha^1}} \left[ \int_t^{t_{I(u)}} \frac{1}{2} \|\alpha^1_s\|^2 \, ds - \sum_{i:\{t \leq t_i \leq t_{I(u)}\}} \log f_i(\mathbf{y}_{t_i} | \mathbf{X}^{\alpha^1}_{t_i}) + \mathcal{V}_{I(u)+1}(t_{I(u)}, \mathbf{X}^{\alpha^1}_{t_{I(u)}}) \right] \tag{95}$$

$$\geq \min_{\alpha^1 \in \mathbb{A}[t, t_{I(u)})} \mathbb{E}^{t,\mathbf{x}_t}_{\mathbb{P}^{\alpha^1}} \left[ \int_t^{t_{I(u)}} \frac{1}{2} \|\alpha^1_s\|^2 \, ds - \sum_{i:\{t \leq t_i \leq t_{I(u)}\}} \log f_i(\mathbf{y}_{t_i} | \mathbf{X}^{\alpha^1}_{t_i}) + \mathcal{V}_{I(u)+1}(t_{I(u)}, \mathbf{X}^{\alpha^1}_{t_{I(u)}}) \right] \tag{96}$$

$$= \min_{\tilde{\alpha} \in \mathbb{A}[t, T]} \mathbb{E}^{t,\mathbf{x}_t}_{\mathbb{P}^{\tilde{\alpha}}} \left[ \int_t^{t_{I(u)}} \frac{1}{2} \|\tilde{\alpha}_s\|^2 \, ds - \sum_{i:\{t \leq t_i \leq t_{I(u)}\}} \log f_i(\mathbf{y}_{t_i} | \mathbf{X}^{\tilde{\alpha}}_{t_i}) + \mathcal{V}_{I(u)+1}(t_{I(u)}, \mathbf{X}^{\tilde{\alpha}}_{t_{I(u)}}) \right] \tag{97}$$

$$\geq \mathcal{V}(t, \mathbf{x}). \tag{98}$$

Combining both inequalities in (92, 97-98), we arrive at the desired result:

$$\mathcal{V}(t, \mathbf{x}) = \min_{\alpha \in \mathbb{A}} \mathbb{E}_{\mathbb{P}^{\alpha}} \left[ \int_t^{t_{I(u)}} \frac{1}{2} \|\alpha_s\|^2 \, ds - \sum_{i:\{t \leq t_i \leq t_{I(u)}\}} \log f_i(\mathbf{y}_{t_i} | \mathbf{X}^{\alpha}_{t_i}) + \mathcal{V}_{I(u)+1}(t_{I(u)}, \mathbf{X}^{\alpha}_{t_{I(u)}}) | \mathbf{X}^{\alpha}_t = \mathbf{x}_t \right]. \tag{99}$$

This concludes the proof. $\square$

### B.3 PROOF OF THEOREM 3.3

**Theorem 3.3** (Verification Theorem). *Suppose there exist a sequence of left continuous functions $\mathcal{V}_i(t, \mathbf{x}) \in C^{1,2}([t_{i-1}, t_i), \mathbb{R}^d)$, for all $i \in [1 : k]$, satisfying the following Hamiltonian-Jacobi-Bellman (HJB) equation:*

$$\partial_t \mathcal{V}_{i,t} + \mathcal{A}_t \mathcal{V}_{i,t} + \min_{\alpha \in \mathbb{A}} \left[ (\nabla_{\mathbf{x}} \mathcal{V}_{i,t})^\top \alpha_{i,t} + \frac{1}{2} \|\alpha_{i,t}\|^2 \right] = 0, \quad t_{i-1} \leq t < t_i \tag{100}$$

$$\mathcal{V}_i(t_i, \mathbf{x}) = -\log f_i(\mathbf{y}_{t_i} | \mathbf{x}) + \mathcal{V}_{i+1}(t_i, \mathbf{x}), \quad t = t_i, \quad \forall i \in [1 : k], \tag{101}$$

*where a minimum is attained by* $\alpha_i^\star(t, \mathbf{x}) = \nabla_{\mathbf{x}} \mathcal{V}_i(t, \mathbf{x})$. *Now, define a function* $\alpha : [0, t_k] \times \mathbb{R}^d \to \mathbb{R}^d$ *by integrating the optimal controls* $\{\alpha_i\}_{i \in \{1, \cdots, k\}}$,

$$\alpha^\star(t, \mathbf{x}) := \sum_{i=1}^{k} \alpha_i^\star(t, \mathbf{x}) \mathbf{1}_{[t_{i-1}, t_i)}(t) \tag{102}$$

*Then,* $\mathcal{J}(t, \mathbf{x}_t, \alpha^\star) \leq \mathcal{J}(t, \mathbf{x}_t, \alpha)$ *holds for any* $(t, \mathbf{x}_t) \in [0, T] \times \mathbb{R}^d$ *and* $\alpha \in \mathbb{A}$. *In other words,* $\alpha^\star$ *is optimal control for* $\mathcal{V}$ *in* (9).

*Proof.* Without loss of generality, consider $t \in [t_{i-1}, t_i)$. By applying the Itô's formula to the value function $\mathcal{V}$ and taking expectation with respect to $\mathbb{P}^\alpha$, we obtain

$$\mathbb{E}_{\mathbb{P}^\alpha}^{t, \mathbf{x}_t} \left[ \mathcal{V}_i(t_i, \mathbf{X}_{t_i}^\alpha) \right] = \mathcal{V}_i(t, \mathbf{x}) + \mathbb{E}_{\mathbb{P}^\alpha}^{t, \mathbf{x}_t} \left[ \int_t^{t_i} \left( \partial_t \mathcal{V}_{i,s} + \mathcal{A}_t \mathcal{V}_{i,s} + (\nabla_{\mathbf{x}} \mathcal{V}_{i,s})^\top \alpha_{i,s} \right) ds \right], \tag{103}$$

where we denote $\mathbb{E}_{\mathbb{P}}^{t, \mathbf{x}}[\cdot] = \mathbb{E}_{\mathbb{P}}[\cdot | \mathbf{X}_t = \mathbf{x}]$. By adding the Lagrangian term $\mathbb{E}_{\mathbb{P}^\alpha}^{t, \mathbf{x}_t} \left[ \int_t^{t_i} \frac{1}{2} \|\alpha_{i,s}\|^2 ds \right]$ to both sides of (103), we get for the LHS of (103)

$$\text{LHS} = \mathbb{E}_{\mathbb{P}^\alpha}^{t, \mathbf{x}_t} \left[ \mathcal{V}_i(t_i, \mathbf{X}_{t_i}^\alpha) \right] + \mathbb{E}_{\mathbb{P}^\alpha}^{t, \mathbf{x}_t} \left[ \int_t^{t_i} \frac{1}{2} \|\alpha_{i,s}\|^2 ds \right] \tag{104}$$

$$= \mathbb{E}_{\mathbb{P}^\alpha}^{t, \mathbf{x}_t} \left[ \mathcal{V}_i(t_i, \mathbf{X}_{t_i}^\alpha) + \int_t^{t_i} \frac{1}{2} \|\alpha_{i,s}\|^2 ds \right] \tag{105}$$

$$\overset{(i)}{=} \mathbb{E}_{\mathbb{P}^\alpha}^{t, \mathbf{x}_t} \left[ \int_t^{t_i} \frac{1}{2} \|\alpha_{i,s}\|^2 ds - \log f_i(\mathbf{y}_{t_i} | \mathbf{X}_{t_i}^\alpha) + \mathcal{V}_{i+1}(t_i, \mathbf{X}_{t_i}^\alpha) \right] \tag{106}$$

$$= \mathcal{J}_i(t, \mathbf{x}, \alpha), \tag{107}$$

where $(i)$ follows from the definition of HJB equation in (11). Now for the RHS of (103), we have:

$$\text{RHS} = \mathcal{V}_i(t, \mathbf{x}) + \mathbb{E}_{\mathbb{P}^\alpha}^{t, \mathbf{x}_t} \left[ \int_t^{t_i} \left( \partial_t \mathcal{V}_{i,s} + \mathcal{A}_t \mathcal{V}_{i,s} + (\nabla_{\mathbf{x}} \mathcal{V}_{i,s})^\top \alpha_{i,s} \right) ds \right] + \mathbb{E}_{\mathbb{P}^\alpha}^{t, \mathbf{x}_t} \left[ \int_t^{t_i} \frac{1}{2} \|\alpha_{i,s}\|^2 ds \right] \tag{108}$$

$$= \mathcal{V}_i(t, \mathbf{x}) + \mathbb{E}_{\mathbb{P}^\alpha}^{t, \mathbf{x}_t} \left[ \int_t^{t_i} \left( \partial_t \mathcal{V}_{i,s} + \mathcal{A}_t \mathcal{V}_{i,s} + \left[ (\nabla_{\mathbf{x}} \mathcal{V}_{i,s})^\top \alpha_{i,s} + \frac{1}{2} \|\alpha_{i,s}\|^2 \right] \right) ds \right] \tag{109}$$

$$\overset{(i)}{=} \mathcal{V}_i(t, \mathbf{x}) + \mathbb{E}_{\mathbb{P}^\alpha}^{t, \mathbf{x}_t} \left[ \int_t^{t_i} \left( \left[ (\nabla_{\mathbf{x}} \mathcal{V}_{i,s})^\top \alpha_{i,s} + \frac{1}{2} \|\alpha_{i,s}\|^2 \right] - \min_{\alpha \in \mathbb{A}} \left[ (\nabla_{\mathbf{x}} \mathcal{V}_{i,s})^\top \alpha_{i,s} + \frac{1}{2} \|\alpha_{i,s}\|^2 \right] \right) ds \right], \tag{110}$$

where $(i)$ follows from the definition of HJB equation in (10). Therefore, we get the following result: $\mathcal{J}_i(t, \mathbf{x}, \alpha) = \mathcal{V}_i(t, \mathbf{x})$

$$+ \mathbb{E}_{\mathbb{P}^\alpha}^{t, \mathbf{x}_t} \left[ \int_t^{t_i} \left( \left[ (\nabla_{\mathbf{x}} \mathcal{V}_{i,s})^\top \alpha_{i,s} + \frac{1}{2} \|\alpha_{i,s}\|^2 \right] - \min_{\alpha \in \mathbb{A}} \left[ (\nabla_{\mathbf{x}} \mathcal{V}_{i,s})^\top \alpha_{i,s} + \frac{1}{2} \|\alpha_{i,s}\|^2 \right] \right) ds \right]. \tag{111}$$

Due to the fact that

$$\int_t^{t_i} \left( \left[ (\nabla_{\mathbf{x}} \mathcal{V}_{i,s})^\top \alpha_{i,s} + \frac{1}{2} \|\alpha_{i,s}\|^2 \right] - \min_{\alpha \in \mathbb{A}} \left[ (\nabla_{\mathbf{x}} \mathcal{V}_{i,s})^\top \alpha_{i,s} + \frac{1}{2} \|\alpha_{i,s}\|^2 \right] \right) ds \geq 0,$$

for all $t \in [t_{i-1}, t_i)$ and $\mathbb{P}^\alpha$ almost $\mathbf{x}$, we conclude that $\mathcal{J}_i(t, \mathbf{x}, \alpha) \geq \mathcal{V}_i(t, \mathbf{x})$, where the equality holds for $\alpha_{i,t}^\star = \min_{\alpha \in \mathbb{A}} \left[ (\nabla_{\mathbf{x}} \mathcal{V}_{i,t})^\top \alpha_{i,t} + \frac{1}{2} \|\alpha_{i,t}\|^2 \right] = -\nabla_{\mathbf{x}} \mathcal{V}_{i,t}$. Additionaly, it implies that $\mathcal{J}_i(t, \mathbf{x}, \alpha^\star) = \mathcal{V}_i(t, \mathbf{x})$. Subsequently, for any $t \in [t_{i-1}, t_i)$, the recursion in (9) yields:

$$\mathcal{V}(t, \mathbf{x}_t) = \min_{\alpha \in \mathbb{A}} \mathbb{E}_{\mathbb{P}^\alpha}^{t, \mathbf{x}_t} \left[ \int_t^{t_{I(u)}} \frac{1}{2} \|\alpha_s\|^2 ds - \sum_{i:\{t \leq t_i \leq t_{I(u)}\}} \log f_i(\mathbf{y}_{t_i} | \mathbf{X}_{t_i}^\alpha) + \mathcal{V}_{I(u)+1}(t_{I(u)}, \mathbf{X}_{t_{I(u)}}^\alpha) \right] \tag{112}$$

$$= \min_{\alpha \in \mathbb{A}} \mathbb{E}_{\mathbb{P}^\alpha}^{t, \mathbf{x}_t} \left[ \int_t^{t_i} \frac{1}{2} \|\alpha_{i,s}\|^2 ds - \log f_i(\mathbf{y}_{t_i} | \mathbf{X}_{t_i}^\alpha) + \mathcal{V}_{i+1}(t_i, \mathbf{X}_{t_i}^\alpha) \right] \tag{113}$$

$$= \mathcal{V}_i(t, \mathbf{x}_t). \tag{114}$$

This implies that the optimal control for the value function $\mathcal{V}$ in (9) over the interval $t \in [t_{i-1}, t_i)$ is $\alpha_i^\star$. Finally, $\mathcal{V}$ in (9) can be represented as the integrated form :

$$\mathcal{V}(t, \mathbf{x}_t) = \sum_{i=1}^k \mathcal{V}_i(t, \mathbf{x}_t) \mathbf{1}_{[t_{i-1}, t_i)}. \tag{115}$$

Therefore, the optimal control for $\mathcal{V}$ in (9) becomes $\alpha(t, \mathbf{x}) = \sum_{i=1}^k \alpha_i^\star(t, \mathbf{x}) \mathbf{1}_{[t_{i-1}, t_i)}(t)$. $\qquad\square$

### B.4 PROOF OF LEMMA 3.4

We first restate the Feynman-Kac formula (Oksendal, 1992; Baldi, 2017), which gives a probabilistic representation of the solution to certain PDEs using expectations of stochastic processes. It relies on the fact that the conditional expectation is martingale process.

**Lemma B.5** (The Feynman-Kac formula). *Let us define $f \in C^2(\mathbb{R}^d)$ and $g \in C(\mathbb{R}^d)$. Then, a function $h(t, \mathbf{x}_t) = \mathbb{E}_{\mathbb{P}}\left[ e^{-\int_t^T f(s, \mathbf{X}_s) ds} g(\mathbf{X}_T) | \mathbf{X}_t = \mathbf{x}_t \right]$ is a solution of the following linear PDE:*

$$\partial_t h_t + \mathcal{A}_t h_t - f h_t = 0, \quad 0 \le t < T, \tag{116}$$
$$h(t, \mathbf{x}) = g(\mathbf{X}_T), \quad t = T. \tag{117}$$

*Proof.* Define the process $\mathbf{Y}_t = e^{-\int_t^T f(s, \mathbf{X}_s) ds} h(t, \mathbf{X}_t)$. Since $h$ is a conditional expectation with respect to $\mathbb{P}$, implies that $\mathbf{Y}_t$ is martingale process. By applinyg Itô formula, we have:

$$d\mathbf{Y}_t = -f(t, \mathbf{X}_t) e^{-\int_t^T f(s, \mathbf{X}_s) ds} h(t, \mathbf{X}_t) dt + e^{-\int_t^T f(s, \mathbf{X}_s) ds} dh(t, \mathbf{X}_t). \tag{118}$$

Next, we apply Itô formula to $h(t, \mathbf{X}_t)$:

$$dh(t, \mathbf{X}_t) = \left( \frac{\partial h}{\partial t} + \mathcal{A}_t h_t \right) dt + \nabla_{\mathbf{x}} h(t, \mathbf{X}_t)^\top d\mathbf{W}_t, \tag{119}$$

Thus, by substituting equation (119) into equation (118), we get

$$d\mathbf{Y}_t = -f(t, \mathbf{X}_t) \mathbf{Y}_t dt + e^{-\int_t^T f(s, \mathbf{X}_s) ds} \left( \frac{\partial h}{\partial t} + \mathcal{A}_t h_t \right) dt + e^{-\int_t^T f(s, \mathbf{X}_s) ds} \nabla_{\mathbf{x}} h(t, \mathbf{X}_t)^\top d\mathbf{W}_t. \tag{120}$$

For $\mathbf{Y}_t$ to be a martingale process, it will have zero drift. Therefore, it implies that

$$\frac{\partial h}{\partial t}(t, \mathbf{X}_t) + \mathcal{A}_t h(t, \mathbf{X}_t) - f(t, \mathbf{X}_t) h(t, \mathbf{X}_t) = 0, \tag{121}$$

where $h(T, \mathbf{X}_T) = g(\mathbf{X}_T)$ by definition. It concludes the proof. $\qquad\square$

Now let us provide the proof of the Lemma 3.4.

**Lemma 3.4** (Hopf-Cole Transformation). *The $h$ function satisfying the following linear PDE:*

$$\partial_t h_{i,t} + \mathcal{A}_t h_{i,t} = 0, \quad t_{i-1} \le t < t_i \tag{122}$$
$$h_i(t_i, \mathbf{x}) = f_i(\mathbf{y}_{t_i} | \mathbf{x}) h_{i+1}(t_i, \mathbf{x}), \quad t = t_i, \quad \forall i \in [1 : k]. \tag{123}$$

*Moreover, for a logarithm transformation $\mathcal{V} = -\log h$, $\mathcal{V}$ satisfying the HJB equation in (10-11).*

*Proof.* The linaer PDE presented in (122-123) can be directly derived from the function $h_i(t, \mathbf{x}_t) = \mathbb{E}_{\mathbb{P}}\left[ \prod_{j \ge i-1}^k f_j(\mathbf{X}_{t_j}) | \mathbf{X}_t = \mathbf{x}_t \right]$. This is done by applying the Feynman-Kac formula in Lemma B.5 with $f := 0$ and $g = \prod_{j=i-1}^k f_j(\mathbf{X}_{t_j})$. Now, let us consider the function $\mathcal{V}_i(t, \mathbf{x}) = -\log h_i(t, \mathbf{x})$ (or $h_i(t, \mathbf{x}) = e^{-\mathcal{V}_i(t, \mathbf{x})}$) for all $i \in [1 : k]$ and compute

$$\partial_t h_{i,t} = -h_{i,t} \partial_t \mathcal{V}_{i,t}, \quad \nabla_{\mathbf{x}} h_{i,t} = -h_{i,t} \nabla_{\mathbf{x}} \mathcal{V}_{i,t}, \quad \nabla_{\mathbf{xx}} h_{i,t} = h_{i,t}(\|\nabla_{\mathbf{x}} \mathcal{V}_{i,t}\|^2 - \nabla_{\mathbf{xx}} \mathcal{V}_{i,t}). \tag{124}$$

Then, it is straightforward to compute:

$$h_{i,t}\partial_t \mathcal{V}_{i,t} = -\partial_t h_{i,t} \overset{(i)}{=} \mathcal{A}_t h_{i,t} \tag{125}$$

$$= (\nabla_{\mathbf{x}} h_{i,t})^\top b_t + \frac{1}{2} Trace\left[\nabla_{\mathbf{xx}} h_{i,t}\right] \tag{126}$$

$$= (-h_{i,t}\nabla_{\mathbf{x}}\mathcal{V}_{i,t})^\top b_t + \frac{1}{2} Trace\left[h_{i,t}(\|\nabla_{\mathbf{x}}\mathcal{V}_{i,t}\|^2 - h_{i,t}\nabla_{\mathbf{xx}}\mathcal{V}_{i,t})\right] \tag{127}$$

$$= (-h_{i,t}\nabla_{\mathbf{x}}\mathcal{V}_{i,t})^\top b_t + \frac{1}{2} Trace\left[h_{i,t}\|\nabla_{\mathbf{x}}\mathcal{V}_{i,t}\|^2\right] - \frac{1}{2} Trace\left[h_{i,t}\nabla_{\mathbf{xx}}\mathcal{V}_{i,t}\right], \tag{128}$$

where $(i)$ follows from (122). Now, we can simplify (128) by dividing both sides with $h > 0$:

$$\partial_t \mathcal{V}_{i,t} = (-\nabla_{\mathbf{x}}\mathcal{V}_{i,t})^\top b_t + \frac{1}{2} Trace\left[\|\nabla_{\mathbf{x}}\mathcal{V}_{i,t}\|^2\right] - \frac{1}{2} Trace\left[\nabla_{\mathbf{xx}}\mathcal{V}_{i,t}\right] \tag{129}$$

$$= -\mathcal{A}_t \mathcal{V}_{i,t} + \frac{1}{2} \|\nabla_{\mathbf{x}}\mathcal{V}_{i,t}\|^2 \tag{130}$$

Therefore, combining the above results, we have

$$\partial_t \mathcal{V}_{i,t} + \mathcal{A}_t \mathcal{V}_{i,t} - \frac{1}{2} \|\nabla_{\mathbf{x}}\mathcal{V}_{i,t}\|^2 = 0, \quad \mathcal{V}_i(t_i, \mathbf{x}) = -\log f_i(\mathbf{y}_{t_i}|\mathbf{x}) + \mathcal{V}_{i+1}(t_i, \mathbf{x}). \tag{131}$$

Since $\min_{\alpha \in \mathbb{A}}\left[(\nabla_{\mathbf{x}}\mathcal{V}_{i,t})^\top \alpha_{i,t} + \frac{1}{2}\|\alpha_{i,t}\|^2\right] = -\frac{1}{2}\|\nabla_{\mathbf{x}}\mathcal{V}_{i,t}\|^2$, this concludes the proof. $\square$

## B.5 PROOF OF COROLLARY 3.5

**Corollary 3.5** (Optimal Control). *For optimal control $\alpha^\star$ induced by the cost function (7) with dynamics (6), it satisfies $\alpha^\star = \nabla_{\mathbf{x}}\log h$. In other words, we can simulate the conditional SDEs in (5) by simulating the controlled SDE (6) with optimal control $\alpha^\star$.*

*Proof.* Lemma 3.4 implies that we can obtain the relation $-\mathcal{V}_i(t, \mathbf{x}) = \log h_i(t, \mathbf{x})$. Moreover, by following the definition of the optimal control $\alpha^\star$ in (12) and the value function $\mathcal{V}$ in (115), it suggest that $\alpha^\star(t, \mathbf{x}) = -\nabla_{\mathbf{x}}\mathcal{V}(t, \mathbf{x})$ for all $t \in [0, T]$. Finally, combining the definition of the $h$ function in (4) and the results from Lemma 3.4, we can conclude that $\alpha^\star(t, \mathbf{x}) = -\nabla_{\mathbf{x}}\mathcal{V}(t, \mathbf{x}) = \nabla_{\mathbf{x}}\log h(t, \mathbf{x})$ for all $t \in [0, T]$. $\square$

## B.6 PROOF OF THEOREM 3.6

The Donsker–Varadhan variational principle provides a variational formula for the large deviations of functionals of Brownian motion, often related to free-energy minimization problems (Boué & Dupuis, 1998). Moreover, through Girsanov's theorem, this principle extends to a wide range of Markov processes, including Itô diffusion processes (Hartmann et al., 2017; Tzen & Raginsky, 2019).

**Lemma B.6** (Donsker–Varadhan Variational Principle). *For a bounded and measurable functions $\mathcal{W} : C([0, T], \mathbb{R}^d) \to \mathbb{R}$, following relation holds:*

$$-\log \mathbb{E}_{\mathbf{X}\sim\mathbb{P}}\left[e^{-\mathcal{W}(\mathbf{X}_{0:T})}\right] = \min_{\mathbb{Q}\ll\mathbb{P}}\left[\mathbb{E}_{\mathbf{Y}\sim\mathbb{Q}}\left[\mathcal{W}(\mathbf{Y}_{0:T})\right] + D_{KL}(\mathbb{Q}|\mathbb{P})\right] \tag{132}$$

*Proof.* The proof relies on the change of measure and the Jensen's inequality:

$$-\log \mathbb{E}_{\mathbf{X}\sim\mathbb{P}}\left[e^{-\mathcal{W}(\mathbf{X}_{0:T})}\right] = -\log \mathbb{E}_{\mathbf{Y}\sim\mathbb{Q}}\left[e^{-\mathcal{W}(\mathbf{Y}_{0:T})}\frac{d\mathbb{P}}{d\mathbb{Q}}(\mathbf{Y}_{0:T})\right] \tag{133}$$

$$\leq \mathbb{E}_{\mathbf{Y}\sim\mathbb{Q}}\left[\mathcal{W}(\mathbf{Y}_{0:T}) - \log \frac{d\mathbb{P}}{d\mathbb{Q}}(\mathbf{Y}_{0:T})\right] \tag{134}$$

$$= \mathbb{E}_{\mathbf{Y}\sim\mathbb{Q}}\left[\mathcal{W}(\mathbf{Y}_{0:T})\right] + D_{\mathrm{KL}}(\mathbb{Q}|\mathbb{P}), \tag{135}$$

where the equality holds if and only if

$$\frac{d\mathbb{Q}}{d\mathbb{P}}(\mathbf{Y}_{0:T}) = e^{-\log \mathbb{E}_{\mathbf{X}\sim\mathbb{P}}\left[e^{-\mathcal{W}(\mathbf{X}_{0:T})}\right] - \mathcal{W}(\mathbf{Y}_{0:T})}. \tag{136}$$

It concludes the proof. $\square$

**Theorem 3.6** (Tight Variational Bound). *Let us assume that the path measure $\mathbb{P}^\alpha$ induced by (6) for any $\alpha \in \mathbb{A}$ satisfies $D_{KL}(\mathbb{P}^\alpha|\mathbb{P}^\star) < \infty$. Then, for a cost function $\mathcal{J}$ in (7) and $\mu_0^\star$ in (5), it holds:*

$$D_{KL}(\mathbb{P}^\alpha|\mathbb{P}^\star) = D_{KL}(\mu_0|\mu_0^\star) + \mathbb{E}_{\mathbf{x}_0 \sim \mu_0}\left[\mathcal{J}(0, \mathbf{x}_0, \alpha)\right] = \mathcal{L}(\alpha) + \log g(\mathcal{H}_{t_k}) \geq 0, \qquad (137)$$

*where the objective function $\mathcal{L}(\alpha)$ (negative ELBO) is given by:*

$$\mathcal{L}(\alpha) = \mathbb{E}_{\mathbb{P}^\alpha}\left[\int_0^T \frac{1}{2}\|\alpha(s, \mathbf{X}_s^\alpha)\|^2 \, ds - \sum_i^k \log g_i(\mathbf{y}_{t_i}|\mathbf{X}_{t_i}^\alpha)\right] \geq -\log g(\mathcal{H}_{t_k}). \qquad (138)$$

*Moreover, assume that $\mathcal{L}(\alpha)$ has a global minimum $\alpha^\star = \arg\min_{\alpha \in \mathbb{A}} \mathcal{L}(\alpha)$. Then the equality holds in (16) i.e., $\mathcal{L}(\alpha^\star) = -\log g(\mathcal{H}_{t_k})$ and $\mu_0^\star = \mu_0$ almost everywhere with respect to $\mu_0$.*

*Proof.* We begin by deriving the KL-divergence between $\mathbb{P}^\alpha$ and $\mathbb{P}^\star$:

$$D_{KL}(\mathbb{P}^\alpha|\mathbb{P}^\star) \stackrel{(i)}{=} D_{KL}(\mu_0|\mu_0^\star) + \mathbb{E}_{\mathbf{x}_0 \sim \mu_0}\left[D_{KL}(\mathbb{P}^\alpha(\cdot|\mathbf{X}_0^\alpha)|\mathbb{P}^\star(\cdot|\mathbf{X}_0^\alpha))|\mathbf{X}_0^\star = \mathbf{x}_0\right] \qquad (139)$$

$$= D_{KL}(\mu_0|\mu_0^\star) + \mathbb{E}_{\mathbf{x}_0 \sim \mu_0}\mathbb{E}_{\mathbb{P}^\alpha}\left[\log\frac{d\mathbb{P}^\alpha}{d\mathbb{P}^\star}(\mathbf{X}_{0:T}^\alpha)|\mathbf{X}_0^\alpha = \mathbf{x}_0\right] \qquad (140)$$

$$= D_{KL}(\mu_0|\mu_0^\star) + \mathbb{E}_{\mathbf{x}_0 \sim \mu_0}\mathbb{E}_{\mathbb{P}^\alpha}\left[\log\frac{d\mathbb{P}^\alpha}{d\mathbb{P}}(\mathbf{X}_{0:T}^\alpha) + \log\frac{d\mathbb{P}}{d\mathbb{P}^\star}(\mathbf{X}_{0:T}^\alpha)|\mathbf{X}_0^\alpha = \mathbf{x}_0\right], \quad (141)$$

where $(i)$ follows from the disintegration theorem (Léonard, 2013). For the second term of the RHS of (141), applying the Girsanov's theorem in B.4, we have:

$$\mathbb{E}_{\mathbb{P}^\alpha}^{0,\mathbf{x}_0}\left[\log\frac{d\mathbb{P}^\alpha}{d\mathbb{P}}(\mathbf{X}_{0:T}^\alpha)\right] = \mathbb{E}_{\mathbb{P}^\alpha}^{0,\mathbf{x}_0}\left[\int_0^T \alpha_t d\tilde{\mathbf{W}}_t + \int_0^T \frac{1}{2}\|\alpha_t\|^2 \, dt\right] = \mathbb{E}_{\mathbb{P}^\alpha}^{0,\mathbf{x}_0}\left[\int_0^T \frac{1}{2}\|\alpha_t\|^2 \, dt\right]. \tag{142}$$

Moreover, by the definition of $\mathbb{P}^\star$, we get:

$$\mathbb{E}_{\mathbb{P}^\alpha}^{0,\mathbf{x}_0}\left[\log\frac{d\mathbb{P}}{d\mathbb{P}^\star}(\mathbf{X}_{0:T}^\alpha)\right] = \mathbb{E}_{\mathbb{P}^\alpha}^{0,\mathbf{x}_0}\left[-\sum_{i=1}^k \log f_i(\mathbf{y}_{t_i}|\mathbf{X}_{t_i}^\alpha)\right]. \qquad (143)$$

Combining the results from (142) and (143), we obtain:

$$\mathbb{E}_{\mathbb{P}^\alpha}^{0,\mathbf{x}_0}\left[\log\frac{d\mathbb{P}^\alpha}{d\mathbb{P}}(\mathbf{X}_{0:T}^\alpha) + \log\frac{d\mathbb{P}}{d\mathbb{P}^\star}(\mathbf{X}_{0:T}^\alpha)\right] = \mathbb{E}_{\mathbb{P}^\alpha}^{0,\mathbf{x}_0}\left[\int_0^T \frac{1}{2}\|\alpha(t, \mathbf{X}_t^\alpha)\|^2 \, dt - \sum_{i=1}^k \log f_i(\mathbf{y}_{t_i}|\mathbf{X}_{t_i}^\alpha)\right] \tag{144}$$

$$= \mathbb{E}_{\mathbb{P}^\alpha}^{0,\mathbf{x}_0}\left[\int_0^T \frac{1}{2}\|\alpha(t, \mathbf{X}_t^\alpha)\|^2 \, dt - \sum_{i=1}^k \log f_i(\mathbf{y}_{t_i}|\mathbf{X}_{t_i}^\alpha)\right] \tag{145}$$

$$= \mathcal{J}(0, \mathbf{x}_0, \alpha). \tag{146}$$

Moreover, by the definition of $\{f_i\}_{i \in [1:k]}$, we get

$$\mathcal{J}(0, \mathbf{x}_0, \alpha) = \mathbb{E}_{\mathbb{P}^\alpha}\left[\int_0^T \frac{1}{2}\|\alpha(t, \mathbf{X}_t^\alpha)\|^2 \, dt - \sum_{i=1}^k \log f_i(\mathbf{y}_{t_i}|\mathbf{X}_{t_i})|\mathbf{X}_0^\alpha = \mathbf{x}_0\right] \qquad (147)$$

$$\stackrel{(i)}{=} \mathbb{E}_{\mathbb{P}^\alpha}\left[\int_0^T \frac{1}{2}\|\alpha(t, \mathbf{X}_t^\alpha)\|^2 \, dt - \sum_{i=1}^k \log g_i(\mathbf{y}_{t_i}|\mathbf{X}_{t_i})|\mathbf{X}_0^\alpha = \mathbf{x}_0\right] + \log \mathbf{Z}(\mathcal{H}_{t_k}), \qquad (148)$$

where $(i)$ follows from normalizing property in (44). Hence, it result the that:

$$D_{KL}(\mathbb{P}^\alpha|\mathbb{P}^\star) = D_{KL}(\mu_0|\mu_0^\star) + \mathbb{E}_{\mathbf{x}_0 \sim \mu_0}\left[\mathcal{J}(0, \mathbf{x}_0, \alpha)\right] \qquad (149)$$

$$= D_{KL}(\mu_0|\mu_0^\star) + \mathcal{L}(\alpha) + \log \mathbf{Z}(\mathcal{H}_{t_k}). \qquad (150)$$

For a KL-divergence term $D_{\mathrm{KL}}(\mu_0|\mu_0^\star)$, since $d\mu_0^\star(\mathbf{x}_0) = h_1(t_0, \mathbf{x}_0)d\mu_0(\mathbf{x}_0)$, we obtain

$$D_{\mathrm{KL}}(\mu_0|\mu_0^\star) = \mathbb{E}_{\mathbf{x}_0 \sim \mu_0}\left[\log \frac{d\mu_0}{d\mu_0^\star}(\mathbf{x}_0)\right] = \mathbb{E}_{\mathbf{x}_0 \sim \mu_0}\left[-\log h_1(0, \mathbf{x}_0)\right] \tag{151}$$

$$= \mathbb{E}_{\mathbf{x}_0 \sim \mu_0}\left[\mathcal{V}(0, \mathbf{x}_0)\right] \tag{152}$$

$$= \mathbb{E}_{\mathbf{x}_0 \sim \mu_0}\left[\mathcal{J}(0, \mathbf{x}_0, \alpha^\star)\right] \tag{153}$$

$$\overset{(i)}{=} \mathbb{E}_{\mathbf{x}_0 \sim \mu_0}\left[\tilde{\mathcal{J}}(0, \mathbf{x}_0, \alpha^\star)\right] + \log \mathbf{Z}(\mathcal{H}_{t_k}) \tag{154}$$

$$\overset{(ii)}{=} \mathcal{L}(\alpha^\star) + \log \mathbf{Z}(\mathcal{H}_{t_k}), \tag{155}$$

where $(i)$ follows from:

$$\mathcal{J}(0, \mathbf{x}_0, \alpha) = \mathbb{E}_{\mathbb{P}^\alpha}\left[\int_0^T \frac{1}{2}\|\alpha(s, \mathbf{X}_s^\alpha)\|^2 \, ds - \sum_i^k \log f_i(\mathbf{y}_{t_i}|\mathbf{X}_{t_i}^\alpha)|\mathbf{X}_0^\alpha = \mathbf{x}_0\right] \tag{156}$$

$$= \underbrace{\mathbb{E}_{\mathbb{P}^\alpha}\left[\int_0^T \frac{1}{2}\|\alpha(s, \mathbf{X}_s^\alpha)\|^2 \, ds - \sum_i^k \log g_i(\mathbf{y}_{t_i}|\mathbf{X}_{t_i}^\alpha)|\mathbf{X}_0^\alpha = \mathbf{x}_0\right]}_{\tilde{\mathcal{J}}(0, \mathbf{x}_0, \alpha)} + \log \mathbf{Z}(\mathcal{H}_{t_k}). $$

$$\tag{157}$$

Thus, we obtain $\alpha^\star = \arg\min_{\alpha \in \mathbb{A}} \mathcal{J}(0, \mathbf{x}_0, \alpha) = \arg\min_{\alpha \in \mathbb{A}} \tilde{\mathcal{J}}(0, \mathbf{x}_0, \alpha)$ as $\log \mathbf{Z}(\mathcal{H}_{t_k})$ is constant. It result that $\mathcal{J}(0, \mathbf{x}_0, \alpha^\star) = \tilde{\mathcal{J}}(0, \mathbf{x}_0, \alpha^\star) + \log \mathbf{Z}(\mathcal{H}_{t_k})$. Additionally, $(ii)$ follows from

$$\mathbb{E}_{\mathbf{x}_0 \sim \mu_0}\left[\min_{\alpha \in \mathbb{A}} \tilde{\mathcal{J}}(0, \mathbf{x}_0, \alpha)\right] = \min_{\alpha \in \mathbb{A}} \mathbb{E}_{\mathbf{x}_0 \sim \mu_0}\left[\tilde{\mathcal{J}}(0, \mathbf{x}_0, \alpha)\right] = \min_{\alpha \in \mathbb{A}} \mathcal{L}(\alpha) = \mathcal{L}(\alpha^\star) \tag{158}$$

since the minimization is independent of the initial condition, as implied by the disintegration theorem (Léonard, 2013). Hence, since the Donsker–Varadhan variational principle in Lemma B.6 with $\mathcal{W}(\mathbf{X}_{0:T}^\alpha) = -\sum_{i=1}^k \log g_i(\mathbf{y}_{t_i}|\mathbf{X}_{t_i}^\alpha)$ yields

$$\mathbb{E}_{\mathbb{P}^\alpha}\left[\int_0^T \frac{1}{2}\|\alpha(t, \mathbf{X}_t^\alpha)\|^2 \, dt - \sum_{i=1}^k \log g_i(\mathbf{y}_{t_i}|\mathbf{X}_{t_i}^\alpha)\right] + \log \mathbb{E}_{\mathbb{P}}\left[\prod_{i=1}^k g_i(\mathbf{y}_{t_i}|\mathbf{X}_{t_i}^\alpha)\right] \geq 0, \tag{159}$$

it implies that $D_{\mathrm{KL}}(\mu_0|\mu_0^\star) = \mathcal{L}(\alpha^\star) + \log \mathbf{Z}(\mathcal{H}_{t_k}) = 0$ for the optimal control $\alpha^\star$. In other words, $\mu_0^\star = \mu_0$ almost everywhere with respect to $\mu_0$ and from the variational bound

$$D_{\mathrm{KL}}(\mathbb{P}^\alpha|\mathbb{P}^\star) = \mathcal{L}(\alpha) + \log \mathbf{Z}(\mathcal{H}_{t_k}) \geq 0, \tag{160}$$

the equality holds if and only if $\alpha \to \alpha^\star$. It concludes the proof. $\qquad \square$

## B.7 DERIVATION OF AMORTIZED ELBO IN (28).

Let $\mathbf{o}_{0:T}$ is given time-series data. Then, for an auxiliary variable $\mathbf{y}_{0:T} \sim q_\phi(\mathbf{y}_{0:T}|\mathbf{o}_{0:T})$, the ELBO is given as

$$\log p_\psi(\mathbf{o}_{0:T}) \geq \mathbb{E}_{q_\phi(\mathbf{y}_{0:T}|\mathbf{o}_{0:T})}\left[\log \frac{\prod_{i=1}^K p_\psi(\mathbf{o}_{t_i}|\mathbf{y}_{t_i})g(\mathbf{y}_{0:T})}{\prod_{i=1}^K q_\phi(\mathbf{y}_{t_i}|\mathbf{o}_{t_i})}\right] \tag{161}$$

$$= \mathbb{E}_{q_\phi(\mathbf{y}_{0:T}|\mathbf{o}_{0:T})}\left[\sum_{i=1}^K \log p_\psi(\mathbf{o}_{t_i}|\mathbf{y}_{t_i}) + \log g(\mathbf{y}_{0:T}) - \log \prod_{i=1}^K q_\phi(\mathbf{y}_{t_i}|\mathbf{o}_{t_i})\right] \tag{162}$$

$$\geq \mathbb{E}_{q_\phi(\mathbf{y}_{0:T}|\mathbf{o}_{0:T})}\left[\sum_{i=1}^K \log p_\psi(\mathbf{o}_{t_i}|\mathbf{y}_{t_i}) - \mathcal{L}(\alpha) - \log \prod_{i=1}^K q_\phi(\mathbf{y}_{t_i}|\mathbf{o}_{t_i})\right] \tag{163}$$

$$= \mathbb{E}_{q_\phi(\mathbf{y}_{0:T}|\mathbf{o}_{0:T})}\left[\sum_{i=1}^K \log p_\psi(\mathbf{o}_{t_i}|\mathbf{y}_{t_i}) - \mathcal{L}(\alpha) - \sum_{i=1}^K \log q_\phi(\mathbf{y}_{t_i}|\mathbf{o}_{t_i})\right] \tag{164}$$

$$\overset{(i)}{\geq} \mathbb{E}_{q_\phi(\mathbf{y}_{0:T}|\mathbf{o}_{0:T})}\left[\sum_{i=1}^K \log p_\psi(\mathbf{o}_{t_i}|\mathbf{y}_{t_i}) - \mathcal{L}(\alpha)\right], \tag{165}$$

where $(i)$ follows from $\mathbb{E}_{q_\phi(\mathbf{y}_{0:T}|\mathbf{o}_{0:T})}\left[-\sum_{i=1}^K \log q_\phi(\mathbf{y}_{t_i}|\mathbf{o}_{t_i})\right] = C \geq 0$ since $q_\phi$ is Gaussian distribution with constant covariance matrix.

## B.8    Proof of Theorem 3.8

**Theorem 3.8** (Simulation-free estimation). *Let us consider sequences of SPD matrices $\{\mathbf{A}_i\}_{i\in[1:k]}$ that admit the eigen-decomposition $\mathbf{A}_i = \mathbf{E}\mathbf{D}_i\mathbf{E}^\top$ with $\mathbf{E} \in \mathbb{R}^{d\times d}$ and $\mathbf{D}_i \in diag(\mathbb{R}^d) \succeq 0$ for all $i \in [1:k]$, control vectors $\{\alpha_i\}_{i\in[1:k]}$ and following control-affine SDEs for all $i \in [1:k]$:*

$$d\mathbf{X}_t = [-\mathbf{A}_i\mathbf{X}_t + \alpha_i]\,dt + \sigma d\mathbf{W}_t, \quad t \in [t_{i-1}, t_i). \tag{166}$$

*Then, with $\mathbf{X}_0 \sim \mathcal{N}(\mathbf{m}_0, \boldsymbol{\Sigma}_0)$, the solution of (166) is a Gaussian process $\mathcal{N}(\mathbf{m}_{t_i}, \boldsymbol{\Sigma}_{t_i})$ with:*

$$\mathbf{m}_{t_i} = \mathbf{E}\left(e^{-\sum_{j=1}^i (t_j - t_{j-1})\mathbf{D}_j}\hat{\mathbf{m}}_{t_0} - \sum_{k=1}^i e^{-\sum_{j=k}^i (t_j - t_{j-1})\mathbf{D}_j}\mathbf{D}_k^{-1}\left(\mathbf{I} - e^{(t_k - t_{k-1})\mathbf{D}_k}\right)\hat{\alpha}_k\right),$$

$$\boldsymbol{\Sigma}_{t_i} = \mathbf{E}\left(e^{-2\sum_{j=1}^i (t_j - t_{j-1})\mathbf{D}_j}\hat{\boldsymbol{\Sigma}}_{t_0} - \frac{1}{2}\sum_{k=1}^i e^{-2\sum_{j=k}^i (t_j - t_{j-1})\mathbf{D}_j}\mathbf{D}_k^{-1}\left(\mathbf{I} - e^{2(t_k - t_{k-1})\mathbf{D}_k}\right)\right)\mathbf{E}^\top,$$

*where $\hat{\mathbf{m}}_{t_i} = \mathbf{E}^\top\mathbf{m}_{t_i}, \hat{\boldsymbol{\Sigma}}_{t_i} = \mathbf{E}^\top\boldsymbol{\Sigma}_{t_i}\mathbf{E}$ and $\hat{\alpha}_i = \mathbf{E}^\top\alpha_i$ for all $i \in [1:k]$.*

*Proof.* Note that since $\{\mathbf{A}_i\}_{i\in[1:k]}$ are SPD matrices, we can apply the transformation outlined in Remark 3.7 and express the original dynamics (166) in the projected form using the eigenbasis $\mathbf{E}$. Then, for any $t \in [t_i, t_{i+1})$, the solution to (166) at time $t$ is given as

$$\hat{\mathbf{X}}_t = e^{-\Delta_i(t)\mathbf{D}_i}\left(\hat{\mathbf{X}}_{t_i} + \int_{t_i}^t e^{\Delta_i(s)\mathbf{D}_i}\hat{\alpha}_i ds + \int_{t_i}^t e^{\Delta_i(s)\mathbf{D}_i}d\hat{\mathbf{W}}_s\right), \Delta_i(t) = \begin{cases} t - t_i, \text{ for } t > t_i \\ 0, \text{ for } t \leq t_i, \end{cases} \tag{167}$$

where $\hat{\mathbf{m}}_{t_i} = \mathbf{E}^\top\mathbf{m}_{t_i}, \hat{\boldsymbol{\Sigma}}_{t_i} = \mathbf{E}^\top\boldsymbol{\Sigma}_{t_i}\mathbf{E}$ and $\hat{\alpha}_i = \mathbf{E}^\top\alpha_i$ for all $i \in [1:k]$ and $\hat{\mathbf{W}}_t = \mathbf{E}^\top\mathbf{W}_t$. Given that we have defined $\mathbf{X}_0 \sim \mathcal{N}(\mathbf{m}_0, \boldsymbol{\Sigma}_0), \hat{\mathbf{X}}_{t_i} = \mathbf{E}^\top\mathbf{X}_{t_i}$ is a Gaussian process for any $i \in [1:k]$. The first two moments of Gaussian process can be computed from (167). First, since $\mathbf{D}_i$ is diagonal, the integral can be computed as $\int_{t_i}^t e^{\Delta_i(s)\mathbf{D}_i}ds = -\mathbf{D}_i^{-1}(\mathbf{I} - e^{\Delta_i(t)\mathbf{D}_i})$ and $\mathbf{M}_i(t) := \int_{t_i}^t e^{\Delta_i(s)\mathbf{D}_i}d\hat{\mathbf{W}}_s$ is a martingale process with respect to $\mathbb{P}^\alpha$ *i.e.*, $\mathbb{E}_{\mathbb{P}^\alpha}[\mathbf{M}_i(t)] = 0$. Hence, since $\hat{\alpha}_i$ is time-invariant vector, the mean $\mathbb{E}_{\mathbb{P}}[\hat{\mathbf{X}}_t] = \hat{\mathbf{m}}_t$ for $t \in [t_i, t_{i+1})$ can be computed as

$$\hat{\mathbf{m}}_t = e^{-\Delta_i(t)\mathbf{D}_i}\hat{\mathbf{m}}_{t_i} - e^{-\Delta_i(t)\mathbf{D}_i}\mathbf{D}_i^{-1}(\mathbf{I} - e^{\Delta_i(t)\mathbf{D}_i})\hat{\alpha}_i. \tag{168}$$

Secondly, for a covariance $\mathbb{E}_{\mathbb{P}}[(\hat{\mathbf{X}}_t - \hat{\mathbf{m}}_t)(\hat{\mathbf{X}}_t - \hat{\mathbf{m}}_t)^\top] = \hat{\boldsymbol{\Sigma}}_t$, we can compute

$$\hat{\boldsymbol{\Sigma}}_t = \mathbb{E}_{\mathbb{P}^\alpha}\left[e^{-2\Delta_i(t)\mathbf{D}_i}\left(\hat{\mathbf{X}}_{t_i} - \hat{\mathbf{m}}_{t_i} + \mathbf{M}_i(t)\right)\left(\hat{\mathbf{X}}_{t_i} - \hat{\mathbf{m}}_{t_i} + \mathbf{M}_i(t)\right)^\top\right] \tag{169}$$

$$\overset{(i)}{=} e^{-2\Delta_i(t)\mathbf{D}_i}\mathbb{E}_{\mathbb{P}^\alpha}\left[(\hat{\mathbf{X}}_{t_i} - \hat{\mathbf{m}}_{t_i})(\hat{\mathbf{X}}_{t_i} - \hat{\mathbf{m}}_{t_i})^\top + \|\mathbf{M}_i(t)\|_2^2\right] \tag{170}$$

$$\overset{(ii)}{=} e^{-2\Delta_i(t)\mathbf{D}_i}\hat{\boldsymbol{\Sigma}}_{t_i} - \frac{1}{2}e^{-2\Delta_i(t)\mathbf{D}_i}\mathbf{D}_i^{-1}(\mathbf{I} - e^{2\Delta_i(t)\mathbf{D}_i}), \tag{171}$$

where $(i)$ follows from the fact that $\mathbf{M}_i(t)$ is a martingale and we use Itô isometry in $(ii)$:

$$\mathbb{E}_{\mathbb{P}^\alpha}\left[\|\mathbf{M}_i(t)\|_2^2\right] = \mathbb{E}_{\mathbb{P}^\alpha}\left[\int_{t_i}^t \left\|e^{\Delta_i(s)\mathbf{D}_i}\right\|_2^2 ds\right] = -\frac{1}{2}e^{-2\Delta_i(t)\mathbf{D}_i}\mathbf{D}_i^{-1}(\mathbf{I} - e^{2\Delta_i(t)\mathbf{D}_i}). \tag{172}$$

Hence, we get the Gaussian law of $\hat{\mathbf{X}}_t$ at time $t \in [t_i, t_{i+1}), \mathcal{N}(\hat{\mathbf{m}}_t, \hat{\boldsymbol{\Sigma}}_t)$. Furthermore, given recurrence forms of mean (168) and covariance (171), the first two moments of Gaussian distribution

for each time steps $t_i$ can be computed sequentially. For a mean $\hat{\mathbf{m}}_{t_i}$ we have,

$$\hat{\mathbf{m}}_{t_1} = e^{-\Delta_0(t_1)\mathbf{D}_1}\hat{\mathbf{m}}_{t_0} - e^{-\Delta_0(t_1)\mathbf{D}}\mathbf{D}_1^{-1}(\mathbf{I} - e^{\Delta_0(t_1)\mathbf{D}_1})\hat{\alpha}_1 \tag{173}$$

$$\hat{\mathbf{m}}_{t_2} = e^{-\sum_{j=1}^{2}\Delta_{j-1}(t_j)\mathbf{D}_j}\hat{\mathbf{m}}_{t_0} \tag{174}$$

$$- e^{-\sum_{j=1}^{2}\Delta_{j-1}(t_j)\mathbf{D}_j}\mathbf{D}_1^{-1}(\mathbf{I} - e^{\Delta_0(t_1)\mathbf{D}_1})\hat{\alpha}_1 - e^{-\Delta_1(t_2)\mathbf{D}_2}\mathbf{D}_2^{-1}(\mathbf{I} - e^{\Delta_1(t_2)\mathbf{D}_2})\hat{\alpha}_2 \tag{175}$$

$$\vdots \tag{176}$$

$$\hat{\mathbf{m}}_{t_i} = e^{-\sum_{j=1}^{i}\Delta_{j-1}(t_j)\mathbf{D}_j}\hat{\mathbf{m}}_{t_0} - \sum_{k=1}^{i} e^{-\sum_{j=k}^{i}\Delta_{j-1}(t_j)\mathbf{D}_j}\mathbf{D}_k^{-1}\left(\mathbf{I} - e^{\Delta_{k-1}(t_k)\mathbf{D}_k}\right)\hat{\alpha}_k \tag{177}$$

Moreover, for a covariance $\hat{\boldsymbol{\Sigma}}_{t_i}$, similar calculation yields

$$\hat{\boldsymbol{\Sigma}}_{t_1} = e^{-2\Delta_0(t_1)\mathbf{D}_1}\hat{\boldsymbol{\Sigma}}_{t_0} - \frac{1}{2}e^{-2\Delta_0(t_1)\mathbf{D}}\mathbf{D}_1^{-1}(\mathbf{I} - e^{2\Delta_0(t_1)\mathbf{D}_1}) \tag{178}$$

$$\hat{\boldsymbol{\Sigma}}_{t_2} = e^{-2\sum_{j=1}^{2}\Delta_{j-1}(t_j)\mathbf{D}_j}\hat{\boldsymbol{\Sigma}}_{t_0} \tag{179}$$

$$- \frac{1}{2}e^{-2\sum_{j=1}^{2}\Delta_{j-1}(t_j)\mathbf{D}_j}\mathbf{D}_1^{-1}(\mathbf{I} - e^{2\Delta_0(t_1)\mathbf{D}_1}) - \frac{1}{2}e^{-2\Delta_1(t_2)\mathbf{D}_2}\mathbf{D}_2^{-1}(\mathbf{I} - e^{2\Delta_1(t_2)\mathbf{D}_2}) \tag{180}$$

$$\vdots \tag{181}$$

$$\hat{\boldsymbol{\Sigma}}_{t_i} = e^{-2\sum_{j=1}^{i}\Delta_{j-1}(t_j)\mathbf{D}_j}\hat{\boldsymbol{\Sigma}}_{t_0} - \frac{1}{2}\sum_{k=1}^{i} e^{-2\sum_{j=k}^{i}\Delta_{j-1}(t_j)\mathbf{D}_j}\mathbf{D}_k^{-1}\left(\mathbf{I} - e^{2\Delta_{k-1}(t_k)\mathbf{D}_k}\right) \tag{182}$$

Now, since $\mathbf{D}_i = \mathbf{E}\mathbf{A}_i\mathbf{E}^\top$ and the orthonormality of $\mathbf{E}$, we can express the mean and covariance in the original canonical basis. For the mean, we get

$$\mathbf{E}\hat{\mathbf{m}}_{t_i} = \mathbf{E}\left(e^{-\sum_{j=1}^{i}\Delta_{j-1}(t_j)\mathbf{D}_j}\hat{\mathbf{m}}_{t_0} - \sum_{k=1}^{i} e^{-\sum_{j=k}^{i}\Delta_{j-1}(t_j)\mathbf{D}_j}\mathbf{D}_k^{-1}\left(\mathbf{I} - e^{\Delta_{k-1}(t_k)\mathbf{D}_k}\right)\hat{\alpha}_k\right) \tag{183}$$

$$\overset{(i)}{=} \mathbf{E}\left(e^{-\sum_{j=1}^{i}\Delta_{j-1}(t_j)\mathbf{D}_j}\mathbf{E}^\top\mathbf{m}_{t_0} - \sum_{k=1}^{i} e^{-\sum_{j=k}^{i}\Delta_{j-1}(t_j)\mathbf{D}_j}\mathbf{E}^\top\mathbf{A}_k^{-1}\mathbf{E}\left(\mathbf{I} - e^{\Delta_{k-1}(t_k)\mathbf{D}_k}\right)\mathbf{E}^\top\alpha_k\right) \tag{184}$$

$$\overset{(ii)}{=} e^{-\sum_{j=1}^{i}\Delta_{j-1}(t_j)\mathbf{A}_j}\mathbf{m}_{t_0} - \sum_{k=1}^{i} e^{-\sum_{j=k}^{i}\Delta_{j-1}(t_j)\mathbf{A}_j}\mathbf{A}_k^{-1}\left(\mathbf{I} - e^{\Delta_{k-1}(t_k)\mathbf{A}_k}\right)\alpha_k \tag{185}$$

$$= \mathbf{m}_{t_i}, \tag{186}$$

where $(i)$ follows from $\hat{\mathbf{m}}_{t_0} = \mathbf{E}^\top\mathbf{m}_{t_0}$, and $\hat{\alpha}_i = \mathbf{E}^\top\alpha_i$ for all $i \in [1:k]$ and $(ii)$ follows from $\mathbf{D}_i^{-1} = \mathbf{E}^\top\mathbf{A}_i^{-1}\mathbf{E}$ and $e^{-\mathbf{D}_i^{-1}} = \mathbf{E}^\top e^{-\mathbf{A}_i^{-1}}\mathbf{E}$. Similarly, for the covariance, we get

$$\mathbf{E}\hat{\boldsymbol{\Sigma}}_{t_i}\mathbf{E}^\top = \mathbf{E}\left(e^{-2\sum_{j=1}^{i}\Delta_{j-1}(t_j)\mathbf{D}_j}\hat{\boldsymbol{\Sigma}}_{t_0} - \frac{1}{2}\sum_{k=1}^{i} e^{-2\sum_{j=k}^{i}\Delta_{j-1}(t_j)\mathbf{D}_j}\mathbf{D}_k^{-1}\left(\mathbf{I} - e^{2\Delta_{k-1}(t_k)\mathbf{D}_k}\right)\right)\mathbf{E}^\top \tag{187}$$

$$\overset{(i)}{=} \mathbf{E}\left(e^{-2\sum_{j=1}^{i}\Delta_{j-1}(t_j)\mathbf{D}_j}\mathbf{E}^\top\boldsymbol{\Sigma}_{t_0}\mathbf{E} - \frac{1}{2}\sum_{k=1}^{i} e^{-2\sum_{j=k}^{i}\Delta_{j-1}(t_j)\mathbf{D}_j}\mathbf{E}^\top\mathbf{A}_k^{-1}\mathbf{E}\left(\mathbf{I} - e^{2\Delta_{k-1}(t_k)\mathbf{D}_k}\right)\right)\mathbf{E}^\top \tag{188}$$

$$= e^{-2\sum_{j=1}^{i}\Delta_{j-1}(t_j)\mathbf{A}_j}\boldsymbol{\Sigma}_{t_0} - \frac{1}{2}\sum_{k=1}^{i} e^{-2\sum_{j=k}^{i}\Delta_{j-1}(t_j)\mathbf{A}_j}\mathbf{A}_k^{-1}\left(\mathbf{I} - e^{2\Delta_{k-1}(t_k)\mathbf{A}_k}\right) \tag{189}$$

$$= \boldsymbol{\Sigma}_{t_i}, \tag{190}$$

where $(i)$ follows from $\hat{\boldsymbol{\Sigma}}_{t_0} = \mathbf{E}^\top\boldsymbol{\Sigma}_{t_0}\mathbf{E}$. By applying the procedure from (168-171) to the original SDE (166), we recover the mean and covariance expressions in (185) and (189). This shows that the transformed mean and covariance in (177) and (182) can indeed be projected back into the original basis, completing the proof. $\square$

---

**Algorithm 1** Parallel Scan for Mean and Covariance

---

1: **Input.** Given time stamps $\mathcal{T} = \{t_1, t_2, \ldots, t_K\}$, initial mean $\mathbf{m}_{t_0}$ and covariance $\boldsymbol{\Sigma}_{t_0}$, control policies $\{\hat{\alpha}_i\}_{i \in [1:k]}$, matrices $\{\mathbf{D}\}_{i \in [1:k]}$.
2: Compute $\{\Delta_i(t_i), \hat{\mathbf{D}}_i, \hat{\mathbf{C}}_i, \bar{\mathbf{D}}_i, \bar{\mathbf{C}}_i\}_{i \in [1:k]}$
3: Set $\{\mathbf{M}_i\}_{i \in [1:k]} = \{(\hat{\mathbf{D}}_i, \hat{\mathbf{C}}_i \hat{\alpha}_i)\}_{i \in [1:k]}$ and $\{\mathbf{S}_i\}_{i \in [1:k]} = \{(\bar{\mathbf{D}}_i, \bar{\mathbf{C}}_i \mathbf{1})\}_{i \in [1:k]}$.
4: Parallel Scan $\{\mathbf{M}'_i, \mathbf{S}'_i\}_{i \in [1:k]} = \texttt{ParallelScan}(\{\mathbf{M}_i, \mathbf{S}_i\}_{i \in [1:k]}, \otimes)$
5: $\Rightarrow$ Algorithm 2 for $\texttt{ParallelScan}$
6: **for** $i = 1$ to $K$ **do in parallel**
7: $\quad \mathbf{m}_{t_i} = \mathbf{M}'^{(1)}_i \mathbf{m}_{t_0} + \mathbf{M}'^{(2)}_i$
8: $\quad \boldsymbol{\Sigma}_{t_i} = \mathbf{S}'^{(1)}_i \boldsymbol{\Sigma}_{t_0} + \mathbf{S}'^{(2)}_i$
9: **end for**
10: **Return** $\mathbf{m}_{t \in \mathcal{T}}, \boldsymbol{\Sigma}_{t \in \mathcal{T}}$

---

## C PARALLEL SCAN

Given an associative operator $\otimes$ and a sequence of elements $[s_{t_1}, \cdots s_{t_K}]$, the parallel scan algorithm (Blelloch, 1990) computes the all-prefix-sum which returns the sequence $[s_{t_1}, (s_{t_1} \otimes s_{t_2}), \cdots, (s_{t_1} \otimes s_{t_2} \otimes \cdots \otimes s_{t_K})]$ in $\mathcal{O}(\log K)$ time. Since we have verified that moments $\{\hat{\mathbf{m}}_{t_i}, \hat{\boldsymbol{\Sigma}}_{t_i}\}_{i \in [1:k]}$ of the controlled distributions can be estimated by the recurrences in (168,171):

$$\hat{\mathbf{m}}_{t_i} = \hat{\mathbf{D}}_i \hat{\mathbf{m}}_{t_{i-1}} + \hat{\mathbf{C}}_i \hat{\alpha}_i \tag{191}$$

$$\hat{\boldsymbol{\Sigma}}_{t_i} = \bar{\mathbf{D}}_i \hat{\boldsymbol{\Sigma}}_{t_{i-1}} + \bar{\mathbf{C}}_i \mathbf{1}, \tag{192}$$

where, we define

$$\hat{\mathbf{D}}_i = e^{-\Delta_{i-1}(t_i)\mathbf{D}_i}, \quad \hat{\mathbf{C}}_i = -e^{-\Delta_{i-1}(t_i)\mathbf{D}_i} \mathbf{D}_i^{-1} (\mathbf{I} - e^{\Delta_{i-1}(t_i)\mathbf{D}_i}) \tag{193}$$

$$\bar{\mathbf{D}}_i = e^{-2\Delta_{i-1}(t_i)\mathbf{D}_i}, \quad \bar{\mathbf{C}}_i = -\frac{1}{2} e^{-2\Delta_{i-1}(t_i)\mathbf{D}_i} \mathbf{D}_i^{-1} (\mathbf{I} - e^{2\Delta_{i-1}(t_i)\mathbf{D}_i}), \tag{194}$$

and $\mathbf{1} = (1, \cdots, 1) \in \mathbb{R}^d$. For a parallel scan, we will define the sequence of tuple $\{\mathbf{M}_i\}_{i \in [1:k]}$, such that each element is $\mathbf{M}_i = (\hat{\mathbf{D}}_i, \hat{\mathbf{C}}_i \alpha_i)$ and $\{\mathbf{S}_i\}_{i \in [1:k]}$, such that each element is $\mathbf{S}_i = (\bar{\mathbf{D}}_i, \bar{\mathbf{C}}_i \mathbf{1})$ for $\{\mathbf{m}_{t_i}\}_{i \in [1:k]}$ and $\{\boldsymbol{\Sigma}_{t_i}\}_{i \in [1:k]}$, respectively.

Now, let us define a binary operator $\otimes$:

$$\mathbf{M}_s \otimes \mathbf{M}_t = (\hat{\mathbf{D}}_t \circ \hat{\mathbf{D}}_s, \hat{\mathbf{D}}_t \circ \hat{\mathbf{C}}_s \hat{\alpha}_s + \hat{\mathbf{C}}_t \hat{\alpha}_t) \tag{195}$$

$$\mathbf{S}_s \otimes \mathbf{S}_t = (\bar{\mathbf{D}}_t \circ \bar{\mathbf{D}}_s, \bar{\mathbf{D}}_t \circ \bar{\mathbf{C}}_s \mathbf{1} + \bar{\mathbf{C}}_t \mathbf{1}) \tag{196}$$

We can verify that $\otimes$ is associative operator since it satisfying:

$$(\mathbf{M}_s \otimes \mathbf{M}_t) \otimes \mathbf{M}_u = (\hat{\mathbf{D}}_t \circ \hat{\mathbf{D}}_s, \hat{\mathbf{D}}_t \circ \hat{\mathbf{C}}_s \hat{\alpha}_s + \hat{\mathbf{C}}_t \hat{\alpha}_t) \otimes (\hat{\mathbf{D}}_u, \hat{\mathbf{C}}_u \hat{\alpha}_u) \tag{197}$$

$$= \left( \hat{\mathbf{D}}_u \circ \left( \hat{\mathbf{D}}_t \circ \hat{\mathbf{D}}_s \right), \hat{\mathbf{D}}_u \circ \left( \hat{\mathbf{D}}_t \circ \hat{\mathbf{C}}_s \hat{\alpha}_s + \hat{\mathbf{C}}_t \hat{\alpha}_t \right) + \hat{\mathbf{C}}_u \hat{\alpha}_u \right) \tag{198}$$

$$= \left( \left( \hat{\mathbf{D}}_u \circ \hat{\mathbf{D}}_t \right) \circ \hat{\mathbf{D}}_s, \left( \hat{\mathbf{D}}_u \circ \hat{\mathbf{D}}_t \right) \circ \hat{\mathbf{C}}_s \hat{\alpha}_s + \hat{\mathbf{D}}_u \circ \hat{\mathbf{C}}_t \hat{\alpha}_t + \hat{\mathbf{C}}_u \hat{\alpha}_u \right) \tag{199}$$

$$= \mathbf{M}_s \otimes (\mathbf{M}_t \otimes \mathbf{M}_u) \tag{200}$$

Thus we get $(\mathbf{M}_s \otimes \mathbf{M}_t) \otimes \mathbf{M}_u = \mathbf{M}_s \otimes (\mathbf{M}_t \otimes \mathbf{M}_u)$. Moreover, we can get similar results for $\mathbf{S}_i$:

$$(\mathbf{S}_s \otimes \mathbf{S}_t) \otimes \mathbf{S}_u = (\bar{\mathbf{D}}_t \circ \bar{\mathbf{D}}_s, \bar{\mathbf{D}}_t \circ \bar{\mathbf{C}}_s \mathbf{1} + \bar{\mathbf{C}}_t \mathbf{1}) \otimes (\bar{\mathbf{D}}_u, \bar{\mathbf{C}}_u \mathbf{1}) \tag{201}$$

$$= \left( \bar{\mathbf{D}}_u \circ \left( \bar{\mathbf{D}}_t \circ \bar{\mathbf{D}}_s \right), \bar{\mathbf{D}}_u \circ \left( \bar{\mathbf{D}}_t \circ \bar{\mathbf{C}}_s \mathbf{1} + \bar{\mathbf{C}}_t \mathbf{1} \right) + \bar{\mathbf{C}}_u \mathbf{1} \right) \tag{202}$$

$$= \left( \left( \bar{\mathbf{D}}_u \circ \bar{\mathbf{D}}_t \right) \circ \bar{\mathbf{D}}_s, \left( \bar{\mathbf{D}}_u \circ \bar{\mathbf{D}}_t \right) \circ \bar{\mathbf{C}}_s \mathbf{1} + \bar{\mathbf{D}}_u \circ \bar{\mathbf{C}}_t \mathbf{1} + \bar{\mathbf{C}}_u \mathbf{1} \right) \tag{203}$$

$$= \mathbf{S}_s \otimes (\mathbf{S}_t \otimes \mathbf{S}_u) \tag{204}$$

Now, both means and covariances along the interval $\{\mathbf{m}_{t_i}\}_{i \in [1:k]}$ and $\{\boldsymbol{\Sigma}_{t_i}\}_{i \in [1:k]}$ can be computed (parallel in time $K$) using a parallel scan algorithm described in Algorithm 1.

---

**Algorithm 2** `ParallelScan`

---

1: **Input.**  Sequence of tuples $\{\mathbf{F}_1, \mathbf{F}_2, \ldots, \mathbf{F}_K\}$, associative operator $\otimes$.
2: `Up-Sweep Stage.`
3: **for** $d = 0$ to $\lceil \log_2 K \rceil - 1$ **do**
4:    **for** each sub-tree of height $d$ **do in parallel**
5:      Let $i = 2^{d+1}k + 2^{d+1} - 1$ for $k = 0, 1, \ldots$
6:      **if** $i < K$
7:       $\mathbf{F}_i = \mathbf{F}_{i-2^d} \otimes \mathbf{F}_i$
8:      **end if**
9:    **end for**
10: **end for**
11: `Down-Sweep Stage.`
12: $\mathbf{F}_K = \mathbf{I}$, where $\mathbf{I}$ is the identity element for $\otimes$.
13: **for** $d = \lceil \log_2 K \rceil - 1$ to $0$ **do**
14:    **for** each sub-tree of height $d$ **do in parallel**
15:      Let $i = 2^{d+1}k + 2^{d+1} - 1$ for $k = 0, 1, \ldots$
16:      **if** $i < K$
17:       $\mathbf{F}_{i-2^d} = \mathbf{F}_{i-2^d} \otimes \mathbf{F}_i$
18:       $\mathbf{F}_i = \mathbf{F}_{i-2^d}$
19:      **end iF**
20:    **end for**
21: **end for**
22: **for** $i = 1$ to $K$ **do in parallel**
23:    $\mathbf{F}'_i = \mathbf{F}_1 \otimes \mathbf{F}_2 \otimes \cdots \otimes \mathbf{F}_i$
24: **end for**
25: **Return** Scanned sequence $\{\mathbf{F}'_1, \mathbf{F}'_2, \ldots, \mathbf{F}'_K\}$.

---

## D   IMPLEMENTATION DETAILS

### D.1   DATASETS

**Human Activity**   The Human Activity dataset[6] consists of time series data collected from five individuals performing different activities. Following the preprocessing steps described in (Rubanova et al., 2019), we obtained $6,554$ sequences, each with $211$ time points and a fixed sequence length of $50$ irregularly sampled time stamps. The time range was rescaled to $[0, 1]$. The classification task involves assigning each time point to one of seven categories: "walking", "falling", "lying", "sitting", "standing up", "on all fours", or "sitting on the ground". The dataset was split into 4,194 sequences for training, 1,049 for validation, and 1,311 for testing.

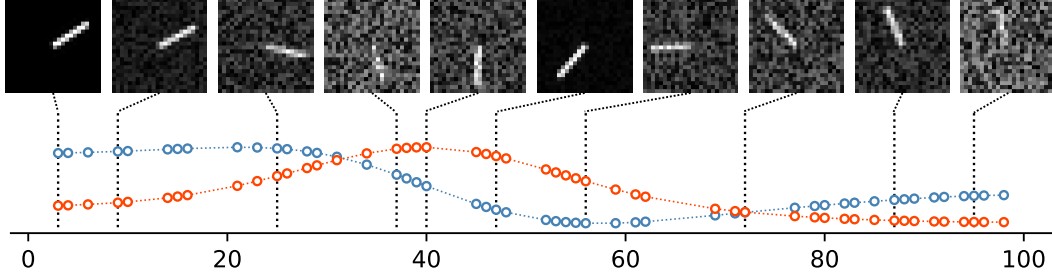

Figure 3: **Example of the pendulum sequence**. (**Up**) The input image sequences $\{\mathbf{o}\}_{t \in \mathcal{T}}$ observed at irregular time stamps. (**Down**) The angular values of $\sin(\theta_t)$ and $\cos(\theta_t)$ where $\theta_t$ represents the angle of the pendulum at time $t \in [0, 100]$, are used as regression targets.

---

[6]https://doi.org/10.24432/C57G8X under CC BY 4.0

**Pendulum** The pendulum images were algorithmically generated through numerical simulation as outlined in (Becker et al., 2019). We followed the setup described in (Schirmer et al., 2022), where 4,000 image sequences were generated. Each sequence consists of 50 time stamps, irregularly sampled from $T = 100$, with each image being a $24 \times 24$ pixel representation. The sequences was further corrupted by a correlated noise process, as detailed in (Becker et al., 2019). For our experiments, we used 2,000 sequences for training and 1,000 sequences for validation and testing.

**USHCN** The USHCN dataset (Menne et al., 2015)[7] includes daily measurements from $1,218$ weather stations across the US, covering five variables: precipitation, snowfall, snow depth, and minimum and maximum temperature. We follow the pre-processing steps outlined in (De Brouwer et al., 2019), but select a subset of $1,168$ stations over a four-year period starting from 1990, consistent with (Schirmer et al., 2022). Moreover, we make the time series irregular by subsampling $50\%$ of the time points and randomly removing $20\%$ of the measurements. We normalize the features to lie within the range $[0, 1]$ and split into $60\%$ for training, $20\%$ for validation, and $20\%$ for testing.

**Physionet** The Physionet dataset (Silva et al., 2012)[8] contains 8000 multivariate clinical time-series obtained from the intensive care unit (ICU). Each time-series includes various clinical features recorded during the first 48 hours after the patient's admission to the ICU. We preprocess the data as in (Rubanova et al., 2019). Although the dataset contains a total of 41 measurements, we eliminate 4 static features, *i.e.*, age, gender, height, and ICU-type, leaving 37 time-varying features. We round the time-steps to 6-minute intervals, following (Schirmer et al., 2022). We normalize the features to lie within the range $[0, 1]$ and split into $60\%$ for training, $20\%$ for validation, and $20\%$ for testing.

Table 4: Training Hyper-parameters

| Dataset | Learning Rate | Train Epoch | Time Scale | Batch Size | $\mathbb{R}^d$ | # of base matrices (L) | # of parameters |
|---|---|---|---|---|---|---|---|
| Human Activity | $1 \times 10^{-3}$ | 400 | 1/221 | 256 | 288 | 256 | 1.65M |
| Pendulum | $1 \times 10^{-3}$ | 500 | 0.1 | 50 | 20 | 15 | 19.6K |
| USHCN | $1 \times 10^{-3}$ | 500 | 0.2 | 50 | 20 | 20 | 18.5K |
| Physionet | $1 \times 10^{-3}$ | 500 | 0.3 | 100 | 24 | 20 | 28.5K |

### D.2 MASKING SCHEME

Figure D.2 provides a detailed illustration of the masked attention mechanism described in the assimilation schemes introduced in Section 3.3.

### D.3 TRAINING DETAILS

**Training** For all experiments, except for human activity classification, we followed the same experimental setup as CRU (Schirmer et al., 2022)[9]. For the human activity classification task, we used the setup described in mTAND (Shukla & Marlin, 2021)[10]. For a fair comparison, we kept the number of model parameters similar to mTAND for the Human Activity dataset and CRU for the other datasets. The model was trained using the Adam optimizer (Diederik, 2014) in all experiments. For the per-point classification and regression tasks, we applied a weight decay of $1 \times 10^{-2}$ and applied gradient clipping for classification task, while no weight decay was used for other tasks. Additionally, to prevent overfitting, we limited the training epochs to 200 for the Physionet extrapolation experiments. The remaining training hyper-parameters are detailed in Table 4.

To estimate the objective function (28), for the decoder $p_\psi(\cdot|\mathbf{y}) = \mathcal{N}(\mathbf{p}_\psi(\mathbf{y}), \boldsymbol{\Sigma}_p)$, we used a Gaussian likelihood with a fixed variance of $\boldsymbol{\Sigma}_p = 0.01 \cdot \mathbf{I}$, following the approach outlined in (Rubanova et al., 2019) for the Pendulum, USHCN, and Physionet datasets. For the Human Activity dataset, we employed a categorical likelihood. Additionally, for $\mathcal{L}(\alpha)$, the initial conditions $(\mathbf{m}_0, \boldsymbol{\Sigma}_0)$ of the latent state were trainable parameters, initialized randomly. The covariance $\boldsymbol{\Sigma}_0$ was set using an exponential transformation, with a small constant ($\epsilon = 10^{-6}$) added to ensure positivity. The latent state

---

[7]https://data.ess-dive.lbl.gov/view/doi:10.3334/CDIAC/CLI.NDP019 under CC BY 4.0

[8]https://physionet.org/content/challenge-2012/1.0.0/ under ODC-BY 1.0

[9]https://github.com/boschresearch/Continuous-Recurrent-Units under AGPL-3.0 license

[10]https://github.com/reml-lab/mTAN under MIT license

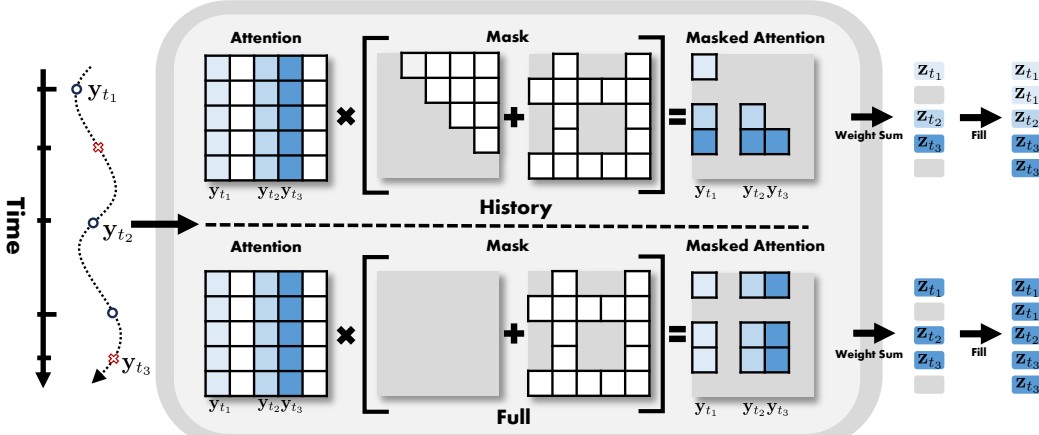

Figure 4: **Illustraion of Masked Attention**. Given the observed time stamps $\mathcal{T} = \{t_i\}_{i=1}^3$ ($\circ$) and the unseen time stamps $\mathcal{T}_u$ ($\times$), attention scores are computed based on the latent observations $\{\mathbf{y}_t\}_{t \in \mathcal{T}}$. **(Up)** The mask of the **History assimilation scheme** consists of two components: one masks attention scores related to future time stamps, and the other masks those related to unseen time stamps. **(Down)** The mask of the **Full assimilation scheme** only blocks the attention scores corresponding to unseen time stamps. Using these masks, masked attention is calculated. For **History assimilation scheme**, the latent variables $\mathbf{z}_{t_i}$ include information up to time $t_i$, while for **Full assimilation scheme**, $\mathbf{z}_{t_i}$ incorporates all available information. Finally, latent variables corresponding to unseen time stamps are filled with the nearest past latent variable value.

---

**Algorithm 3** Training ACSSM

---

**Input.** Time-series $\mathbf{o}_{t \in \mathcal{T}'}$ observed over the entire time stamps $\mathcal{T}' = \mathcal{T} \cup \mathcal{T}_u = \{t_1, \cdots, t_{k'}\}$ with the observed time stamps $\mathcal{T}$ and unseen time stamps $\mathcal{T}_u$, encoder neural network $q_\psi$, decoder neural network $p_\phi$, trainable latent parameters $(\mathbf{m}_0, \mathbf{\Sigma}_0)$ and neural networks $\mathbf{f}_\theta$ and $\mathbf{T}_\theta$.
**for** $m = 1, \cdots, M$ **do**
    Compute $q_\phi(\mathbf{y}_{t \in \mathcal{T}} | \mathbf{o}_{t \in \mathcal{T}})$ by using (25) on observed time stamps $\mathcal{T}$.
    Sample latent observations $\mathbf{y}_{t \in \mathcal{T}} \sim q_\phi(\mathbf{y}_{t \in \mathcal{T}} | \mathbf{o}_{t \in \mathcal{T}})$
    **Parallel computation of objective function** $\mathcal{L}(\alpha)$ **in (16)**
    **if** *history assimilation* **then**
        Estimate latent variables $\mathbf{z}_{t \in \mathcal{T}'}$ such that $\mathbf{z}_t = \mathbf{T}_\theta(\mathcal{H}_t)$ illustrated in Fig (D.2)
    **else if** *full assimilation* **then**
        Estimate latent variables $\mathbf{z}_{t \in \mathcal{T}'}$ such that $\mathbf{z}_t = \mathbf{T}_\theta(\mathcal{H}_T)$ illustrated in Fig (D.2)
    **end if**
    Compute $\mathbf{A}_i = \mathbf{f}_\theta(\mathbf{z}_{t_i})$ and $\alpha_i = \mathbf{B}_\theta \mathbf{z}_{t_i}$ for all $i \in [1 : k']$ by using (22).

    Estimate $\{\mathbf{m}_t, \mathbf{\Sigma}_t\}_{t \in \mathcal{T}'}$ by parallel scan algorithm described in Appendix C.
    Sample $\mathbf{X}_{t \in \mathcal{T}'}^\alpha \overset{i.i.d}{\sim} \otimes_{t \in \mathcal{T}'} \mathcal{N}(\mathbf{m}_t, \mathbf{\Sigma}_t)$ on entire time stamps $\mathcal{T}'$.
    Compute $\tilde{\mathcal{L}}(\theta) = \mathbb{E}\left[\sum_{i=1}^k \left(\frac{(t_i - t_{i-1})}{2} \|\alpha_i\|^2 - \log g_i(\mathbf{y}_{t_i} | \mathbf{X}_{t_i}^\alpha)\right)\right]$ by using (16)
    Sample latent predictions $\tilde{\mathbf{y}}_{t \in \mathcal{T}'} \sim g(\mathbf{y}_{t \in \mathcal{T}'} | \mathbf{X}_{t \in \mathcal{T}'}^\alpha)$ on entire time stamps $\mathcal{T}'$.

    **do in parallel**

    Compute $p_\psi(\mathbf{o}_{t \in \mathcal{T}'} | \tilde{\mathbf{y}}_{t \in \mathcal{T}})$ by using (26) on entire time stamps $\mathcal{T}'$
    Optimize ELBO$(\psi, \phi, \theta)$ by using (28) with gradient descent.
**end for**

---

transition matrix $\mathbf{A}(\cdot)$ was initialized as $\mathbf{A} = \sum_{l=1}^L \mathbf{E} \mathbf{D}_l \mathbf{E}^\top$, where $\mathbf{E}$ was initialized orthonormally following (Lezcano Casado, 2019), and the diagonal matrices $\{\mathbf{D}_l\}_{l \in [1:L]}$ were initialized randomly and passed through a negative exponential to keep the values negative. For the potentials $\{g_i\}_{i \in [1:k]}$, we used a Gaussian likelihood $g_t(\cdot | \mathbf{x}) = \mathcal{N}(\cdot | r_t(\mathbf{m}_t, \mathcal{H}_t), \mathbf{\Sigma}_g)$, where $r_t(\mathbf{x}) = \mathbf{M}_\theta \mathbf{m}_t + \mathbf{z}_t$ with a trainable matrix $\mathbf{M}_\theta$, transformer output $\mathbf{z}_t = \mathbf{T}_\theta(\mathcal{H}_t)$, and a covariance $\mathbf{\Sigma}_t$.

---

**Algorithm 4** Inference with ACSSM

---

**Input.** Time-series $\mathbf{o}_{t\in\mathcal{T}}$ observed only on the observed time stamps $\mathcal{T}$, unseen time stamps $\mathcal{T}_u$ trained model parameters $\{\phi, \psi, \theta\}$.

**Parallel Sampling of latent dynamics**

Sample latent observations $\mathbf{y}_{t\in\mathcal{T}} \sim q_\phi(\mathbf{y}_{t\in\mathcal{T}}|\mathbf{o}_{t\in\mathcal{T}})$ by using (25) on observed time stamps $\mathcal{T}$.

**if** *history assimilation* **then**

    Estimate latent variables $\mathbf{z}_{t\in\mathcal{T}'}$ such that $\mathbf{z}_t = \mathbf{T}_\theta(\mathcal{H}_t)$ illustrated in Fig (D.2)

**else if** *full assimilation* **then**

    Estimate latent variables $\mathbf{z}_{t\in\mathcal{T}'}$ such that $\mathbf{z}_t = \mathbf{T}_\theta(\mathcal{H}_T)$ illustrated in Fig (D.2)

**end if**

Compute $\mathbf{A}_i = \mathbf{f}_\theta(\mathbf{z}_{t_i})$ and $\alpha_i = \mathbf{B}_\theta \mathbf{z}_{t_i}$ for all $i \in [1:k']$ by using (22).

Estimate $\{\mathbf{m}_t, \mathbf{\Sigma}_t\}_{t\in\mathcal{T}'}$ with parallel scan algorithm described in Appendix C.

Sample $N$ trajectories $\mathbf{X}_{t\in\mathcal{T}'}^{\alpha,n} \overset{i.i.d}{\sim} \otimes_{t\in\mathcal{T}}\mathcal{N}(\mathbf{m}_t, \mathbf{\Sigma}_t)$ on entire time stamps $\mathcal{T}'$.

Sample latent predictions $\tilde{\mathbf{y}}_{t\in\mathcal{T}'}^n \sim g(\mathbf{y}_{t\in\mathcal{T}'}|\mathbf{X}_{t\in\mathcal{T}'}^{\alpha,n})$ on entire time stamps $\mathcal{T}'$.

Sample predictions $\tilde{\mathbf{o}}_{t\in\mathcal{T}'}^n \sim p_\psi(\mathbf{o}_{t\in\mathcal{T}'}|\tilde{\mathbf{y}}_{t\in\mathcal{T}'}^n)$ on entire time stamps $\mathcal{T}'$.

(right brace) **do in parallel**

---

**Architecture** In all experiments except for the Pendulum dataset, the time series $\mathbf{o}_{0,T}$ was provided with the observation mask concatenated. We was used a dropout rate of 0.2 for a Human Activity, while no dropout rate was used for the other experiments. For our method, the networks used for each dataset are listed in below, where `d` is the dimension of latent space $\mathbb{R}^d$ as described in Table 4.

- **Human Activity** (Input size, `I=12`)

- Encoder network $\mathbf{q}_\phi$: `Input(2×I) → Linear(d) → ReLU() → Linear(d)`
- Transformer $\mathbf{T}_\theta$: `Input(d) → Masked Attn(d) → FFN(d) → GRU(d)`
- Weight network $\mathbf{f}_\theta$: `Input(d) → Linear(d) → Softmax()`
- Decoder network $\mathbf{p}_\psi$: `Input(d) → Linear(d)`

- **Pendulum** (Input size, `I=1×24×24`)

- Encoder network $\mathbf{q}_\phi$: `Input(I) → Conv2d(1, 12, kernel_size=5, stride=4, padding=2) → ReLU() → MaxPool2d(kernel_size=2, stride=2 → Conv2d(12,12, kernel_size=3, stride=2, padding=1) → ReLU() → MaxPool2d(kernel_size=2, stride=2) → Flatten(108) → Linear(d) → ReLU()`
- Transformer $\mathbf{T}_\theta$: `Input(d) →` $4 \times$ `[Masked Attn(d) → FFN(d)]` `→ GRU(d)`
- Weight network $\mathbf{f}_\theta$: `Input(d) → Linear(d) → Softmax()`
- Decoder network $\mathbf{p}_\psi$: `Input(d) → Linear(d) → ReLU() → Linear(d)`

- **USCHCN** & **Physionet** (Input size, `I=5` for USHCN and `I=37` for Physionet)

- Encoder network $\mathbf{q}_\phi$: `Input(2×I) → Linear(d) → ReLU() → LayerNorm(d) → Linear(d) → ReLU() → LayerNorm(d) → Linear(d) → ReLU() → LayerNorm(d) → Linear(d)`
- Transformer $\mathbf{T}_\theta$: `Input(d) →` $4 \times$ `[Masked Attn(d) → FFN(d)]` `→ GRU(d)`
- Weight network $\mathbf{f}_\theta$: `Input(d) → Linear(d) → Softmax()`
- Decoder network $\mathbf{p}_\psi$: `Input(d) → Linear(d) → ReLU() → Linear(d) → ReLU() → Linear(d) → ReLU() → Linear(d)`

- FFN

- `Input(d) → LayerNorm(d) → Linear(d) → GeLu() → Linear(d) → Residual(Input(d))`

- Masked Attn

- `Input(Q, K, V, mask=M) → Normalize(Q) → Linear(Q) → Linear(K) → Linear(V) → Attention(Q, K) → Masking(M) → Softmax(d) → Dropout() → Matmul(V) → LayerNorm(d) → Linear(d) → Residual(Q)`

