# OpenReview forum: "Amortized Control of Continuous State Space Feynman-Kac Model for Irregular Time Series"
_ICLR.cc/2025/Conference — ICLR 2025 Oral_

### Official Review · Reviewer_vbhs · 2024-10-17

**Soundness:** 3
**Presentation:** 3
**Contribution:** 3
**Rating:** 8
**Confidence:** 3

**Summary:**

The author models irregular time series through a latent controlled SDE. The author first showed that the problem can be solved by a conditioned SDE, and then established the link between such conditioned SDE to a controlled SDE that optimizes a specific form of cost function. Finally, solving the HJB equation for the cost function can be obtained by simulating the controlled SDE through the Feynman-Kac theorem. The simulation was then simplified by analytically solving for the first and second moments at each timestep. Overall the paper is theoretically well-motivated with good empirical results.

**Strengths:**

1. The paper is theoretically motivated with novel ideas and methods.
2. The presentation of the paper is clear.
3. The paper provides improvements on empirical results.
4. The paper provides clear instructions on replicating the experiment.

**Weaknesses:**

1. The motivation for specific implementation details is not clear

**Questions:**

1. Can the authors elaborate on the choice of the cost function in Eq(7)? How should one interpret this cost function? Aside from the theoretical benefit, is there any intuition on the choice of this cost function?
2. Can the author explain the usage of full attention in this scenario? If the full attention is applied to estimate the latent dynamic, does it mean the controls $\alpha_{t_i}$ for the controlled SDE is informed by the future y-observations $o_{t_j}, j > i$? Intuitively the masked attention makes sense to me, but I am unsure about the application of full attention.
3. It seems like the main experiments are all obtained with the full attention scheme. Can the author provide the result of those experiments using masked attention as an ablation study? When is the masked attention used?

---

> ### Author Response · Authors · 2024-11-16
> **Author Response to Reviewer vbhs**
>
> We thank the reviewer for appreciating our research and acknowledging its significance. The questions raised have been addressed in the responses provided below.
>
> ----
>
> **1. Motivation for specific implementation.**
> * To address the reviewer’s concern, we have included a brief note on the key concepts and related works in Appendix A. We hope that this section helps improve the reviewer’s understanding of our approach. If there are still specific aspects or details the reviewer’s find unclear, please let us know, and we would be happy to address them further in the revised version.
>
> ----
>
> **2. Cost functional for SOC problem.**
> * The cost function in Eq (7) is central to our approach as it serves to balance two objectives: minimizing control effort and aligning the generated dynamics with observed data. The intuition behind this choice comes from SOC theory, where the goal is to control a system's trajectory in such a way that it closely follows a desired path (in our case, the latent dynamics inferred from observations) while minimizing control energy.
> * The first term, $\frac{1}{2}||\alpha_t||^2$, represents the control energy, which penalizes large deviations in control inputs for **the principle of least action** to regularize the control effort, thereby encouraging smoother trajectories. In practice, we can interpret it as a regularization term to help generalization by discouraging overly complex control signals and ensuring that the model relies on inherent patterns in the data rather than excessively adjusting controls to fit noise.
> * The second term, $-\log f_i(\mathbf{y}_{t_i} | \cdot)$, ensures that the controlled dynamics are consistent with the observed data points over the time interval [0, T] by adjusting the dynamics to maximize the likelihood of observations.
> * This interpretation is closely related to variational inference, where the objective is to approximate a target posterior distribution by optimizing a variational distribution while including a regularization term (often a Kullback-Leibler divergence). In our formulation, we extend this approach to the path space, where the prior distribution is given by the dynamics in Eq (1), and the target posterior distribution is defined in Eq (2). Here, the control function \alpha plays the role of adjusting the prior dynamics to approximate the posterior path measure, in SOC problem Eq (7) (technically Eq (16)) often referred to as KL-control problem, where the first term acts as KL-regularization term while the second term acts as target log-likelihood. The cost function in Eq (16) acts as a variational bound to achieve this alignment while maintaining regularization over the entire trajectory.
>
> ----
> **3. Assimilation scheme.**
> * The use of full assimilation means that the controls for the controlled SDE indeed incorporate information over a full observation $\mathbb{o}_{t \in [0, T]}$. The type of assimilation is inspired by filtering/smoothing in the standard SSM algorithm, where the smoothing algorithm typically incorporates full observation. This design choice was made to allow for non-causal data assimilation, which improves the expressiveness and flexibility of our model in capturing long-range dependencies. We provided a conceptual illustration of the proposed information assimilation (full/history) scheme in Figure 4.
> ----
> **4. Use of history attention scheme.**
> * We would like to clarify that, in our implementation, we utilize a history assimilation scheme (which we believe the reviewer refers to as "masked attention") for the sequence extrapolation task, because the task involves forecasting based solely on past data. It prevents the use of the full assimilation scheme, which leverages future observations. With history assimilation, our approach effectively behaves like a traditional filtering method which typically relies on past observation to infer the latent states. As a result, in sequential extrapolation, the performance gap between our model and CRU narrows. Because CRU is genuinely a filtering method that typically relies on past observation and with history assimilation, our approach effectively behaves like a filtering method. In this setting, it is worth noting that our method achieves comparable performance with CRU, while avoiding numerical trajectory inference by leveraging parallel state estimation based on Theorem 3.8.

---

> > ### Comment · Reviewer_vbhs · 2024-11-19
> >
> > Thank you for this timely response. The authors have addressed my questions, and I would like to retain my score.

---

### Official Review · Reviewer_UjeK · 2024-11-04

**Soundness:** 3
**Presentation:** 2
**Contribution:** 3
**Rating:** 8
**Confidence:** 2

**Summary:**

The authors introduce a way of amortizing the controller of a state space model (ACSSM) to make it compatible with irregular time series. To do this, they generalize the single-marginal Doob’s $h$-transform to the multi-marginal case. To simulate the resulting continuous dynamics, they use VI to get an ELBO which is then optimized. To make this tractable, they assume that the latent dynamics are locally linear, and use a neural network to get expressive latent dynamics this way. The authors provide a theoretical analysis for this work, showing that the ELBO they obtain is tight, and then offer several experiments where this method shows improved performance over comparable baselines, for both per time classification/regression as well as sequence interpolation/extrapolation.

**Strengths:**

-Novel method of solving Continuous State Space model which learns linear dynamics to accommodate irregular time series.

-Theoretical analysis for the provided algorithm, including the derivation of an ELBO used to solve the dynamics.

-Demonstration of the algorithm on real time series, showing improved performance over other methods, in both Test MSE and also compute time (<5 secs).

-Addresses limitations, such as the errors accumulated from the linear approximation.

**Weaknesses:**

-Some of the background was a bit hard to follow. As someone who is relatively unfamiliar with the literature in this area, I found that some of the stuff in the methods section were not explained (it’s possible that it was common knowledge). I tried looking at the appendix for a more fleshed out explanation and i still can’t say I’m confident I understand everything going on.

-The paper argues that this method is scalable, and one thing I wish I could understand better is how this would work on environments with more complex dynamics, and possibly even partially observable environments. Did you guys try anything along this route?

**Questions:**

-Did you guys try using the affine linear drift function for the latent dynamics? How much better does the learned NN drift function do?

-It is sometimes unclear where you are using the full and when you are using the history attention mechanism. It is claimed that the authors perform better than CRU because they aren’t just using past information, but also future information. However, this seems like an unfair comparison, am I misunderstanding something?

-How much does the "Parallel Computation" stuff increase the speed of inference? Is this a major contributor of why the method is so fast compared the other methods.

---

> ### Author Response · Authors · 2024-11-16
> **Author Response to Reviewer UjeK**
>
> We sincerely appreciate the reviewer’s interest in our research and acknowledgment of its significant contributions. We have addressed the concerns raised by the reviewer below.
>
> ----
>
> **1. Background**
>
> * We have added further explanations in Appendix A to provide a deeper understanding of the theoretical foundations. We hope these revisions help clarify the concepts and address the reviewer’s concerns.
>
> ----
>
> **2. Scalability on more complex dynamics**
>
> * Thank you for your insightful suggestions to improve our work. We have addressed this concern in the general response. For a more detailed explanation, kindly refer to that section.
>
> ----
>
> **3. Neural network drift function**
>
> * In general, using an NN drift can be effective for modeling complex latent dynamics. However, employing an NN drift requires a numerical solver, which can lead to instability as the time-series length increases [1]. In such cases, we expect simplifying the dynamics to derive a closed-form solution can actually be more beneficial, as it leads to more stable learning. We believe that our experiments demonstrate this by showing superior performance compared to other baselines that utilize NN drift functions, as highlighted in the Human Activity Classification task in Section 4.1.
>
> ----
>
> **4. Assimilation schemes**
>
> * With the exception of specific settings such as sequence extrapolation, it is generally expected that utilizing all available data for predictions will result in better performance. In our experiments on classification and regression in Section 4.1, all methods, except for CRU, leverage this advantage. In cases where future data is accessible, not utilizing it for predictions, as in the case of CRU, could lead to inefficiencies.
>
> * In this regard, we do not see this as an unfair comparison, but rather as an indication of the inherent limitations in CRU's modeling approach, which presents a challenge that needs to be addressed. In contrast, our method introduces a control formulation that offers an effective solution to overcome this limitation
>
> ----
>
> **5. Parallel computation**
>
> The key factors that make our algorithm faster compared to other methods are: (1) the use of locally linear dynamics with a diagonalizable matrix $A$, and (2) the incorporation of a parallel scan algorithm.
> * **(1)** While locally linear dynamics alone do not directly guarantee faster inference, they do provide computational advantages compared to neural differential equation models. By using locally linear dynamics, we can simplify the heavy numerical simulations often required, reducing the problem to two ODEs as described in equations (18-19). This allows us to leverage efficient matrix operation tricks, making computations more efficient. However, it is important to note that this approach still involves ODE integration and matrix operations, which can be computationally intensive.
> * **(2)** The significant speedup in our method comes from applying the parallel scan algorithm, particularly in cases where the matrix $A$ is diagonalizable, as shown in Theorem 3.8. Typically, inferring the Bayesian posterior distribution requires $\mathcal{O}(k)$ computation for $k$ observations due to the sequential update. By leveraging the parallel scan algorithm, we can reduce this time complexity from $\mathcal{O}(k)$ to $\mathcal{O}(\log k)$, allowing the processing of the data simultaneously. This results in a substantial reduction in computation time, especially for large-scale datasets, making our method significantly faster than others that rely on sequential processing.
>
> ----
>
>     [1] Iakovlev et al., “Latent Neural ODEs with Sparse Bayesian Multiple Shooting”

---

> > ### Author Response · Authors · 2024-11-22
> > **Gentle Reminder**
> >
> > Dear Reviewer UjeK,
> >
> > We were wondering whether our response has sufficiently addressed the reviewer's questions as the discussion period nears its end. If so, we would greatly appreciate it if the reviewer could consider updating the score to reflect this.
> >
> > If the reviewer has any additional comments or questions, please let us know, and we will do our utmost to address them before the deadline. Thank you for your time and consideration.
> >
> > Best regards,
> >
> > Authors

---

> > > ### Comment · Reviewer_UjeK · 2024-11-22
> > >
> > > I sincerely apologize for the delay in replying to this! I appreciate the thoughtful answers to my questions. I went through the updated Appendix covering the related work, and I now believe I have a better understanding of this paper's contributions to the field. I have also been convinced that using the affine drift function is sufficient for modelling these problems (the third bullet point in the general response cleared things up for me). I also did not appreciate the innovation of the parallel scan you guys introduced to reduce the time complexity to $O(\log k)$ upon my first read, but I now see this as one of the main strengths of the paper. With all this being said, I have updated my score.

---

### Official Review · Reviewer_6X8u · 2024-11-05

**Soundness:** 4
**Presentation:** 2
**Contribution:** 4
**Rating:** 8
**Confidence:** 2

**Summary:**

The paper introduces the Amortized Control of Continuous State Space Model (ACSSM) to handle irregularly sampled time-series. ACSSM aims to model the path measure of trajectories in a latent space, conditioned on observations in the data space. By using a latent space, the model captures a flexible and structured representation of the underlying dynamics that generate observed data, which is especially useful for irregularly sampled observations. To construct this conditional path measure, the authors introduce a novel multi-marginal Doob’s h-transform. This extension of the traditional Doob’s transform induces a class of stochastic differential equations (SDEs) that define the desired path measure in the latent space. However, simulating these SDEs directly is computationally infeasible due to the need for expensive normalization constants and conditional expectations. To overcome this challenge, the authors leverage stochastic optimal control to define a variational objective that approximates the optimal control needed to produce the conditioned dynamics. To further enhance computational efficiency, they propose working with affine linear SDEs with known Gaussian perturbation kernels, allowing simulation-free estimation of the latent trajectories and significantly speeding up the inference process.

**Strengths:**

The diffusion and stochastic optimal control (SOC) perspective for time-series modeling offers a compelling alternative to classical recurrent network methods, as it allows for continuous, flexible representations of latent dynamics that are suited to handle irregular timesteps.

The paper demonstrates superior performance on real-world datasets. Additionally, the efficiency gains in training time make it feasible for high-dimensional data, where classical SDE-based methods might be expensive.

**Weaknesses:**

The absence of a Related Work section limits the reader’s ability to understand how ACSSM compares to existing time-series modeling approaches, particularly those used in irregular time-series contexts, and those compared in the experiments section (e.g., recurrent networks, attention mechanisms, and previous SDE-based models).

The paper leans heavily on measure-theoretic concepts and complex SDE formulations, which could make it difficult for readers not specialized in these areas. More accessible explanations or visual intuitions could enhance understanding.

It remains to be seen how the model scales in practice with very high-dimensional or complex latent spaces, as affine SDE simplifications may reduce the expressiveness of the dynamics in these settings.

**Questions:**

- The introduction lists the contributions, but it could benefit from more intuition to guide the reader through the chain of thought. For instance, there is no explanation of what a Feynman-Kac model is or how it facilitates sequential analysis.

- The multi-marginal Doob’s  h-transform is a central component of the approach, but its presentation lacks intuitive guidance, which is reflective of the overall style in the paper. Adding more accessible explanations would enhance understanding.

- Some sentences have syntax errors or missing words. I recommend proofreading the text to improve readability.

---

> ### Author Response · Authors · 2024-11-16
> **Author Response to Reviewer 6X8u**
>
> We sincerely appreciate the reviewer's interest in our research and acknowledgment of its significant contributions. We have provided detailed responses in the subsequent.
>
> ----
>
> **1. Related Works and intuition of background.**
>
> * In response to the reviewers' suggestions, we have clarified key concepts (such as probabilistic SSMs, the Feynman-Kac model, and Doob’s h-transform) in the revised manuscript. Additionally, we have added the related work to include relevant literature. For more details, please refer to Appendix A of the revised version. We hope this addresses the reviewer’s concerns and provides the clarity the reviewer was seeking.
>
> ----
>
> **2. How the model scales in practice with very high-dimensional or complex latent spaces.**
>
> * Thanks for the valuable suggestion to enhance our work. We have addressed the reviewer’s point in the general responses. Kindly refer to that part for our detailed answer.
>
> ----
>
> **3. Sentence.**
>
> * We appreciate the feedback regarding readability. We will conduct a thorough proofreading pass to address any syntactic errors, enhance sentence flow, and improve overall clarity.

---

> ### Author Response · Authors · 2024-12-04
> **Gentle Reminder**
>
> Dear Reviewer 6X8u,
>
> As the discussion period is coming to a close, we kindly invite you to share your feedback on our response and consider revising your score if you find it appropriate.
>
> Best regards,
>
> Authors

---

### Official Review · Reviewer_jj93 · 2024-11-11

**Soundness:** 4
**Presentation:** 4
**Contribution:** 3
**Rating:** 8
**Confidence:** 2

**Summary:**

The authors propose ACSSM approach for modeling irregular time series which uses continuous-discrete state space models (CD-SSMs).  The authors extend doob's-h transform to multi-marginal case and solve the problem using stochastic optimal control. The formulation leads to a  evidence lower bound (ELBO) and the authors propose a VI based loss function to model the irregular time series.

Authors make linear approximation for the SDEs, which allows them to be able to perform simulation free estimation and exploit transformers for parallel computations. Overall, the proposed approach is faster and performs better than existing simulation based approaches, neural differential equations and other RNN based approaches tailored to handle irregular time series.

**Strengths:**

- The paper makes significant theoretical contributions which should be relevant other researchers in this field.
- The proposed approach has impressive empirical results showing better results with faster training while being theoretically grounded.

**Weaknesses:**

One of the weakness of this work that I see is that it makes several simplifying approximations to make the solution faster/tractable. Authors already acknowledge this but it would have been nice to understand if there are any practical trade-off due to these approximation.

**Questions:**

- How many learnable parameters are use for each method? Was each method trained with a similar paramter budget?
- Given several approximations, how does the method perform on larger datasets ?

---

> ### Author Response · Authors · 2024-11-16
> **Author Response to Reviewer jj93**
>
> We sincerely appreciate the reviewer's interest in our work and recognition of its contributions. Detailed responses are provided in below.
>
> ----
>
>
> **1. Performance on larger datasets.**
>
> * We thank the reviewer for the valuable suggestions to enhance our work. This point has been addressed in the general responses section. Please refer to that section for a detailed explanation.
>
> **2. Parameter budget.**
>
> * To ensure a fair comparison, all methods, including ACSSM and the baseline models, were trained using a comparable parameter budget. As outlined in Table 4, we maintained similar parameter counts to those of the baseline methods. For instance, we matched the parameter count to [1] for the Human Activity dataset and to [2] for the other datasets.
>
> ----
>     [1] Shukla & Marlin, “ Multi-time attention networks for irregularly sampled time series.”
>     [2] Schirmer et al., “Modeling irregular time series with continuous recurrent units”

---

> > ### Comment · Reviewer_jj93 · 2024-11-30
> >
> > Thank you for the response. I am somewhat convinced by the arguments presented to support the usefulness of the method for even larger-scale datasets. But I feel experimental validation is still necessary to support these claims.
> >
> > However, this paper makes other significant contributions to warrant publication, so I will maintain the score.

---

### Author Response · Authors · 2024-11-16
**Author response to all reviewers**

We sincerely thank the reviewers for thoughtful feedback. We are pleased that all reviewers recognized the importance of our theoretical contributions, and acknowledged the novelty and impact of our proposed method. We are also encouraged by the multiple reviewers who found the paper to be clearly presented and well-structured.


We believe our work makes a significant contribution by integrating stochastic processes and stochastic optimal control frameworks to advance continuous state space modeling. While our primary focus is on addressing the challenges associated with irregular time-series data, we believe our approach has the potential to benefit a broad range of sequential data applications.

----
Although all reviewers acknowledged the technical novelty of our work, they raised concerns regarding (1) the scalability to larger datasets and (2) the insufficient clarification of key theoretical concepts. In response to their valuable feedback, we have provided some general responses below.


**1. The scalability to larger datasets.**

* While approximations (such as locally linear approximation) may seem to limit the expressiveness of the model, our empirical results indicate that, for the datasets we tested (e.g., Physionet, USHCN, and Human Activity), which are (we believe) already large-scale (e.g., Physionet has 8000 observations with 37 dim) real-world datasets benchmarked by several prior works, it does not lead to any sacrifice in model performance. Moreover, the Pendulum dataset contains 4,000 observations with a 576 dimension (24x24 pixels), and exhibits partial observability due to being corrupted by a correlated noise process, as illustrated in Figure 3.

* This is because, by leveraging amortized inference, our approach efficiently scales to high-dimensional and complex latent spaces by decoupling representation learning from the latent dynamics. This helps mitigate some of the trade-offs associated with these approximations, thereby preserving both efficiency and accuracy.

* Additionally, while linear approximations might appear to reduce expressiveness, state-space models utilizing linear dynamics have already been successfully applied in [1] to large-scale datasets. Moreover, [2] observed that the performance of generative models may depend not so much on the linearity of the forward process, but rather on the complexity of the backward generation process, this aligns with our observation that linearizing the prior distribution is less critical compared to capturing the complexity of the posterior distribution. To address this, we leverage the expressiveness of transformer architectures to achieve sufficient flexibility in modeling the posterior.

* For more complex modeling needs, we believe that utilizing more suitable architectures will be the key to further improvements. We believe that the balance we have struck between computational efficiency and model flexibility can extend to even more complex time-series data, similar to these successful cases. In fact, we plan to adapt our method to large-scale medical datasets in future work.

**2. Insufficient clarification of key theoretical concepts.**

* In line with the reviewer’s suggestion, we have included a comprehensive Related Work section and added further brief explanations in **Appendix A** in the revision. We sincerely hope this section will improve the grasp of both the reviewers and potential readers, thereby increasing their confidence in understanding our paper.

----
Notable changes are highlighted in magenta in the revised manuscript. We have made every effort to address all concerns raised, as detailed in the individual responses below.

----

    [1] Smith et al., “Simplified state space layers for sequence modeling”
    [2] Deng et al., “Variational Schrödinger Diffusion Models”

---

### Meta-Review · Area_Chair_fQVd · 2024-12-17

**Metareview:**

This paper proposes the Amortized Control of continuous State Space Model for continuous dynamical modeling of time series for irregular and discrete observations. It extends Doob's h-transform to the multi-marginal setting, and defines a variational inference algorithm with a tight ELBO. To speed up training and inference, ACSSM assumes locally linear latent dynamics and employs transformers for parallel computation.

All reviewers praise its novel and significant theoretical contribution to modeling continuous state space with irregular observations. The empirical studies on real-world datasets show superior performance in both computational efficiency and accuracy.

I recommend accepting this paper based on the consensus from reviewers.

**Additional Comments On Reviewer Discussion:**

Reviewers had some shared concerns on the scalability of the proposed method, the potential weakness of the local linearity assumption, and the readability of the manuscript due to heavy theoretical concepts. The authors rebuttal provides more clarification in the appendix and arguments about the model's flexibility and evidence of its scalability in the existing experiments. Those concerns have been mostly addressed. One reviewer would still like to see experiments on even-larger datasets to show its scalability.

---

### Decision · Program_Chairs · 2025-01-22

Accept (Oral)